# ☀ Aurora: towArds Universal geneRative multimOdal time seRies forecAsting

**Xingjian Wu, Jianxin Jin, Wanghui Qiu, Peng Chen, Yang Shu, Bin Yang, Chenjuan Guo**[✉]
East China Normal University
{xjwu,jxjin,onehui,pchen}@stu.ecnu.edu.cn,
{yshu,byang,cjguo}@dase.ecnu.edu.cn

## Abstract

Cross-domain generalization is very important in Time Series Forecasting because similar historical information may lead to distinct future trends due to different domain-specific characteristics. Recent works focus on building unimodal time series foundation models and end-to-end multimodal supervised models. Since domain-specific knowledge is often contained in modalities like texts, the former lacks the explicit utilization of them, thus hindering the performance; and the latter is tailored for end-to-end scenarios and does not support zero-shot inference for cross-domain scenarios. In this work, we introduce Aurora, *the first Multimodal Time Series Foundation Model*, which supports multimodal inputs and zero-shot inference. Pretrained on Cross-domain Multimodal Time Series Corpus, Aurora adaptively extracts and focuses on key domain knowledge contained in corresponding text or image modalities, thus possessing strong cross-domain generalization capability. Through tokenization, encoding, and distillation, Aurora extracts multimodal domain knowledge as guidance and then utilizes a Modality-Guided Multi-head Self-Attention to inject them into the modeling of temporal representations. In the decoding phase, the multimodal representations are used to generate the conditions and prototypes of future tokens, contributing to a novel Prototype-Guided Flow Matching for generative probabilistic forecasting. Comprehensive experiments on 5 well-recognized benchmarks, including TimeMMD, TSFM-Bench, ProbTS, TFB, and EPF, demonstrate the consistent state-of-the-art performance of Aurora on both unimodal and multimodal scenarios.

○ https://github.com/decisionintelligence/Aurora.
🤗 https://huggingface.co/DecisionIntelligence/Aurora.

## 1 Introduction

Time series forecasting has gained sustained attention for decades of years due to its significant values in multiple domains, including economy, transportation, meteorology, and public health. In recent years, the key pivot comes with the surge of deep learning, which brings the boom of merticulously-designed deep forecasting models (Cirstea et al., 2022; Nie et al., 2023; Qiu et al., 2025e; Wu et al., 2025d). Through learning the inherent dynamics within the raw data, deep learning models can outperform classic statistical methods (Box & Pierce, 1970; Mei et al., 2014) and obey the scaling law (Shi et al., 2024a; Yao et al.). Due to the success, it also brings the most commonly-used forecasting paradigm, which utilizes the past information to infer how the series goes in the coming horizon. Although this paradigm contributes to impressive performance under domain-specific scenarios, its effectiveness is uncertain when facing cross-domain inference, where *similar historical information may lead to different futures due to domain differences.*

As shown in Figure 1, current research of time series forecasting explores the cross-domain adaptation in two main perspectives: 1) pre-training on cross-domain time series corpus for unimodal time series foundation models, which partially possess cross-domain generalization capabilities;

2) utilizing cross-modality information in training end-to-end multimodal supervised models, which effectively integrates domain knowledge in forecasting. For time series foundation models, the cross-domain generalization capabilities mainly come from the sensitivity to subtle differences in historical information from different domains. Some of them (Shi et al., 2024b; Liu et al., 2025e) are pretrained on trillion-scale corpus with heavy backbones, thus possessing certain cross-domain adaption capabilities. Others (Wang et al., 2025f;e; Wu et al., 2025a) have specific structures, which excels at capturing cross-domain features. However, their capabilites come from single time modality and lack explict domain knowledge guidance, thus hindering the performance. For end-to-end multimodal supervised models (Jin et al., 2024; Liu et al., 2025b), though they consider the multimodal knowledge to enhance the domain-specific forecasting, they lack the ability to support zero-shot forecasting in cross-domain scenarios. In our view, the *aurora* of

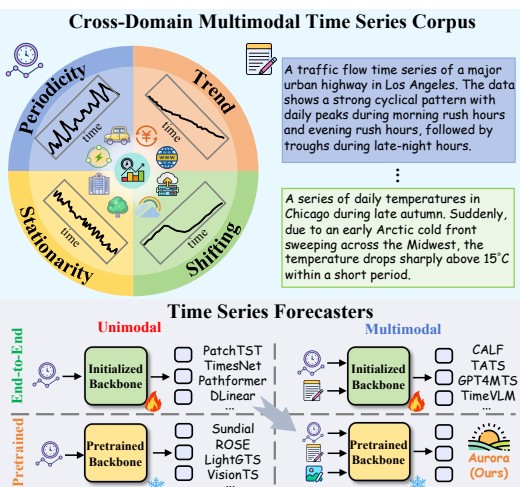

Figure 1: Aurora is pretrained on *cross-domain multimodal* time series corpus, supporting both text and image information to enhance zero-shot time series forecasting.

next-generation time series foundation model lies in pretraining a cross-modality model on cross-domain time series corpus, which can *utilize the domain knowledge within modalities and serve as a versatile out-of-the-box forecaster in complex scenarios.*

See Figure 1, we propose **Aurora**, which pioneers the exploration of multimodal time series foundation model. Specifically, we pretrain Aurora on Cross-Domain Multimodal Time Series Corpus, with time series data and sample-wise, domain-specific text descriptions. Since previous works (Chen et al., 2024a; Yu et al., 2025c) point out the endogenous images of time series contain additional geometric information, we also consider them into cross-modality learning. Considering the model architecture, Aurora adopts a novel cross-modality Encoder. Taking pretrained Bert (Devlin et al., 2019) and ViT (Liu et al., 2021) as modality encoders, Aurora then adopts token distillation to extract the key information in different modalities. To effectively model the cross-modality interaction, we propose a novel *Modality-Guided Self-Attention Mechanism* to utilize the external domain knowledge to adjust the attention of internal information within the time series data to obtain temporal features, and then fuse them with text and image features.

In the Aurora Decoder, we devise a novel flow-matching to fully utilize the fused cross-modality features to support multimodal cross-domain generative probabilistic forecasting. First, we use a ConditionDecoder to generate multimodal conditions for flow matching. Since the future trend of time series is often implied by external text information, and the inherent periodicity of time series is often contained in the endogenous images, we then design a Prototype Bank initialized by Period and Trend prototypes, and leverage a PrototypeRetriever to retrieve the "future prototypes" based on the inherent domain knowledge from texts and images. Compared with DDPM (Ho et al., 2020), Flow Matching (Lipman et al., 2023) serves as a stochastic interpolant, which can start from a random distribution instead of a Gaussian noise, with more flexibilities. We take the generated future prototypes as starting points, which contains the rudiments of periodicity and trend for future tokens, thus can simplify the flow matching process. Our contributions are summarized as follows:

- We propose a multimodal time series foundation model, called Aurora, which is pretrained on cross-domain multimodal time series corpus and supports generative probabilistic forecasting. Through effectively fusing multimodal information during pretraining, Aurora serves as a strong zero-shot forecaster, and can make accurate cross-domain inference.

- We devise a novel cross-modality encoder in Aurora, consisting of token distillation and modality guiding, implemented by merticulously-designed attention structures. It can enhance the temporal representations while effectively fusing representations from texts and images.

- We design a novel flow-matching process in the Aurora Decoder. It obtains multimodal conditions through a Transformer, and obtains future prototypes containing periodic and trend information as the starting points, thus enhancing the ability of flow-matching.

- Experimentally, Aurora achieves state-of-the-art performance on 5 well-recognized benchmarks, including datasets from TimeMMD (Liu et al., 2024b), TSFM-Bench (Li et al., 2025c), ProbTS (Zhang et al., 2024), TFB (Qiu et al., 2024), and EPF (Olivares et al., 2023), covering comprehensive scenarios, i.e., unimodal, multimodal, deterministic, and probabilistic, thus demonstrating a strong out-of-the-box tool of decision intelligence.

## 2  RELATED WORKS

### 2.1  TIME SERIES FORECASTING

Time Series Forecasting is vital in decision-making and has fascinated people for decades of years, which facilitates the emergence of a series of works. In recent years, deep-learning models are widely studied, among them, Autoformer (Wu et al., 2021), Triformer (Cirstea et al., 2022), Times-Net (Wu et al., 2023), Pathformer (Chen et al., 2024b), PatchTST (Nie et al., 2023), Dlinear (Zeng et al., 2023), FiTS (Xu et al., 2024), SparseTSF (Lin et al., 2024a), PDF (Dai et al., 2024a), DUET (Qiu et al., 2025e), and TimeMixer++ (Wang et al.), continuously advancing the state-of-the-arts. However, though they possess the capabilities to extract the inherent dynamics in raw time series data, they only adapt to unimodal end-to-end forecasting scenarios, and often fall short in multimodal forecasting scenarios where the domain knowledge is widely contained in the text modality.

Recently, some works are proposed to explore the multimodal end-to-end supervised models. In summary, they utilize Large Language Models' strong reasoning capabilities to integrate textual domain knowledge to prompt temporal modeling. Among them, Unitime (Liu et al., 2024c), Time-LLM (Jin et al., 2024) and CALF (Liu et al., 2025b) utilize the endogenous textual descriptions as prompts, GTP4MTS (Jia et al., 2024), TATS (Li et al., 2025d) and TimeMMD (Liu et al., 2024b) supports exogenous textual domain knowledge. However, they do not possess generalization capabilities in zero-shot scenarios.

### 2.2  TIME SERIES FOUNDATION MODELS

To support cross-domain generalization, unimodal Time Series Foundation Models are widely studied. The majority of them adopt Tranformer-based architectures, which are pretrained on time series corpus of billion- or trillion- scale to obtain the strong generalization capabilities. Among them, Sundial (Liu et al., 2025e), VisionTS (Chen et al., 2024a), ROSE (Wang et al., 2025f), Light-GTS (Wang et al., 2025e),Time-MoE (Shi et al., 2024b), MOIRAI (Woo et al., 2024), TTM (Ekambaram et al., 2024), Chronos (Ansari et al.), UniTS (Gao et al., 2024), Timer (Liu et al., 2024e), and TimesFM (Das et al., 2024) demonstrate strong zero-shot forecasting performance on unimodal tasks, even outperforming those full-shot supervised models in many cases. Considering the forecasting paradigm, Sundial, MOIRAI, Chronos, and Lag-Llama (Rasul et al., 2023) also support probabilistic forecasting, which provides additional robustness and versatility for decision-making. Despite their endeavors to enhance cross-domain generalization capabilities, when historical series exhibit similarities, the forecasts they generate remain static. This lack of adaptability renders them unable to accommodate the diverse and changing real-world domains.

In this work, we propose Aurora to pioneer the exploration of multimodal time series foundation models. Through pretraining on Cross-Domain Multimodal Time Series Corpus, Aurora can extract the key domain knowledge within the text and image modalities to enhance the modeling of temporal features. Aurora also supports generative probabilistic forecasting, thus covering versatile tasks, including unimodal, multimodal, deterministic and probabilstic forecasting.

## 3  AURORA

In this work, we pretrain Aurora in a cross-modality paradigm, which adopts Channel-Independence (Nie et al., 2023) on time series data, and models corresponding multimodal inter-

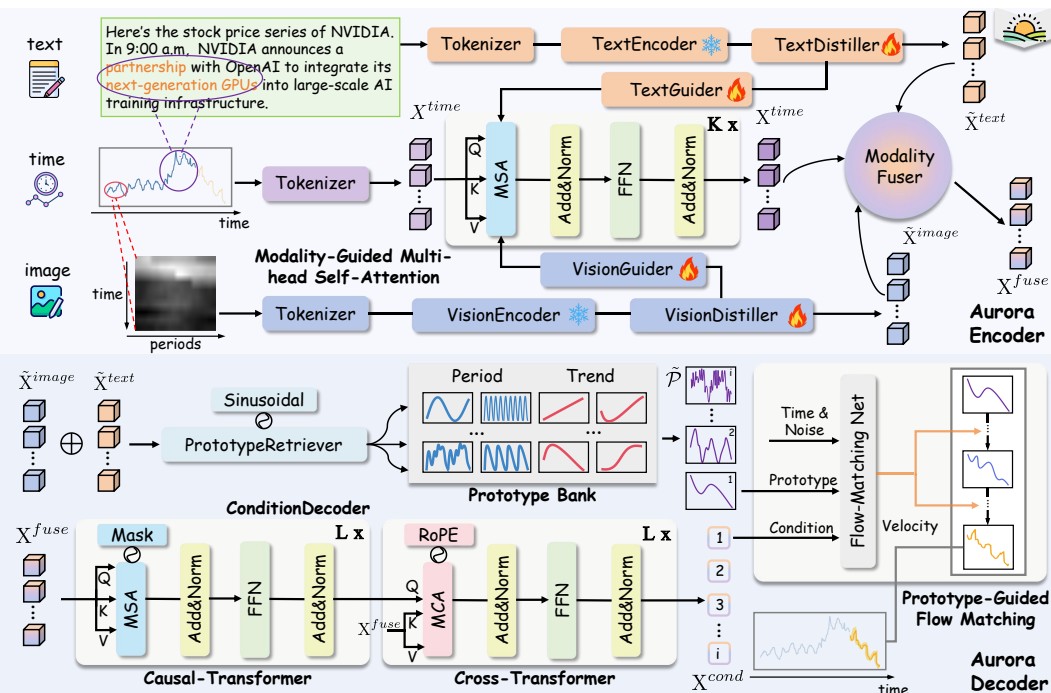

Figure 2: The overview of Aurora. In the Aurora Encoder, the multimodal information is extracted, distilled, and fused. Modality-Guided Multi-head Self-Attention is introduced to inject the domain-specific knowledge into temporal modeling. In the Aurora Decoder, the Prototype-Guided Flow Matching is introduced to support generative probabilistic forecasting.

action to inject domain knowledge. Note that each variable of time series is first normalized through Instance Normalization (Ulyanov et al., 2016) to mitigate the value discrepancy. As shown in Figure 2, Aurora mainly consists of two phases: 1) in Aurora Encoder, we tokenize and encode each modality into modal features, then fuse them to form multimodal representations; 2) in Aurora Decoder, we utilize a Condition Decoder to obtain the multimodal conditions of future tokens, leverage a Prototype Retreiver to retrieve the future prototypes based on the domain knowledge, and conduct flow matching on them to make generative probabilistic forecasts.

## 3.1 ENCODING

### 3.1.1 MULTIMODAL TOKENIZATION

Aurora inherits the strong encoding capabilities from ViT (Liu et al., 2021) and Bert (Devlin et al., 2019) to extract the representations from images and texts, and adopts a temporal Channel-Independent Transformer as the main backbone. Therefore, inputs of all modalities are required to be tokenized first.

Given a univariate time series $X \in \mathbb{R}^T$, we adopt RevIN (Kim et al., 2021) technique to mitigate the inherent non-stationarity of time series. The time series tokens $X^{time}$ are formed through non-overlapped Patching and Embedding (Cirstea et al., 2022; Nie et al., 2023):

$$X' = \text{LeftPad}(X), X^P = \text{Patching}(X') \in \mathbb{R}^{n^{time} \times p^{time}}, \quad (1)$$

$$X^{time} = \text{Embedding}(X^P) \in \mathbb{R}^{n^{time} \times d^{time}}, \quad (2)$$

where Embedding is a linear projection, $X^{time} \in \mathbb{R}^{n^{time} \times d^{time}}$ are the embedded time series tokens, with $n^{time}$ representations of dimension $d^{time}$.

To obtain the endogenous image tokens, we utilize the rendering techniques (Chen et al., 2024a) to make the transformation:

$$\mathcal{A} = \text{Amp}(\text{FFT}(X)), \mathcal{F} = \arg\max(\mathcal{A}), \text{P} = \lceil T/F \rceil, \tag{3}$$

$$\tilde{X} = \text{LeftPeriodPad}(X, \text{P}), X^{2D} = \text{Reshape}(\tilde{X}) \in \mathbb{R}^{m \times \text{P}}, \tag{4}$$

$$X^{3D} = \text{Resize}(\text{Repeat}(X^{2D})) \in \mathbb{R}^{3 \times w \times h}, \tag{5}$$

$$\tilde{X}^{3D} = \text{ImagePatching}(X^{3D}) \in \mathbb{R}^{n^{image} \times 3 \times \frac{w}{p^{image}} \times \frac{h}{p^{image}}} \tag{6}$$

$$X^{image} = \text{Embedding}(\text{Flatten}((\tilde{X}^{3D})) \in \mathbb{R}^{n^{image} \times d^{image}}, \tag{7}$$

where the time series is first processed into 2D structure $X^{2D} \in \mathbb{R}^{m \times \text{P}}$ based on the period P. Then the endogenous image $X^{3D} \in \mathbb{R}^{3 \times w \times h}$ is rendered through repeating $X^{2D}$ along channel dimension, and resizing into the standard input size of ViT. Finally, the image tokens $X^{image} \in \mathbb{R}^{n^{image} \times d^{image}}$ are obtained through ImagePatching and Embedding.

For the corresponding texts, the text tokens $X^{text} \in \mathbb{R}^{n^{text} \times d^{text}}$ can be easily obtained through tokenization and retrievement from the vocabulary of Bert.

### 3.1.2 TOKEN DISTILLATION

After obtaining the tokens from all the modalities, the hidden representations of texts and images are then generated through the pretrained VisionEncoder (ViT) and TextEncoder (Bert):

$$\tilde{X}^{image} = \text{VisionEncoder}(X^{image}) \in \mathbb{R}^{n^{image} \times d^{image}}, \tag{8}$$

$$\tilde{X}^{text} = \text{TextEncoder}(X^{text}) \in \mathbb{R}^{n^{text} \times d^{text}} \tag{9}$$

Inituitively, there exists informative redundancy in texts and images for multimodal time series forecasting. For texts as additional domain knowledge, key descriptions which can affect the future trend of time series often deserve only several words. For the endogenous image, we consider it as a technique to extract the varying inherent periodic information in time series data from multiple domains, where the information is also sparse. Therefore, we distill the tokens from text and image modalities to extract the key information and improve the efficiency:

$$\text{X}^{image} = \text{VisionDistiller}(R^{image}, \tilde{X}^{image}) \in \mathbb{R}^{K^{image} \times d^{image}}, \tag{10}$$

$$\text{X}^{text} = \text{TextDistiller}(R^{text}, \tilde{X}^{text}) \in \mathbb{R}^{K^{text} \times d^{text}}, \tag{11}$$

where VisionDistiller and TextDistiller are based on the Multi-head Cross-Attention Mechanism. The $R^{image} \in \mathbb{R}^{K^{image} \times d^{image}}$ and $R^{text} \in \mathbb{R}^{K^{text} \times d^{text}}$ are learnable vectors (with $K^{text} < n^{text}, K^{image} < n^{image}$), which are the queries and can serve as semantic clustering centroids (Zhang & Yan, 2022) to help compress the information in $\tilde{X}^{image}$ and $\tilde{X}^{text}$. And $\text{X}^{image}$ and $\text{X}^{text}$ are the distilled image and text tokens.

### 3.1.3 MULTIMODAL ALIGNMENT

In multimodal time series forecasting, the time series modality occupies the dominant role and information from other modalities can serve as domain-specific knowledge to guide the extraction of temporal representations, thus enhancing the cross-domain generalization capability. In Aurora, we explicitly implement the above informative flow through a Modality-Guided Multi-head Self-Attention mechanism. First, we capture the correlations between the time series modality and others through Cross-Attention based VisionGuider and TextGuider:

$$\text{VAttn} = \text{VisionGuider}(X^{time}, \text{X}^{image}) \in \mathbb{R}^{n^{time} \times K^{image}}, \tag{12}$$

$$\text{TAttn} = \text{TextGuider}(X^{time}, \text{X}^{text}) \in \mathbb{R}^{n^{time} \times K^{text}}, \tag{13}$$

$$\text{Corr} = \text{VAttn} \cdot W \cdot \text{TAttn}^T \in \mathbb{R}^{n^{time} \times n^{time}}, \tag{14}$$

where VAttn and TAttn are unnormalized attention scores, separately denoting the correlations between time modality and image or text modality. Corr $\in \mathbb{R}^{n^{time} \times n^{time}}$ denotes the inherent correlations inside the time series modality, bridged through the text-image correlations. We also introduce

$W \in \mathbb{R}^{K^{image} \times K^{text}}$ as a learnable metric (Qiu et al., 2025e) to further tune the semantic distances. Therefore, this process helps bridge the correlations between time series tokens via a perspective of multimodal domain information. We then inject Corr into the temporal encoding process:

$$Q = X^{time} \cdot W^Q, \; K = X^{time} \cdot W^K, \; V = X^{time} \cdot W^V \tag{15}$$

$$S = (Q \cdot K^T + \text{Corr})/\sqrt{d^{time}}, O = \text{Softmax}(S) \cdot V, \tag{16}$$

$$O^{norm} = \text{LayerNorm}(X^{time} + O), \tag{17}$$

$$X^{time} = \text{LayerNorm}(\text{FeedForward}(O^{norm}) + O^{norm}), \tag{18}$$

where $W^Q, W^K, W^V \in \mathbb{R}^{d^{time} \times d^{time}}$. $X^{time} \in \mathbb{R}^{n^{time} \times d^{time}}$ denotes the generated temporal representations. The Corr matrix contains domain knowledge, which guides the attention scores to focus on the appropriate time series tokens (empirically validated in Section C.4). Finally, we fuse the representations from three modalities through a Cross-Attention based modality fuser:

$$\tilde{X}^{image} = \text{CrossAttn}(X^{time}, X^{image}) \in \mathbb{R}^{n^{time} \times d^{time}}, \tag{19}$$

$$\tilde{X}^{text} = \text{CrossAttn}(X^{time}, X^{text}) \in \mathbb{R}^{n^{time} \times d^{time}}, \tag{20}$$

$$X^{fuse} = X^{time} + \tilde{X}^{image} + \tilde{X}^{text}, \tag{21}$$

where $X^{fuse} \in \mathbb{R}^{n^{time} \times d^{time}}$ are the fused multimodal representations.

## 3.2 DECODING

### 3.2.1 CONDITION DECODING

Inspired by DiT (Peebles & Xie, 2023), we utilize an L-stacked Transformer to decode the conditions of future tokens, which helps construct the stable Flow Matching process. Specifically, the ConditionDecoder consists of a Causal-Transformer and a Cross-Transformer:

$$X^{cond} = \text{Causal-Transformer}(\text{Repeat}(X^{fuse}[-1], F)), \tag{22}$$

$$X^{cond} = \text{Cross-Transformer}(X^{cond}, X^{fuse}), \tag{23}$$

where $F$ denotes the number of future tokens. The last token of $X^{fuse}$ is first copied $F$ times and fed to the Causal-Transformer to generate the future conditions $X^{cond} \in \mathbb{R}^{F \times d^{time}}$, then we adopt a Cross-Transformer integrated with RoPE (Su et al., 2024) to further refine them into $X^{cond} \in \mathbb{R}^{F \times d^{time}}$, where $X^{fuse}$ is set as Key and Value embeddings. Therefore, the ConditionDecoder can efficiently output all $F$ conditions.

### 3.2.2 PROTOTYPE-GUIDED FLOW MATCHING

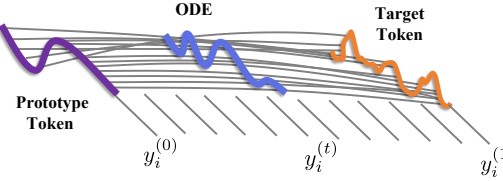

Figure 3: Prototype-Guided Flow Matching. The starting point is set as a prototype instead of a random gaussian noise, which provides an intuitive guidance in generation process.

**Algorithm 1** Prototype-Guided Flow Matching

1: Given condition $X_i^{cond}$, steps $J$, and Prototype $\tilde{\mathcal{P}}_i$.
2: Sample a noise $\epsilon_i \sim \mathcal{N}(\mathbf{0}, \mathbf{I})$.
3: $\Delta t = 1/J, h_i = X_i^{cond}, \hat{y}_i = \tilde{\mathcal{P}}_i + \epsilon_i$
4: **for** $j$ **in** $\{0, 1 \dots, J-1\}$ **do**
5: $\quad \hat{y}_i \leftarrow \hat{y}_i + v_{j\Delta t}^{\theta}(\hat{y}_i|h_i)\Delta t$
6: **end for**
7: **Return:** $\hat{y}_i$

Different from DDPM (Ho et al., 2020), which can be treated as an SDE solver to transform data from fixed Gaussian distributions to realistic target distributions, Flow Matching (Lipman et al., 2023) serves as a more intuitive and smooth ODE solver, which learns the Velocity Field between a random initial distribution and the target distribution. However, current methods (Liu et al., 2025e;

Kollovieh et al.) still set the initial distributions as Standard Gaussians, which neglects the capability of Flow Matching to work like a stochastic interpolant. Obviously, constructing appropriate prototype as the inital starting point can enhance the intuitiveness and stability of Flow Matching.

Based on the motivation that the future trends and periodicities of time series mainly rely on the multimodal domain knowledge in texts and images, we intuitively devise a Prototype Bank and a PrototypeRetriever to adaptively construct initial prototypes for Flow Matching. The Prototype Bank $\mathcal{P} \in \mathbb{R}^{M \times p^{time}}$ contains $M$ learnable period and trend prototypes, initialized through trigonometric, exponential, logarithmic, and polynomial bases. The Transformer-based PrototypeRetriever receives the text representations $\tilde{X}^{text}$ and image representations $\tilde{X}^{image}$ as inputs, considers the positional information of future tokens through Sinusoidal Embeddings (Vaswani et al., 2017), and outputs the categorical distributions of the all $M$ prototypes through Softmax:

$$\mathcal{D} = \text{PrototypeRetriever}(\tilde{X}^{text}, \tilde{X}^{image}) \in \mathbb{R}^{F \times M}, \tag{24}$$

where $\mathcal{D}$ denotes the weights of prototypes, we then generate the new prototypes through: $\tilde{\mathcal{P}} = \mathcal{D} \cdot \mathcal{P} \in \mathbb{R}^{F \times p^{time}}$, where the generated prototype $\tilde{\mathcal{P}}$ contains the approximate future periodicities and trends. As shown in Figure 3, the motivation of Flow Matching is to fit the velocity field between the initial prototype $y_i^{(0)} = \tilde{\mathcal{P}}_i + \epsilon_i$ and the target horizon $y_i^{(1)} = y_i$, where $y_i \in \mathbb{R}^{p^{time}}$ is the groundtruth of the $i$-th future token, and $\epsilon_i \sim \mathcal{N}(\mathbf{0}, \mathbf{I})$ is used to increase the diversity during training. We design the Flow-Matching Network with an MLP structure and utilize the AdaLN (Peebles & Xie, 2023) to integrate the multimodal conditions $h_i = X_i^{cond}$. We adopt the conditional optimal-transport path, which is energy-optimal and contributes to a uniform velocity field. And the function of Flow-Matching Network $v_t^\theta$ is to predict the velocity based on the current position $y_i^{(t)}$ and condition $h_i$. To achieve this, the token-wise optimization objective $\mathcal{L}$ is designed as:

$$\mathcal{L}(\theta, h_i) = \mathbb{E}_{t, y_i^{(0)}, y_i^{(1)}} \| v_t^\theta(y_i^{(t)} | h_i) - (y_i^{(1)} - y_i^{(0)}) \|^2, \tag{25}$$

where $t \in [0, 1]$, $y_i^{(1)} - y_i^{(0)}$ denotes the targeted fixed velocity field. $y_i^{(t)} = t y_i^{(1)} + (1 - t) y_i^{(0)}$ is the expected position in the uniform velocity field at moment $t$. The objective is to tutor the Flow-Matching Network $v_t^\theta$ to output the velocity when given the position and condition.

In the inference phase, the sampling process is a discretized integration process–see Algorithm 1. The Gaussian noise $\epsilon_i \sim \mathcal{N}(\mathbf{0}, \mathbf{I})$ helps support probabilistic forecasting. Finally, we can obtain the forecasts $\hat{y}_i \in \mathbb{R}^{p^{time}}$ of the $i$-th future token. And the forecasts of the future horizon are $\hat{Y} = \text{Concat}\{\hat{y}_i\} \in \mathbb{R}^{F \times p^{time}}$.

## 4 EXPERIMENTS

We make extensive experiments to evaluate the performance of Aurora. Specifically, we introduce the experimental settings in Section 4.1. In Section 4.2, we evaluate the zero-shot and few-shot performance of Aurora on multimodal forecasting scenarios. Considering the modal absence in the realistic world, we also evaluate the zero-shot performance of Aurora on unimodal forecasting scenarios–see Sections 4.3, 4.4. To analyze the key components in Aurora,

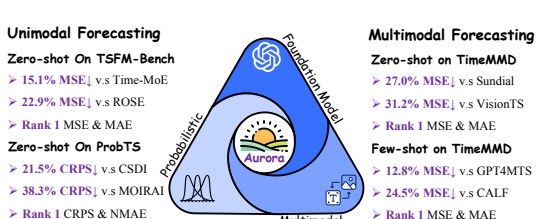

Figure 4: Evaluation summary of Aurora.

we also make detailed model analyses in Section 4.5. In summary–see Figure 4, our proposed Aurora achieves state-of-the-art performance in both unimodal and multimodal forecasting scenarios.

### 4.1 EXPERIMENTAL SETTINGS

**Cross-Domain Multimodal Time Series Corpus**. We first collect a substantial number of open-source time series datasets across diverse domains, then generate the corresponding sample-wise textual descriptions using Large Language Model (Liu et al., 2024a), which simulates the downstream scenarios with domain-specific textual information.

**Benchmarks**. We evaluate both the multimodal forecasting and unimodal forecasting performance of Aurora on 5 benchmarks, including TimeMMD (Liu et al., 2024b), TSFM-Bench (Li et al., 2025c), ProbTS (Zhang et al., 2024), TFB (Qiu et al., 2024), and EPF (Wang et al., 2024b). Note that these benchmarking datasets are strictly excluded from the pretraining time series corpus.

**Baselines**. We compare Aurora with 11 well-known unimodal time series foundation models, including Sundial (Liu et al., 2025e), VisionTS (Chen et al., 2024a), ROSE (Wang et al., 2025f), Time-MoE (Shi et al., 2024b), MOIRAI (Woo et al., 2024), TTM (Ekambaram et al., 2024), TimesFM (Das et al., 2024), Timer (Liu et al., 2024e), UniTS (Gao et al., 2024), Chronos (Ansari et al.), and Lag-Llama (Rasul et al., 2023). We also consider multiple strong end-to-end supervised models, including multimodal ones like GPT4MTS (Jia et al., 2024), TATS (Li et al., 2025d), CALF (Liu et al., 2025b), and Time-VLM (Zhong et al.), and unimodal ones like TimeXer (Wang et al., 2024b), PatchTST (Nie et al., 2023), iTransformer (Liu et al., 2024d) TSDiff (Kollovieh et al., 2023), CSDI (Tashiro et al., 2021), TimeGrad (Rasul et al., 2021a), and GRU MAF (Rasul et al., 2021b). The detailed information is provided in Appendix A.

## 4.2 MULTIMODAL FORECASTING

We compare the zero-shot forecasting performance of Aurora with unimodal Foundation Models, and compare the few-shot (10%) forecasting performance with Full-shot Multimodal End-to-end Supervised Models. As shown in Table 1, compared with unimodal foundation models, Aurora obviously possesses stronger generalization capability by achieving most 1st counts. Compared with previous state-of-the-arts Sundial and VisionTS, Aurora achieves average MSE reduction of *27.0%* and *31.2%* on TimeMMD. When compared with Full-shot Multimodal End-to-end Supervised Models, Aurora is trained on only 10% of data and outperforms all baselines in most settings. Compared with well-known baselines like GPT4MTS and CALF, Aurora achieves average MSE reduction of *12.8%* and *24.5%*. On some datasets such as Climate and Environment, even the zero-shot performance of Aurora has outperformed those full-shot baselines. These empirical evidences can provide strong support of Aurora's the multimodal generalization capability.

Table 1: Average results of multimodal zero-shot & few-shot forecasting experiments on datasets from TimeMMD. Lower MSE or MAE values indicate better predictions. **Red**: the best, Blue: the 2nd best. All the results are listed in Table 12 of Appendix B.

| Type | ⃠ Zero-shot Foundation Models | | | | | | | | | | 10% few-shot | | ⏱ Full-shot Multimodal End-to-end Supervised Models | | | | | | | |
|---|---|---|---|---|---|---|---|---|---|---|---|---|---|---|---|---|---|---|---|---|
| Models | Aurora (Ours) | | Sundial (2025) | | VisionTS (2025) | | ROSE (2025) | | MOIRAI (2024) | | Aurora (Ours) | | GPT4MTS (2025) | | TATS (2025) | | CALF (2025) | | Time-VLM (2025) | |
| Metrics | MSE | MAE | MSE | MAE | MSE | MAE | MSE | MAE | MSE | MAE | MSE | MAE | MSE | MAE | MSE | MAE | MSE | MAE | MSE | MAE |
| Agriculture | **0.272** | 0.348 | 0.373 | 0.392 | 0.290 | **0.336** | 0.345 | 0.372 | 0.272 | 0.403 | **0.212** | **0.293** | 0.225 | 0.298 | 0.215 | 0.301 | 0.250 | 0.315 | 0.237 | 0.302 |
| Climate | **0.865** | **0.749** | 1.154 | 0.881 | 1.307 | 0.930 | 1.475 | 0.987 | 1.921 | 1.095 | 0.862 | 0.746 | 1.182 | 0.889 | 1.180 | 0.887 | 1.286 | 0.922 | 1.195 | 0.899 |
| Economy | **0.033** | **0.146** | 0.291 | 0.432 | 0.301 | 0.442 | 0.289 | 0.433 | 0.405 | 0.512 | **0.016** | **0.099** | 0.017 | 0.103 | 0.017 | 0.104 | 0.163 | 0.307 | 0.024 | 0.125 |
| Energy | **0.255** | 0.370 | 0.272 | 0.367 | 0.304 | 0.420 | 0.386 | 0.479 | 0.324 | 0.417 | **0.230** | **0.329** | 0.262 | 0.380 | 0.255 | 0.368 | 0.244 | 0.365 | 0.260 | 0.374 |
| Environment | **0.276** | **0.379** | 0.336 | 0.416 | 0.354 | 0.436 | 0.392 | 0.456 | 0.351 | 0.403 | **0.265** | **0.372** | 0.323 | 0.400 | 0.319 | 0.396 | 0.325 | 0.387 | 0.319 | 0.397 |
| Health | **1.553** | **0.850** | 1.970 | 0.992 | 2.436 | 1.221 | 2.598 | 1.201 | 2.736 | 1.241 | **1.343** | 0.776 | 1.464 | 0.799 | 1.356 | 0.767 | 1.491 | 0.775 | 1.489 | 0.834 |
| Security | 72.475 | 4.084 | 70.441 | 4.005 | 79.598 | 4.597 | 84.324 | 4.765 | 93.245 | 5.173 | **70.062** | **3.988** | 71.487 | 4.068 | 72.406 | 4.097 | 76.376 | 4.300 | 73.731 | 4.181 |
| Social Good | **0.838** | **0.516** | 1.036 | 0.573 | 1.126 | 0.618 | 1.141 | 0.581 | 1.430 | 0.651 | 0.814 | 0.494 | 0.920 | 0.450 | 0.918 | 0.428 | 0.906 | 0.401 | 0.868 | 0.444 |
| Traffic | **0.161** | **0.289** | 0.271 | 0.405 | 0.281 | 0.407 | 0.341 | 0.451 | 0.406 | 0.468 | **0.157** | 0.290 | 0.203 | 0.261 | 0.179 | 0.238 | 0.222 | 0.293 | 0.216 | 0.319 |
| **1st Count** | **31** | **26** | 4 | 7 | 0 | 4 | 0 | 0 | 1 | 0 | 30 | 23 | 1 | 1 | 4 | 4 | 1 | 8 | 0 | 0 |

## 4.3 UNIMODAL FORECASTING

Considering the modality absence phenomenon in many downstream scenarios, Aurora also supports forecasting without textual inputs through random masking in the pretraining phase. And endogenous images can be always obtained from raw time series. To evaluate the unimodal zero-shot forecasting performance, we conduct experiments on TSFM-Bench and ProbTS. As shown in Table 2–3, Aurora achieves state-of-the-art performance on both deterministic and probabilistic forecasting tasks. Compared with Time-MoE and ROSE, Aurora achieves average MSE reduction of *15.1%* and *22.9%* on TSFM-Bench, demonstrating strong deterministic forecasting capability. When evaluated on probabilistic forecasting benchmark ProbTS, Aurora also outperforms CSDI and MOIRAI with average CRPS reduction of *21.5%* and *38.3%*. Aurora is proven to be the best unimodal time series foundation model, ensuring the robustness when modality absence occurs.

Table 2: Average results of unimodal zero-shot deterministic forecasting experiments on datasets from TSFM-Bench. Lower MSE or MAE values indicate better predictions. ('-') denotes datasets included in the model's pretraining and therefore excluded from testing. **Red**: the best, Blue: the 2nd best. All the results are listed in Table 13 of Appendix B.

| Type | 🚫 Zero-shot Foundation Models | | | | | | | | | | | | | | | | | | | |
|---|---|---|---|---|---|---|---|---|---|---|---|---|---|---|---|---|---|---|---|---|
| Models | Aurora (Ours) | | Sundial (2025) | | ROSE (2025) | | Timer (2024) | | TimesFM (2023) | | Chronos (2024) | | Time-MoE (2024) | | UniTS (2024) | | MOIRAI (2024) | | TTM (2024) | |
| Metrics | MSE | MAE | MSE | MAE | MSE | MAE | MSE | MAE | MSE | MAE | MSE | MAE | MSE | MAE | MSE | MAE | MSE | MAE | MSE | MAE |
| ETT (Avg) | 0.331 | 0.376 | 0.335 | 0.379 | 0.393 | 0.411 | 0.551 | 0.478 | 0.415 | 0.406 | 0.442 | 0.408 | 0.357 | 0.390 | 0.471 | 0.437 | 0.382 | 0.388 | 0.441 | 0.430 |
| Weather | 0.230 | 0.267 | 0.234 | 0.270 | 0.265 | 0.305 | 0.292 | 0.313 | - | - | 0.288 | 0.309 | 0.256 | 0.289 | 0.275 | 0.298 | 0.260 | 0.275 | 0.265 | 0.307 |
| Electricity | 0.178 | 0.275 | 0.169 | 0.265 | 0.234 | 0.320 | 0.297 | 0.375 | - | - | - | - | - | - | 0.198 | 0.291 | 0.188 | 0.273 | 0.222 | 0.317 |
| Traffic | 0.524 | 0.352 | - | - | 0.588 | 0.412 | 0.613 | 0.407 | - | - | 0.615 | 0.421 | - | - | - | - | - | - | 0.564 | 0.386 |
| Solar | 0.203 | 0.289 | 0.221 | 0.252 | 0.505 | 0.549 | 0.771 | 0.604 | 0.500 | 0.397 | 0.393 | 0.319 | 0.411 | 0.428 | 0.845 | 0.669 | 0.714 | 0.704 | 0.815 | 0.710 |
| PEMS08 | 0.563 | 0.552 | - | - | 1.369 | 0.979 | 0.866 | 0.695 | 1.485 | 0.907 | 1.707 | 1.024 | - | - | 1.253 | 0.879 | - | - | 1.730 | 1.066 |
| Wind | 1.151 | 0.763 | 1.186 | 0.772 | 1.251 | 0.820 | 1.201 | 0.783 | 1.613 | 0.870 | 1.478 | 0.834 | - | - | 1.425 | 0.848 | 1.299 | 0.795 | 1.337 | 0.829 |
| NYSE | 0.528 | 0.526 | 0.880 | 0.642 | - | - | 0.988 | 0.704 | 0.623 | 0.536 | 1.129 | 0.720 | - | - | 1.220 | 0.820 | - | - | - | - |
| 1st Count | 27 | 21 | 11 | 13 | 3 | 1 | 0 | 0 | 1 | 2 | 0 | 0 | 2 | 2 | 0 | 0 | 0 | 5 | 0 | 0 |

Table 3: Average results of unimodal zero-shot probabilistic forecasting experiments on datasets from ProbTS. Lower MSE or MAE values indicate better predictions. ('-') denotes datasets included in the model's pretraining and therefore excluded from testing. ('/') denotes it takes too long time to run. **Red**: the best, Blue: the 2nd best. All the results are listed in Table 14 of Appendix B.

| Type | 🚫 Zero-shot Foundation Models | | | | | | | | | | 🕐 Full-shot Probabilistic End-to-end Supversied Models | | | | | | | |
|---|---|---|---|---|---|---|---|---|---|---|---|---|---|---|---|---|---|---|
| Models | Aurora (Ours) | | Sundial (2025) | | Chronos (2024) | | MOIRAI (2024) | | Lag-Llama (2023) | | TSDiff (2023) | | CSDI (2022) | | TimeGrad (2022) | | GRU MAF (2021) | |
| Metrics | CRPS | NMAE | CRPS | NMAE | CRPS | NMAE | CRPS | NMAE | CRPS | NMAE | CRPS | NMAE | CRPS | NMAE | CRPS | NMAE | CRPS | NMAE |
| ETT (Avg) | 0.231 | 0.257 | 0.231 | 0.273 | 0.290 | 0.316 | 0.366 | 0.377 | 0.273 | 0.310 | 0.370 | 0.465 | 0.304 | 0.389 | 0.493 | 0.619 | 0.388 | 0.475 |
| Weather | 0.070 | 0.076 | 0.087 | 0.102 | 0.142 | 0.158 | 0.179 | 0.143 | 0.096 | 0.072 | 0.132 | 0.134 | 0.077 | 0.093 | 0.125 | 0.155 | 0.133 | 0.165 |
| Electricity | 0.085 | 0.103 | 0.081 | 0.098 | - | - | 0.247 | 0.290 | - | - | 0.407 | 0.519 | / | / | 0.102 | 0.126 | 0.094 | 0.122 |
| Traffic | 0.220 | 0.262 | - | - | 0.269 | 0.295 | - | - | 0.330 | 0.385 | 0.327 | 0.392 | / | / | 0.225 | 0.264 | / | / |
| Exchange | 0.044 | 0.047 | 0.045 | 0.049 | 0.044 | 0.047 | 0.045 | 0.050 | 0.057 | 0.069 | 0.084 | 0.111 | 0.069 | 0.086 | 0.082 | 0.095 | 0.070 | 0.083 |
| ILI | 0.147 | 0.166 | 0.148 | 0.166 | 0.170 | 0.197 | 0.159 | 0.197 | 0.156 | 0.211 | 0.248 | 0.259 | 0.276 | 0.290 | 0.284 | 0.310 | 0.262 | 0.288 |
| 1st Count | 19 | 24 | 8 | 8 | 1 | 1 | 2 | 1 | 0 | 0 | 0 | 0 | 4 | 1 | 1 | 1 | 0 | 0 |

## 4.4 SHORT-TERM FORECASTING

We also consider the scenarios with insufficient historical time series data, for which foundation models are better suited than end-to-end models. To evaluate Aurora in such scenarios, we simulate with short-term forecasting, where the contextual series is very short and limited for training. Specifically, we conduct experiments on EPF (Wang et al., 2024b) and univariate datasets from TFB (Qiu et al., 2024). As shown in Table 4, Aurora outperforms most-advanced Foundation Models such as Sundial and VisionTS in most evaluations. Compared with full-shot supervised models like TimeXer, and iTransformer, Aurora also achieves competitive performance with them. Focusing on more scenarios, i.e., the 8,068 univariate datasets in TFB–see Figure 5, we report the mean MASE and msMAPE results, which indicate that Aurora also achieves state-of-the-art performance against zero-shot Foundation Models, and full-shot supervised models with versatile neural structures. All of the experiments demonstrate Aurora's strong capability in short-term forecasting scenarios.

Table 4: Results of short-term zero-shot forecasting experiments on datasets from EPF. Lower MSE or MAE values indicate better predictions. ('-') denotes datasets included in the model's pretraining and therefore excluded from testing. **Red**: the best, Blue: the 2nd best.

| Type | 🚫 Zero-shot Foundation Models | | | | | | | | | | 🕐 Full-shot End-to-end Supervised Models | | | | | | | |
|---|---|---|---|---|---|---|---|---|---|---|---|---|---|---|---|---|---|---|
| Models | Aurora (Ours) | | Sundial (2025) | | VisionTS (2025) | | ROSE (2025) | | MOIRAI (2024) | | TimeXer (2024) | | iTransformer (2024) | | PatchTST (2023) | | TimesNet (2023) | |
| Metrics | MSE | MAE | MSE | MAE | MSE | MAE | MSE | MAE | MSE | MAE | MSE | MAE | MSE | MAE | MSE | MAE | MSE | MAE |
| NP | 0.288 | 0.312 | 0.256 | 0.277 | 0.510 | 0.461 | 0.666 | 0.536 | 0.660 | 0.538 | 0.238 | 0.268 | 0.265 | 0.300 | 0.267 | 0.284 | 0.250 | 0.289 |
| PJM | 0.084 | 0.183 | 0.088 | 0.189 | 0.251 | 0.366 | 0.311 | 0.402 | 0.330 | 0.423 | 0.088 | 0.188 | 0.097 | 0.197 | 0.106 | 0.209 | 0.097 | 0.195 |
| BE | 0.361 | 0.257 | 0.371 | 0.270 | 0.679 | 0.457 | 0.815 | 0.514 | 0.837 | 0.534 | 0.374 | 0.241 | 0.394 | 0.270 | 0.403 | 0.264 | 0.419 | 0.288 |
| FR | 0.387 | 0.206 | 0.392 | 0.207 | 0.625 | 0.393 | 0.746 | 0.447 | 0.751 | 0.454 | 0.381 | 0.211 | 0.439 | 0.233 | 0.411 | 0.220 | 0.431 | 0.234 |
| DE | 0.539 | 0.475 | 0.541 | 0.484 | 0.961 | 0.687 | 1.276 | 0.778 | 1.251 | 0.779 | 0.440 | 0.418 | 0.479 | 0.433 | 0.461 | 0.432 | 0.502 | 0.446 |

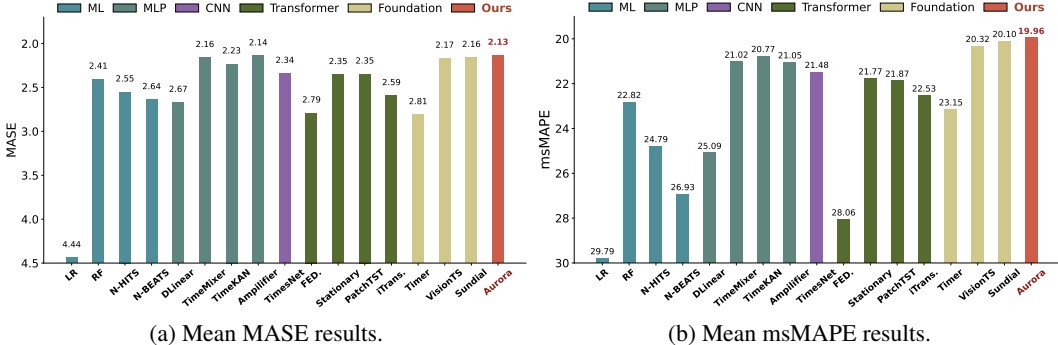

(a) Mean MASE results.        (b) Mean msMAPE results.

Figure 5: Mean MASE and msMAPE results of 8,068 univariate datasets in TFB. The full results can be found in Table 10 and 11 of Appendix B. **Red**: the best.

## 4.5 Model Analysis

**Ablation Studies.** Based on the Modality-Guided Multi-head Self-Attention, Aurora can utilize the domain knowledge contained in text and image modalities to model the temporal features. To validate its effectiveness, we make ablations on it by setting Variant 1, which adopts original Multi-head Self-Attention. Considering the Prototype-Guided Flow Matching, which can generate prototypes of future tokens to simplify the generation process, we make Variant 2, which does not utilize the prototype mechanism and sets the initial distribution as Standard Gaussian. Naturally, we also make Variant 3, which eliminates both of them. As shown in Table 5, results show

Table 5: Ablation studies on without Modality-Guided Multi-head Self-Attention (Variant 1), without Prototype-Guided Flow Matching (Variant 2), and without both of them (Variant 3).

| Models | Aurora | | Variant 1 | | Variant 2 | | Variant 3 | |
|---|---|---|---|---|---|---|---|---|
| Metrics | MSE | MAE | MSE | MAE | MSE | MAE | MSE | MAE |
| Agriculture | **0.272** | **0.348** | 0.298 | 0.351 | 0.290 | 0.334 | 0.324 | 0.366 |
| Climate | **0.865** | **0.749** | 1.176 | 0.868 | 1.008 | 0.836 | 1.447 | 0.962 |
| Economy | **0.033** | **0.146** | 0.277 | 0.419 | 0.045 | 0.172 | 0.296 | 0.440 |
| Energy | **0.255** | **0.370** | 0.268 | 0.383 | 0.257 | 0.372 | 0.272 | 0.388 |
| Environment | **0.276** | **0.379** | 0.324 | 0.398 | 0.354 | 0.411 | 0.388 | 0.459 |
| Health | **1.553** | **0.850** | 1.757 | 0.936 | 1.588 | 0.876 | 2.047 | 1.174 |
| Security | **72.475** | **4.084** | 81.982 | 4.571 | 79.825 | 4.482 | 84.295 | 4.881 |
| Social Good | **0.838** | **0.516** | 1.012 | 0.548 | 1.425 | 0.648 | 1.487 | 0.663 |
| Traffic | **0.161** | **0.289** | 0.244 | 0.378 | 0.273 | 0.418 | 0.335 | 0.467 |

that each above-mentioned module is indispensable, and a cascading effect occurs when both modules are removed, where the performance crashes when the modules are removed.

**Inference Scability.** Adopting a generative probabilistic head, Aurora makes forecasts based on multiple sampling–see Algorithm 1. So that we study the scability of Prototype-Guided Flow Matching by exploring the correlations between the sampling number and forecasting performance in Figure 6. Specifically, experiments are conduct on ProbTS, where the average values of CRPS and NMAE across all datasets are reported. The results indicate that both CRPS and NMAE demonstrate a consis-

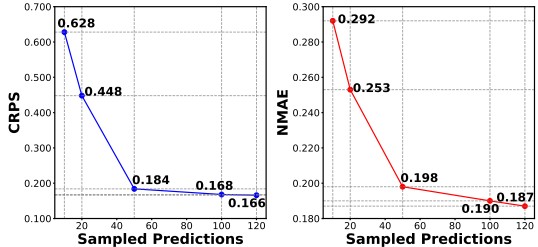

Figure 6: Sampled Predictions.

tent improvement as the sampling number rises. They attain good performance when the sampling number reaches *100*, showing obvious inference scability, and moderate efficiency–see Section A.7.

## 5 Conclusion

In this work, we introduce Aurora, the *first* multimodal time series foundation model. To sum up, Aurora adopts a merticulously designed Modality-Guided Self-Attention to capture temporal dynamics, and a novel Prototype-Guided Flow Matching to enhance forecasting performance and supporting generative probabilistic modeling. Comprehensive experiments on unimodal and multimodal forecasting tasks, including 5 well-recognized benchmarks, demonstrate that Aurora is a strong *out-of-the-box* tool for decision intelligence.

## ACKNOWLEDGEMENTS

This work was partially supported by the National Natural Science Foundation of China (62372179, 62472174), and ECNU Multifunctional Platform for Innovation (001). Chenjuan Guo is the corresponding author of the work.

## ETHICS STATEMENT

Our work exclusively uses publicly available benchmark datasets that contain no personally identifiable information. The Cross-Domain Multimodal Time Series Corpus used to pretrain Aurora is also collected from public datasets, and integrated with LLM-generated textual descriptions, also containing no personally identifiable information. No human subjects are involved in this research.

## REPRODUCIBILITY STATEMENT

The performance of Aurora and datasets used in our work are real, and all experimental results can be reproduced. We have released our model code and checkpoints on Github and Huggingface.

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

## THE USE OF LARGE LANGUAGE MODELS (LLMs)

In this work, we only adopt Large Language Models in our methodology and data generation. Specifically, we leverage Bert as the TextEncoder of Aurora to extract the textual features. To generate the textual descriptions for the cross-domain multimodal time series corpus, we provide domain descriptions and raw time series for GPT-4, encouraging it generate the descriptions of data characteristics, which are only used for pretraining Aurora. Note that we do not use Large Language Models in writing.

## A  EXPERIMENTAL DETAILS

### A.1  CROSS-DOMAIN MULTIMODAL TIME SERIES CORPUS

**Cross-Domain Time Series Corpus.** To pretrain Aurora, we initially make use of an extensive compilation of time series datasets. These datasets are sourced from multiple origins, encompassing specific subsets from repositories such as ERA5 (Liu et al., 2025e), IoT (Liu et al., 2025e), Monash (Godahewa et al., 2021), UEA (Bagnall et al., 2018), and UCR (Dau et al., 2019), as well as several well-established benchmarks (Zhang et al., 2017; Wang et al., 2024a; Liu et al., 2022; McCracken & Ng, 2016; Taieb et al., 2012). A comprehensive list of these datasets is presented in Figure 7, containing more than 1 billion time series points. We take care to ensure that there is no overlap between the pre-training datasets and those employed in downstream evaluations. It should be noted that while both the pre-training and target sets incorporate weather, Energy, Health, and Economy data, they are from different sources.

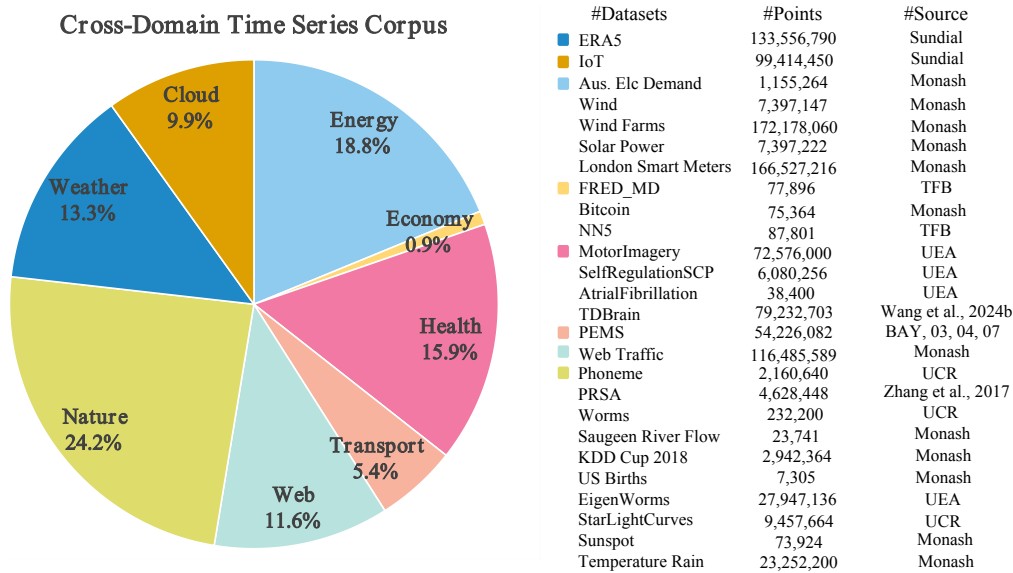

Figure 7: Introduction of time series data sources in Cross-domain Multimodal Time Series Corpus. We list the ratios of different domains and report detailed dataset sources and lengths.

**Textual Descriptions.** Since multimodal time series data is scarce in the real world, current works (Liu et al., 2024b; Kong et al., 2025; Xie et al., 2024) often construct corresponding textual information based on human experiences and Large Language Models, which is proven effective in training models. Following their paradigm, we provide raw time series data in Figure 7 with domain descriptions, encouraging GPT-4 (Achiam et al., 2023) to heuristically generate textual descriptions of sample-wise time series, thus obtaining high-quality multimodal time-series data from simulation. Specifically, after a GPT-4 agent generates the textual descriptions, we first coarsely check the quality with another GPT-4 agent. If the quality is low, the process will be reset. After a batch of textual descriptions are generated, we randomly sample from them and check the quality manually,

then determine whether to regenerate this batch of data and tune the prompts. As shown in Figure 8, here's some samples of our prompts for GTP4 and the generated texts:

**Prompt:** This time series segment records a span of 2000 seconds, reflecting the road flow of some Interstate Highway. Here's an Accident report about this time series, 📄 you can refer to it. Please analyze the future trend and characteristics of this time series, and output textual descriptions with no more than 200 words.

**Generated Text:** The width of the road in the Interstate Highway Accident Report was carrying a low traffic volume, with almost 70 percent of the road being affected. Corresponding time series has experienced drastic fluctuations. It started with a high value of 22.7, then dropped sharply to a low value of 2.3, and after that, it further decreased to a low value of 0.1. Overall, this time series shows a sudden change, including a significant decline and a continuous downward trend.

(a) Case 1

**Prompt:** This time series segment records a span of 100 days, reflecting the daily stock prices of a well-known tech company. Here's an economic analysis report 📄 about this time series, you can refer to it. Please analyze the future trend and characteristics of this time series, and output textual descriptions with no more than 200 words.

**Generated Text:** According to the economic analysis report, the tech company faced strong market competition recently. The time series of its stock prices has been volatile. It opened at $150, rose to a peak of $180, then plunged to $120. Currently, it's hovering around $130. Considering the industry situation, the future trend might show a slow upward recovery as the company launches new products, but short-term fluctuations are still likely due to market sentiment.

(b) Case 2

**Prompt:** This time series segment records a span of 15 minutes, reflecting the real-time water level of a river near a city. Here's a flood-warning report 📄 about this time series, you can refer to it. Please analyze the future trend and characteristics of this time series, and output textual descriptions with no more than 200 words.

**Generated Text:** The flood-warning report shows that there has been heavy rainfall upstream. The water level time series started at 2 meters, then quickly rose to 3.5 meters within 5 minutes. Currently, it stands at 4 meters. Given the continuous rainfall, the water level is likely to keep rising in the short term. However, if the rain stops, the water level may start to decline gradually as the river discharges water downstream. But until the rainfall situation stabilizes, the time series will remain highly volatile.

(c) Case 3

**Prompt:** This time series segment records a span of 36 months, reflecting the monthly electricity consumption of a large industrial park. Here's an energy consumption assessment about this time series, you can refer to it. 📄 Please analyze the future trend and characteristics of this time series, and output textual descriptions with no more than 200 words.

**Generated Text:** The energy consumption assessment indicates that the industrial park has been gradually implementing energy-saving measures. The time series of electricity consumption initially showed an upward trend, reaching a high of 500,000 kWh per month. Then, after the implementation of energy - saving projects, it started to decline steadily. It's now at 350,000 kWh per month. In the future, with the continuous improvement of energy-saving technologies and more efficient production processes, the electricity consumption is expected to continue to decrease gradually.

(d) Case 4

Figure 8: Some cases of prompt-driven textual descriptions.

## A.2 BASELINES

For zero-shot forecasting, we compare Aurora against 11 advanced foundation models: Sundial (Liu et al., 2025e), VisionTS (Chen et al., 2024a), ROSE (Wang et al., 2025f), Timer (Liu et al., 2024e), MOIRAI (Woo et al., 2024), TTM (Olivares et al., 2023), Chronos (Ansari et al.), TimesFM (Das et al., 2024), Time-MoE (Shi et al., 2024b), UniTS (Gao et al., 2024), and Lag-Llama (Rasul et al., 2023). We also compare Aurora with total multiple End-to-end supervised models: TimeXer (Wang et al., 2024b), PatchTST (Nie et al., 2023), iTransformer (Liu et al., 2024d), TimesNet (Wu et al., 2023), GPT4MTS (Jia et al., 2024), TATS (Li et al., 2025d), CALF (Liu et al., 2025b), Time-VLM (Zhong et al.), TSDiff (Kollovieh et al., 2023), TimeGrad (Rasul et al., 2021a), CSDI (Tashiro et al., 2021), and GRU MAF (Rasul et al., 2021b). The corresponding codebases and implementation details are summarized in Table 6.

Table 6: Code repositories for baselines.

| Model Types | Models | Code Repositories |
|---|---|---|
| End-to-end | TSDiff | https://github.com/amazon-science/unconditional-time-series-diffusion |
| | CSDI | https://github.com/ermongroup/CSDI |
| | TimeGrad | https://github.com/Zjh152/TimeGrad |
| | GRU MAF | https://github.com/microsoft/ProbTS |
| | GPT4MTS | https://github.com/Flora-jia-jfr/GPT4MTS-Prompt-based-Large-Language-Model-for-Multimodal-Time-series-Forecasting |
| | TATS | https://github.com/iDEA-iSAIL-Lab-UIUC/TaTS |
| | CALF | https://github.com/Hank0626/CALF |
| | Time-VLM | https://github.com/CityMind-Lab/ICML25-TimeVLM |
| | PatchTST | https://github.com/yuqinie98/PatchTST |
| | iTransformer | https://github.com/thuml/iTransformer |
| | TimeXer | https://github.com/thuml/TimeXer |
| | TimesNet | https://github.com/thuml/TimesNet |
| Foundation | Sundial | https://github.com/thuml/Sundial |
| | VisionTS | https://github.com/Keytoyze/VisionTS |
| | ROSE | https://github.com/decisionintelligence/TSFM-Bench |
| | Timer | https://github.com/thuml/Large-Time-Series-Model |
| | MOIRAI | https://github.com/redoules/moirai |
| | Chronos | https://github.com/amazon-science/chronos-forecasting |
| | TimesFM | https://github.com/google-research/timesfm |
| | Time-MoE | https://github.com/Time-MoE/Time-MoE |
| | Lag-Llama | https://github.com/time-series-foundation-models/lag-llama |
| | UniTS | https://github.com/mims-harvard/UniTS |
| | TTM | https://huggingface.co/ibm-granite/granite-timeseries-ttm-r1 |

## A.3 BENCHMARKS

To thoroughly assess the effectiveness of Aurora, we conduct comprehensive experiments on TimeMMD (Liu et al., 2024b), TSFM-Bench (Li et al., 2025c), ProbTS (Zhang et al., 2024), TFB (Qiu et al., 2024), and EPF (Olivares et al., 2023).

For multimodal forecasting, we use Agriculture, Climate, Economy, Energy, Environment, Health, Security, Social Good, and Traffic. For most datasets, the prediction length is set to $L \in \{6, 8, 10, 12\}$, while Energy and Health use $L \in \{12, 24, 36, 48\}$, and Environment uses $L \in \{48, 96, 192, 336\}$.

For unimodal forecasting, we adopt ETTm1, ETTm2, ETTh1, ETTh2, Weather, Electricity, Traffic, Exchange, PEMS08, Solar, and Wind from ProbTS and TSFM-Bench. The prediction length is set to $L \in \{96, 192, 336, 720\}$, and the specific evaluation settings are different in ProbTS and TSFM-Bench.

For more short-term forecasting scenarios, we adopt datasets from EPF and TFB, the prediction lengths are set as the default settings in these benchmarks.

All models are configured with the contextual length that yields the best performance as recommended in their respective papers. It is crucial to note that, for datasets such as ETTh1 and Traffic,

which are shared between TSFM-Bench and ProbTS, the evaluation settings, particularly strides, differ. A summary of the dataset statistics can be found in Table 7.

Table 7: Statistics of benchmark datasets.

| Dataset | Domain | Frequency | Length/Num | Dim | Split | Stride | Benchmark | Description |
|---|---|---|---|---|---|---|---|---|
| Agriculture | Retail Broiler Composite | Monthly | 496 | 1 | 7:1:2 | 1 | TimeMMD | The record of Retail Broiler Composite between 1983 - Present |
| Climate | Drought Level | Monthly | 496 | 5 | 7:1:2 | 1 | TimeMMD | The record of Drought Level between 1983 - Present |
| Economy | International Trade Balance | Monthly | 423 | 3 | 7:1:2 | 1 | TimeMMD | The record of International Trade between 1989 - Present |
| Energy | Gasoline Prices | Weekly | 1,479 | 9 | 7:1:2 | 1 | TimeMMD | The prices of Gasoline between 1996 - Present |
| Environment | Air Quaility Index | Daily | 11,102 | 4 | 7:1:2 | 1 | TimeMMD | The indices of Air Quality between 1982 - 2023 |
| Health | Influenza Patients Proportion | Weekly | 1,389 | 11 | 7:1:2 | 1 | TimeMMD | The record of Influenza Patients Proportion between 1997 - Present |
| Security | Disaster and Emergency Grants | Monthly | 297 | 1 | 7:1:2 | 1 | TimeMMD | The record of Disaster and Emergency Grants between 1999 - Present |
| Social Good | Unemployment Rate | Monthly | 900 | 1 | 7:1:2 | 1 | TimeMMD | The Unemployment Rate between 1950 - Present |
| Traffic | Travel Volume | Monthly | 531 | 1 | 7:1:2 | 1 | TimeMMD | The Travel Volume between 1980 - Present |
| ETTm1 | Electricity | 15 mins | 57,600 | 7 | 6:2:2 | 1 | TSFM-Bench | Power transformer 1, comprising seven indicators such as oil temperature and useful load |
| ETTm2 | Electricity | 15 mins | 57,600 | 7 | 6:2:2 | 1 | TSFM-Bench | Power transformer 2, comprising seven indicators such as oil temperature and useful load |
| ETTh1 | Electricity | 1 hour | 14,400 | 7 | 6:2:2 | 1 | TSFM-Bench | Power transformer 1, comprising seven indicators such as oil temperature and useful load |
| ETTh2 | Electricity | 1 hour | 14,400 | 7 | 6:2:2 | 1 | TSFM-Bench | Power transformer 2, comprising seven indicators such as oil temperature and useful load |
| Weather | Environment | 10 mins | 52,696 | 21 | 7:1:2 | 1 | TSFM-Bench | Recorded every for the whole year 2020, which contains 21 meteorological indicators |
| Electricity | Electricity | 1 hour | 26,304 | 321 | 7:1:2 | 1 | TSFM-Bench | Electricity records the electricity consumption in kWh every 1 hour from 2012 to 2014 |
| Traffic | Traffic | 1 hour | 17,544 | 862 | 7:1:2 | 1 | TSFM-Bench | Road occupancy rates measured by 862 sensors on San Francisco Bay area freeways |
| Solar | Energy | 10 mins | 52,560 | 137 | 7:1:2 | 1 | TSFM-Bench | Solar production records collected from 137 PV plants in Alabama |
| PEMS08 | Traffic | 5 mins | 17,856 | 170 | 6:2:2 | 1 | TSFM-Bench | Traffic flow time series collected from the CalTrans PeMS |
| Wind | Energy | 15 mins | 48,673 | 7 | 7:1:2 | 1 | TSFM-Bench | Wind power records from 2020-2021 at 15-minute intervals |
| NYSE | Stock | 1 day | 1,243 | 5 | 7:1:2 | 1 | TSFM-Bench | Records opening price, closing price, trading volume, lowest price, and highest price |
| ETTm1 | Electricity | 15 mins | 57,600 | 7 | 6:2:2 | 96 | ProbTS | Power transformer 1, comprising seven indicators such as oil temperature and useful load |
| ETTm2 | Electricity | 15 mins | 57,600 | 7 | 6:2:2 | 96 | ProbTS | Power transformer 2, comprising seven indicators such as oil temperature and useful load |
| ETTh1 | Electricity | 1 hour | 14,400 | 7 | 6:2:2 | 96 | ProbTS | Power transformer 1, comprising seven indicators such as oil temperature and useful load |
| ETTh2 | Electricity | 1 hour | 14,400 | 7 | 6:2:2 | 96 | ProbTS | Power transformer 2, comprising seven indicators such as oil temperature and useful load |
| Weather | Environment | 10 mins | 52,696 | 21 | 7:1:2 | 96 | ProbTS | Recorded every for the whole year 2020, which contains 21 meteorological indicators |
| Electricity | Electricity | 1 hour | 26,304 | 321 | 7:1:2 | 96 | ProbTS | Electricity records the electricity consumption in kWh every 1 hour from 2012 to 2014 |
| Traffic | Traffic | 1 hour | 17,544 | 862 | 7:1:2 | 96 | ProbTS | Road occupancy rates measured by 862 sensors on San Francisco Bay area freeways |
| Exchange | Economic | 1 day | 7,588 | 8 | 7:1:2 | 96 | ProbTS | ExchangeRate collects the daily exchange rates of eight countries |
| ILI | Health | 1 week | 966 | 7 | 7:1:2 | 96 | ProbTS | Recorded indicators of patients data from Centers for Disease Control and Prevention |
| TFB-Yearly | Univariate | Yearly | 1,500 | 1 | / | / | TFB | Univariate Datasets with yearly frequency in TFB |
| TFB-Quarterly | Univariate | Quarterly | 1,514 | 1 | / | / | TFB | Univariate Datasets with quarterly frequency in TFB |
| TFB-Monthly | Univariate | Monthly | 1,674 | 1 | / | / | TFB | Univariate Datasets with monthly frequency in TFB |
| TFB-Weekly | Univariate | Weekly | 805 | 1 | / | / | TFB | Univariate Datasets with weekly frequency in TFB |
| TFB-Daily | Univariate | Daily | 1,484 | 1 | / | / | TFB | Univariate Datasets with daily frequency in TFB |
| TFB-Hourly | Univariate | Hourly | 706 | 1 | / | / | TFB | Univariate Datasets with hourly frequency in TFB |
| TFB-Other | Univariate | Other | 706 | 1 | / | / | TFB | Univariate Datasets with other frequencies in TFB |
| NP | Electricity Price | 1 Hour | 52,179 | 2 | 7:1:2 | 1 | EPF | Using Grid Load and Wind Power to forecast Nord Pool Electricity Price. |
| PJM | Electricity Price | 1 Hour | 52,179 | 2 | 7:1:2 | 1 | EPF | Using System Loads to forecast Pennsylvania-New Jersey-Maryland Electricity Price. |
| BE | Electricity Price | 1 Hour | 52,179 | 2 | 7:1:2 | 1 | EPF | Using Generation and System Load to forecast Belgium's Electricity Price. |
| FR | Electricity Price | 1 Hour | 52,179 | 2 | 7:1:2 | 1 | EPF | Using Generation and System Load to forecast France's Electricity Price. |
| DE | Electricity Price | 1 Hour | 52,179 | 2 | 7:1:2 | 1 | EPF | Using Wind power and Ampirion zonal load to forecast German's Electricity Price. |

## A.4 Experimental Settings

**Pretraining**  In the training of Aurora, we utilize Distributed Data Parallel within the PyTorch framework, as referenced in (Paszke et al., 2019). Due to the limited computational resources, all experiments are executed on only 8 NVIDIA A800 GPUs, each equipped with 80GB of GPU memory, which *takes about 30 days to train Aurora from scratch.* The model is optimized by the AdamW optimizer, with an initial learning rate of $5 \times 10^{-5}$. To gradually decrease the learning rate throughout the training process, we implement a step decay schedule through the StepLR scheduler. The code bases described above are incorporated into the Huggingface framework. During the pre-training phase, we utilize 11 historical time series tokens and 4 prediction tokens, with a reference patch size of $p = 48$. The batch size is configured to be 8,192.

**Downstream Forecasting**  In the context of downstream forecasting tasks, we implement periodic patching strategies that are tailored to the temporal characteristics of each dataset. The quantity of past tokens is maintained at a constant value of 11.

Furthermore, we tackle the *"Drop Last"* issue, which has been emphasized in recent research works (Qiu et al., 2024; 2025c; Li et al., 2025c). Specifically, when *drop_last* is set to True during test evaluation, it may yield misleading outcomes because of incomplete batches. To uphold consistency and fairness, we configure *drop_last* as False for all baseline models within our experimental setup. In TSFM-Bench and ProbTS, all full-shot end-to-end baselines such as TimeKAN, TimePro and AMD about deterministic forecasting, and CSDI, TSDiff, and TimeGrad about probabilistic forecasting, follow the commonly-used settings, where the input sequence length equals to 96. In EPF, all baselines follow the setting of input-168-output-24. In TFB, they follow the default input lengths in short-term forecasting. The multimodal baselines such as TimeVLM, CALF also follow the default settings in TimeMMD.

## A.5 Evaluation Metrics

With respect to evaluation metrics, in accordance with the experimental setup in TSFM-Bench and TimeMMD, for deterministic forecasting, we employ the Mean Squared Error (MSE) and Mean Absolute Error (MAE) as evaluation metrics. In the context of probabilistic forecasting, within ProbTS, we utilize the Continuous Ranked Probability Score (CRPS) and Normalized Mean Absolute Error (NMAE). Consider the scenario featuring $K$ variates and a forecasting horizon of $T$.

**Mean Squared Error (MSE)**   The Mean Squared Error (MSE) serves to quantify the average of the squared discrepancies between the predicted values and their respective ground truth values. The squaring operation within the calculation of MSE results in a more substantial penalty for larger errors. This characteristic renders the MSE highly sensitive to outliers. In a formal sense, the Mean Squared Error is defined as follows:

$$\text{MSE} = \frac{1}{K \times T} \sum_{k=1}^{K} \sum_{t=1}^{T} (x_t^k - \hat{x}_t^k)^2, \tag{26}$$

where $K$ denotes the number of variables, $T$ the prediction horizon, $x_t^k$ the true value, and $\hat{x}_t^k$ the predicted value.

**Mean Absolute Error (MAE)**   The Mean Absolute Error computes the average magnitude of prediction errors, disregarding their direction. By concentrating on the absolute differences, the MAE offers a robust and interpretable metric of accuracy.

$$\text{MAE} = \frac{1}{K \times T} \sum_{k=1}^{K} \sum_{t=1}^{T} |x_t^k - \hat{x}_t^k|, \tag{27}$$

where all terms adhere to the same definition as stated above. In contrast to Mean Squared Error (MSE), Mean Absolute Error (MAE) accords equal treatment to all errors and exhibits lower sensitivity to substantial deviations.

**Continuous Ranked Probability Score (CRPS)**   The CRPS evaluates the quality of probabilistic forecasts by contrasting the predicted cumulative distribution function (CDF) $F$ with the observed outcome $x$. The calculation is as follows:

$$\text{CRPS} = \int_{\mathbb{R}} (F(z) - \mathbb{I}\{x \leq z\})^2 dz, \tag{28}$$

where $\mathbb{I}\{x \leq z\}$ represents the indicator function. The Continuous Ranked Probability Score (CRPS) rewards distributions that assign a high probability to the true value and attains its minimum when the predicted distribution coincides with the true distribution. In practical applications, we approximate the CRPS by utilizing the empirical Cumulative Distribution Function (CDF) $\hat{F}(z) = \frac{1}{n} \sum_{i=1}^{n} \mathbb{I}\{X_i \leq z\}$, which is based on $n = 100$ samples drawn from the conditional predictive distribution $p_\theta(\boldsymbol{x}_t | \boldsymbol{h}_t)$.

**Normalized Mean Absolute Error (NMAE)**   The NMAE is an extension of the MAE. It normalizes the MAE with respect to the total magnitude of the ground-truth values. This normalization process facilitates a fair comparison across datasets that have different scales. The formula for NMAE is as follows:

$$\text{NMAE} = \frac{\sum_{k=1}^{K} \sum_{t=1}^{T} |x_t^k - \hat{x}_t^k|}{\sum_{k=1}^{K} \sum_{t=1}^{T} |x_t^k|} \tag{29}$$

A.6   MODEL CONFIGURATIONS

Table 8: Detailed model configurations of Aurora, including the layers of Encoder, Decoder (Transformers for Time Modality), Flow-Matching Network, TextDistller, VisionDistiller, TextGuider, VisionGuider, the sizes of Prototype Bank, Model Dimension and FFN Dimension.

| Model | Encoder | Decoder | Flow-Net | Model Dim | FFN Dim | Prototype Bank | Distiller | Guider | Parameters |
|---|---|---|---|---|---|---|---|---|---|
| Aurora | 1 | 9 | 3 | 256 | 512 | 1,000 | 1 | 1 | 210.8M |

## A.7 EFFICIENCY ANALYSIS

Table 9: Efficiency analysis of Aurora and other baselines on Environment dataset, evaluated with the horizon of 336 and batch size of 1. We report the Parameter scale, MACs, Max GPU Memory, and Inference Time.

| Models | Parameters | MACs | GPU (MB) | Inference (ms) |
|---|---|---|---|---|
| TATS | 84.0 M | 0.015 G | 670 | 30.3 |
| GPT4MTS | 167.5 M | 1.210 G | 1,008 | 61.2 |
| CALF | 211.2 M | 0.724 G | 839 | 44.7 |
| Time-VLM | 152.2 M | 6.2 G | 1,773 | 57.6 |
| Sundial | 128.3 M | 1.320 G | 586 | 81.3 |
| VisionTS | 111.9 M | 5.510 G | 468 | 7.4 |
| MOIRAI | 311.0 M | 4.23 G | 1,280 | 51.0 |
| Time-MoE | 453.2 M | 19.21 G | 1,807 | 31.4 |
| Aurora | 210.8 M | 18.329 G | 1,265 | 83.5 |

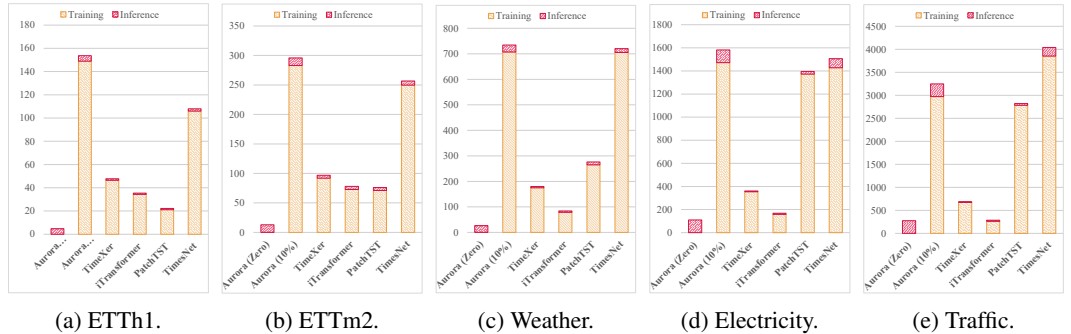

| (a) ETTh1. | (b) ETTm2. | (c) Weather. | (d) Electricity. | (e) Traffic. |
|---|---|---|---|---|

Figure 9: Time cost comparision (seconds) among Aurora (Zero-shot), Aurora (10% few-shot), TimeXer, iTransformer, PatchTST, and TimesNet, on datasets ETTh1, ETTm2, Weather, Electricity, and Traffic. The training and inference time are reported with batch size equals 64 in both phases.

# B FULL RESULTS

Table 10: Full **MASE** results of zero-shot forecasting experiments on datasets from TFB. Lower values indicate better predictions. **Red**: the best, Blue: the 2nd best.

| Type | 🚫 Zero-shot Foundation Models | | | | | | ⏱ Full-shot End-to-end Supervised Models | | | | | | | | | | | |
|---|---|---|---|---|---|---|---|---|---|---|---|---|---|---|---|---|---|---|
| Models | Aurora (Ours) | Sundial (2025) | VisionTS (2025) | Timer (2024) | TimeKAN (2025) | Ampilifier (2025) | iTransformer (2024) | TimeMixer (2024) | PatchTST (2023) | Crossformer (2023) | TimesNet (2023) | DLinear (2023) | N-HITS (2023) | Stationary (2022) | FEDformer (2022) | N-BEATS (2020) | TCN (2018) | LR (2005) | RF (2001) |
| Yearly | 3.597 | 3.579 | 3.441 | 4.17 | 3.982 | 5.035 | 4.461 | 3.559 | 4.131 | 24.461 | 4.276 | 3.968 | 5.349 | 3.776 | 4.030 | 6.013 | 29.205 | 17.146 | 4.276 |
| Quarterly | 1.932 | 1.991 | 1.849 | 2.259 | 2.129 | 1.620 | 2.155 | 1.819 | 2.405 | 19.335 | 2.281 | 2.057 | 2.013 | 2.205 | 2.136 | 1.943 | 20.208 | 1.500 | 2.026 |
| Monthly | 1.451 | 1.600 | 1.914 | 2.275 | 1.783 | 1.391 | 2.093 | 1.736 | 1.891 | 30.604 | 1.682 | 2.143 | 1.736 | 2.073 | 2.259 | 1.722 | 20.287 | 1.484 | 1.675 |
| Weekly | 1.962 | 1.839 | 1.628 | 3.47 | 1.418 | 1.349 | 1.765 | 2.042 | 1.691 | 45.160 | 1.871 | 1.983 | 1.762 | 1.504 | 2.106 | 2.252 | 6.907 | 1.890 | 3.466 |
| Daily | 1.252 | 1.364 | 1.165 | 1.284 | 1.193 | 1.133 | 1.271 | 1.313 | 1.183 | 21.092 | 1.170 | 1.331 | 1.396 | 1.260 | 1.403 | 1.305 | 7.928 | 1.162 | 1.301 |
| Hourly | 2.006 | 1.699 | 2.708 | 4.104 | 1.515 | 1.579 | 3.556 | 1.585 | 1.575 | 31.711 | 1.824 | 5.054 | 1.672 | 4.294 | 5.559 | 1.672 | 5.028 | 0.871 | 1.301 |
| Other | 4.200 | 4.271 | 3.541 | 4.142 | 4.771 | 2.727 | 4.250 | 4.433 | 4.410 | 74.249 | 4.318 | 4.540 | 5.007 | 3.824 | 4.553 | 3.930 | 47.121 | 2.696 | 4.624 |
| Avg | **2.134** | 2.158 | 2.167 | 2.810 | 2.232 | 2.140 | 2.593 | 2.159 | 2.347 | 29.224 | 2.340 | 2.670 | 2.551 | 2.353 | 2.791 | 2.639 | 18.274 | 4.440 | 2.406 |

Table 11: Full **msMAPE** results of zero-shot forecasting experiments on datasets from TFB. Lower values indicate better predictions. **Red**: the best, Blue: the 2nd best.

| Type | 🚫 Zero-shot Foundation Models | | | | | | ⏱ Full-shot End-to-end Supervised Models | | | | | | | | | | | |
|---|---|---|---|---|---|---|---|---|---|---|---|---|---|---|---|---|---|---|---|
| Models | Aurora (Ours) | Sundial (2025) | VisionTS (2025) | Timer (2024) | TimeKAN (2025) | Ampilifier (2025) | iTransformer (2024) | TimeMixer (2024) | PatchTST (2023) | Crossformer (2023) | TimesNet (2023) | DLinear (2023) | N-HITS (2023) | Stationary (2022) | FEDformer (2022) | N-BEATS (2020) | TCN (2018) | LR (2005) | RF (2001) |
| Quarterly | 17.655 | 17.852 | 17.121 | 19.485 | 18.920 | 17.830 | 19.771 | 18.257 | 21.992 | 199.760 | 19.914 | 18.472 | 20.602 | 19.995 | 19.060 | 19.252 | 192.593 | 18.740 | 18.666 |
| Monthly | 15.028 | 15.461 | 17.025 | 18.149 | 16.563 | 16.422 | 18.585 | 16.315 | 17.573 | 197.556 | 16.317 | 17.579 | 17.672 | 11.932 | 17.742 | 16.782 | 136.592 | 21.178 | 15.779 |
| Weekly | 18.839 | 18.908 | 19.359 | 32.63 | 18.873 | 18.724 | 20.041 | 21.595 | 19.690 | 170.084 | 20.506 | 50.882 | 35.622 | 19.704 | 69.550 | 61.084 | 73.372 | 65.833 | 47.956 |
| Daily | 22.318 | 22.506 | 22.331 | 22.466 | 21.720 | 21.010 | 22.401 | 22.788 | 21.709 | 137.802 | 21.607 | 24.269 | 26.550 | 24.140 | 28.350 | 25.708 | 78.426 | 26.779 | 24.214 |
| Hourly | 30.381 | 29.955 | 31.82 | 33.478 | 29.343 | 29.171 | 33.917 | 30.601 | 28.604 | 117.021 | 30.775 | 35.895 | 28.646 | 36.779 | 37.903 | 27.147 | 38.258 | 28.237 | 23.315 |
| Other | 16.165 | 16.531 | 14.666 | 15.682 | 17.223 | 11.825 | 15.463 | 16.477 | 16.708 | 178.247 | 15.755 | 16.737 | 14.322 | 14.876 | 15.806 | 12.845 | 121.324 | 12.605 | 13.185 |
| Avg | **19.964** | 20.096 | 20.317 | 23.153 | 20.774 | 21.048 | 22.528 | 21.024 | 21.866 | 176.571 | 21.479 | 25.089 | 24.787 | 21.775 | 28.061 | 26.925 | 132.472 | 29.787 | 22.822 |

Table 12: Full results of zero-shot & few-shot forecasting experiments on datasets from TimeMMD. Lower MSE or MAE values indicate better predictions. **Red**: the best, Blue: the 2nd best.

| Type | | Zero-shot Foundation Models | | | | | | | | | | 10% few-shot | | Full-shot Multimodal End-to-end Supervised Models | | | | | | | |
|---|---|---|---|---|---|---|---|---|---|---|---|---|---|---|---|---|---|---|---|---|---|
| Models | | Aurora (Ours) | | Sundial (2025) | | VisionTS (2025) | | ROSE (2025) | | MOIRAI (2024) | | Aurora (Ours) | | GPT4MTS (2025) | | TATS (2025) | | CALF (2025) | | Time-VLM (2025) | |
| Metrics | | MSE | MAE | MSE | MAE | MSE | MAE | MSE | MAE | MSE | MAE | MSE | MAE | MSE | MAE | MSE | MAE | MSE | MAE | MSE | MAE |
| Agriculture | 6 | **0.184** | 0.295 | 0.218 | 0.304 | 0.210 | **0.289** | 0.220 | 0.299 | 0.187 | 0.342 | **0.127** | **0.233** | 0.161 | 0.257 | 0.140 | 0.251 | 0.142 | 0.250 | 0.143 | 0.245 |
| | 8 | **0.242** | 0.335 | 0.319 | 0.364 | 0.266 | **0.323** | 0.278 | 0.339 | 0.245 | 0.391 | 0.190 | 0.289 | 0.207 | 0.288 | 0.187 | 0.282 | 0.195 | 0.285 | 0.215 | 0.287 |
| | 10 | **0.297** | 0.365 | 0.425 | 0.423 | 0.307 | 0.348 | 0.408 | 0.406 | 0.297 | 0.423 | 0.236 | 0.310 | 0.230 | 0.305 | 0.244 | 0.320 | 0.350 | 0.370 | 0.271 | 0.320 |
| | 12 | 0.365 | 0.398 | 0.530 | 0.477 | 0.376 | **0.386** | 0.474 | 0.443 | **0.357** | 0.455 | 0.295 | 0.340 | 0.301 | 0.342 | 0.290 | 0.350 | 0.314 | 0.355 | 0.322 | 0.359 |
| | Avg | **0.272** | 0.348 | 0.373 | 0.392 | 0.290 | **0.336** | 0.345 | 0.372 | 0.272 | 0.403 | **0.212** | **0.293** | 0.225 | 0.298 | 0.215 | 0.301 | 0.250 | 0.315 | 0.237 | 0.302 |
| Climate | 6 | **0.859** | **0.747** | 1.180 | 0.891 | 1.316 | 0.932 | 1.488 | 0.993 | 1.624 | 1.016 | **0.867** | **0.744** | 1.199 | 0.895 | 1.194 | 0.897 | 1.231 | 0.910 | 1.218 | 0.907 |
| | 8 | **0.858** | **0.746** | 1.159 | 0.885 | 1.312 | 0.935 | 1.598 | 1.031 | 2.148 | 1.152 | **0.858** | **0.745** | 1.205 | 0.899 | 1.178 | 0.886 | 1.227 | 0.905 | 1.181 | 0.914 |
| | 10 | **0.868** | **0.748** | 1.141 | 0.876 | 1.302 | 0.928 | 1.401 | 0.967 | 1.983 | 1.112 | **0.863** | **0.744** | 1.173 | 0.885 | 1.170 | 0.881 | 1.508 | 0.990 | 1.179 | 0.880 |
| | 12 | **0.875** | **0.753** | 1.134 | 0.870 | 1.297 | 0.925 | 1.414 | 0.957 | 1.929 | 1.101 | **0.869** | **0.749** | 1.152 | 0.876 | 1.179 | 0.885 | 1.177 | 0.883 | 1.203 | 0.896 |
| | Avg | **0.865** | **0.749** | 1.154 | 0.881 | 1.307 | 0.930 | 1.475 | 0.987 | 1.921 | 1.095 | **0.862** | **0.746** | 1.182 | 0.889 | 1.180 | 0.887 | 1.286 | 0.922 | 1.195 | 0.899 |
| Economy | 6 | **0.035** | **0.150** | 0.251 | 0.401 | 0.270 | 0.420 | 0.258 | 0.405 | 0.315 | 0.460 | **0.015** | **0.095** | 0.016 | 0.102 | 0.017 | 0.102 | 0.178 | 0.334 | 0.024 | 0.125 |
| | 8 | **0.033** | **0.145** | 0.277 | 0.423 | 0.296 | 0.440 | 0.300 | 0.450 | 0.431 | 0.526 | **0.015** | **0.099** | 0.016 | 0.101 | 0.017 | 0.103 | 0.200 | 0.353 | 0.023 | 0.121 |
| | 10 | **0.032** | **0.143** | 0.304 | 0.443 | 0.307 | 0.446 | 0.286 | 0.432 | 0.432 | 0.528 | **0.016** | **0.101** | 0.018 | 0.104 | 0.017 | 0.104 | 0.039 | 0.162 | 0.025 | 0.128 |
| | 12 | **0.032** | **0.144** | 0.333 | 0.460 | 0.329 | 0.462 | 0.310 | 0.447 | 0.440 | 0.535 | **0.016** | **0.102** | 0.017 | 0.104 | 0.017 | 0.106 | 0.232 | 0.378 | 0.024 | 0.126 |
| | Avg | **0.033** | **0.146** | 0.291 | 0.432 | 0.301 | 0.442 | 0.289 | 0.433 | 0.405 | 0.512 | **0.016** | **0.099** | 0.017 | 0.103 | 0.017 | 0.104 | 0.163 | 0.307 | 0.024 | 0.125 |
| Energy | 12 | **0.117** | 0.245 | 0.125 | 0.242 | 0.173 | 0.313 | 0.268 | 0.401 | 0.183 | 0.309 | **0.097** | **0.212** | 0.111 | 0.244 | 0.105 | 0.232 | 0.102 | 0.224 | 0.114 | 0.253 |
| | 24 | **0.226** | 0.354 | 0.234 | 0.345 | 0.264 | 0.395 | 0.363 | 0.469 | 0.290 | 0.396 | **0.199** | **0.322** | 0.232 | 0.362 | 0.216 | 0.344 | 0.210 | 0.346 | 0.227 | 0.359 |
| | 36 | **0.292** | 0.409 | 0.318 | 0.409 | 0.346 | 0.454 | 0.413 | 0.497 | 0.367 | 0.449 | **0.271** | **0.352** | 0.308 | 0.418 | 0.309 | 0.418 | 0.300 | 0.420 | 0.309 | 0.410 |
| | 48 | **0.383** | 0.472 | 0.410 | 0.473 | 0.434 | 0.516 | 0.501 | 0.549 | 0.457 | 0.515 | **0.352** | **0.431** | 0.398 | 0.496 | 0.391 | 0.480 | 0.365 | 0.470 | 0.390 | 0.475 |
| | Avg | **0.255** | 0.370 | 0.272 | 0.367 | 0.304 | 0.420 | 0.386 | 0.479 | 0.324 | 0.417 | **0.230** | **0.329** | 0.262 | 0.380 | 0.255 | 0.368 | 0.244 | 0.365 | 0.260 | 0.374 |
| Environment | 48 | **0.281** | 0.380 | 0.330 | 0.410 | 0.345 | 0.426 | 0.402 | 0.459 | 0.352 | 0.404 | **0.269** | **0.372** | 0.315 | 0.400 | 0.307 | 0.389 | 0.313 | 0.382 | 0.304 | 0.387 |
| | 96 | **0.284** | 0.382 | 0.353 | 0.423 | 0.370 | 0.441 | 0.409 | 0.465 | 0.370 | 0.415 | **0.271** | **0.373** | 0.340 | 0.401 | 0.334 | 0.402 | 0.335 | 0.394 | 0.327 | 0.405 |
| | 192 | **0.270** | 0.375 | 0.343 | 0.419 | 0.360 | 0.442 | 0.389 | 0.452 | 0.350 | 0.402 | **0.269** | **0.374** | 0.336 | 0.411 | 0.332 | 0.401 | 0.341 | 0.394 | 0.328 | 0.403 |
| | 336 | **0.269** | 0.377 | 0.317 | 0.411 | 0.340 | 0.436 | 0.369 | 0.447 | 0.332 | 0.390 | **0.251** | **0.368** | 0.299 | 0.390 | 0.302 | 0.391 | 0.312 | 0.377 | 0.320 | 0.395 |
| | Avg | **0.276** | 0.379 | 0.336 | 0.416 | 0.354 | 0.436 | 0.392 | 0.456 | 0.351 | 0.403 | **0.265** | **0.372** | 0.323 | 0.400 | 0.319 | 0.396 | 0.325 | 0.387 | 0.319 | 0.397 |
| Health | 12 | **1.093** | 0.668 | 1.531 | 0.810 | 2.012 | 1.093 | 2.737 | 1.250 | 2.230 | 1.114 | 0.992 | 0.641 | 0.985 | 0.658 | 0.899 | 0.612 | 0.964 | 0.609 | 1.198 | 0.727 |
| | 24 | **1.572** | 0.849 | 2.075 | 1.019 | 2.594 | 1.266 | 2.589 | 1.189 | 2.895 | 1.284 | 1.332 | 0.796 | 1.513 | 0.802 | 1.307 | 0.759 | 1.451 | 0.749 | 1.491 | 0.839 |
| | 36 | **1.688** | 0.920 | 2.122 | 1.058 | 2.686 | 1.291 | 2.629 | 1.210 | 2.924 | 1.289 | 1.467 | 0.818 | 1.601 | 0.846 | 1.523 | 0.827 | 1.713 | 0.851 | 1.567 | 0.865 |
| | 48 | **1.857** | 0.963 | 2.153 | 1.081 | 2.454 | 1.236 | 2.436 | 1.154 | 2.895 | 1.276 | 1.579 | 0.847 | 1.757 | 0.889 | 1.693 | 0.872 | 1.836 | 0.889 | 1.702 | 0.907 |
| | Avg | **1.553** | 0.850 | 1.970 | 0.992 | 2.436 | 1.221 | 2.598 | 1.201 | 2.736 | 1.241 | **1.343** | 0.776 | 1.464 | 0.799 | 1.356 | 0.767 | 1.491 | 0.775 | 1.489 | 0.834 |
| Security | 6 | 67.572 | 3.909 | 64.519 | 3.781 | 71.453 | 4.286 | 78.188 | 4.574 | 69.454 | 4.419 | **64.513** | 3.798 | 65.780 | 3.906 | 65.612 | 3.838 | 67.427 | 3.947 | 67.867 | 3.992 |
| | 8 | 70.576 | 4.013 | 68.380 | 3.934 | 78.023 | 4.573 | 90.703 | 5.089 | 97.574 | 5.316 | **67.828** | 3.930 | 68.914 | 3.955 | 71.860 | 4.146 | 69.608 | 3.993 | 70.928 | 4.084 |
| | 10 | 74.173 | 4.148 | 72.290 | 4.068 | 81.893 | 4.669 | 82.339 | 4.655 | 100.900 | 5.419 | 72.423 | 4.092 | 73.214 | 4.004 | 74.494 | 4.166 | 93.839 | 5.146 | 75.362 | 4.212 |
| | 12 | 77.579 | 4.264 | 76.573 | 4.238 | 87.023 | 4.861 | 86.063 | 4.741 | 105.053 | 5.536 | 75.482 | 4.132 | 78.041 | 4.316 | 77.656 | 4.239 | 74.631 | 4.113 | 80.767 | 4.438 |
| | Avg | 72.475 | 4.084 | 70.441 | 4.005 | 79.598 | 4.597 | 84.324 | 4.765 | 93.245 | 5.173 | 70.062 | 3.988 | 71.487 | 4.068 | 72.406 | 4.097 | 76.376 | 4.300 | 73.731 | 4.181 |
| Social Good | 6 | **0.701** | 0.442 | 0.861 | 0.487 | 0.957 | 0.543 | 0.939 | 0.499 | 0.966 | 0.522 | **0.689** | 0.427 | 0.718 | 0.378 | 0.753 | 0.370 | 0.782 | 0.360 | 0.732 | 0.379 |
| | 8 | **0.804** | 0.493 | 0.994 | 0.549 | 1.106 | 0.605 | 1.168 | 0.588 | 1.532 | 0.653 | **0.784** | 0.461 | 0.942 | 0.505 | 0.875 | 0.409 | 0.874 | 0.386 | 0.822 | 0.427 |
| | 10 | **0.886** | 0.543 | 1.100 | 0.604 | 1.164 | 0.636 | 1.187 | 0.595 | 1.551 | 0.691 | **0.850** | 0.532 | 0.929 | 0.446 | 0.991 | 0.459 | 0.976 | 0.420 | 0.916 | 0.465 |
| | 12 | **0.960** | 0.587 | 1.187 | 0.651 | 1.278 | 0.688 | 1.272 | 0.642 | 1.671 | 0.736 | **0.931** | 0.554 | 1.093 | 0.470 | 1.053 | 0.474 | 0.991 | 0.439 | 1.005 | 0.505 |
| | Avg | **0.838** | 0.516 | 1.036 | 0.573 | 1.126 | 0.618 | 1.141 | 0.581 | 1.430 | 0.651 | **0.814** | 0.494 | 0.920 | 0.450 | 0.918 | 0.428 | 0.906 | 0.401 | 0.868 | 0.444 |
| Traffic | 6 | **0.154** | 0.285 | 0.273 | 0.410 | 0.275 | 0.411 | 0.331 | 0.449 | 0.349 | 0.448 | **0.149** | 0.292 | 0.192 | 0.264 | 0.164 | 0.226 | 0.174 | 0.243 | 0.210 | 0.316 |
| | 8 | **0.158** | 0.286 | 0.275 | 0.408 | 0.282 | 0.410 | 0.365 | 0.455 | 0.461 | 0.499 | **0.155** | 0.284 | 0.195 | 0.256 | 0.178 | 0.242 | 0.176 | 0.232 | 0.212 | 0.313 |
| | 10 | **0.163** | 0.289 | 0.270 | 0.403 | 0.286 | 0.406 | 0.326 | 0.443 | 0.414 | 0.466 | **0.160** | 0.287 | 0.204 | 0.257 | 0.185 | 0.243 | 0.345 | 0.454 | 0.222 | 0.328 |
| | 12 | **0.168** | 0.294 | 0.268 | 0.401 | 0.282 | 0.402 | 0.342 | 0.458 | 0.400 | 0.458 | **0.165** | 0.296 | 0.218 | 0.268 | 0.189 | 0.242 | 0.193 | 0.243 | 0.222 | 0.322 |
| | Avg | **0.161** | 0.289 | 0.271 | 0.405 | 0.281 | 0.407 | 0.341 | 0.451 | 0.406 | 0.468 | **0.157** | 0.290 | 0.203 | 0.261 | 0.179 | 0.238 | 0.222 | 0.293 | 0.216 | 0.319 |
| 1st Count | | **31** | **26** | 4 | 7 | 0 | 4 | 0 | 0 | 1 | 0 | **30** | **23** | 1 | 1 | 4 | 4 | 1 | 8 | 0 | 0 |

Table 13: Full results of zero-shot deterministic forecasting experiments on datasets from TSFM-Bench. Lower MSE or MAE values indicate better predictions. ('-') denotes datasets included in the model's pretraining and therefore excluded from testing. **Red**: the best, Blue: the 2nd best.

| Type | | \multicolumn Zero-shot Foundation Models | | | | | | | | | | | | | | | | | | | |
|---|---|---|---|---|---|---|---|---|---|---|---|---|---|---|---|---|---|---|---|---|---|
| Models | | Aurora (Ours) | | Sundial (2025) | | ROSE (2025) | | Timer (2024) | | TimesFM (2024) | | Chronos (2023) | | Time-MoE (2024) | | UniTS (2024) | | MOIRAI (2024) | | TTM (2024) | |
| Metrics | | MSE | MAE | MSE | MAE | MSE | MAE | MSE | MAE | MSE | MAE | MSE | MAE | MSE | MAE | MSE | MAE | MSE | MAE | MSE | MAE |
| ETTm1 | 96 | 0.294 | 0.351 | 0.280 | 0.334 | 0.512 | 0.460 | 0.817 | 0.611 | 0.363 | 0.369 | 0.402 | 0.373 | 0.281 | 0.341 | 0.761 | 0.530 | 0.353 | 0.363 | 0.738 | 0.541 |
| | 192 | 0.331 | 0.374 | 0.321 | 0.366 | 0.512 | 0.462 | 0.927 | 0.659 | 0.417 | 0.405 | 0.510 | 0.435 | 0.305 | 0.358 | 0.777 | 0.534 | 0.376 | 0.380 | 0.698 | 0.547 |
| | 336 | 0.359 | 0.391 | 0.350 | 0.389 | 0.523 | 0.470 | 1.043 | 0.704 | 0.447 | 0.428 | 0.590 | 0.477 | 0.369 | 0.395 | 0.754 | 0.539 | 0.399 | 0.395 | 0.670 | 0.533 |
| | 720 | 0.398 | 0.414 | 0.394 | 0.418 | 0.552 | 0.490 | 1.044 | 0.722 | 0.513 | 0.470 | 0.703 | 0.525 | 0.469 | 0.472 | 0.750 | 0.569 | 0.432 | 0.417 | 0.660 | 0.550 |
| | Avg | 0.346 | 0.383 | 0.336 | 0.377 | 0.525 | 0.471 | 0.958 | 0.674 | 0.435 | 0.418 | 0.551 | 0.453 | 0.356 | 0.392 | 0.761 | 0.543 | 0.390 | 0.389 | 0.692 | 0.543 |
| ETTm2 | 96 | 0.179 | 0.270 | 0.170 | 0.256 | 0.224 | 0.309 | 0.225 | 0.300 | 0.206 | 0.267 | 0.192 | 0.263 | 0.198 | 0.288 | 0.249 | 0.315 | 0.189 | 0.260 | 0.226 | 0.309 |
| | 192 | 0.232 | 0.307 | 0.229 | 0.300 | 0.266 | 0.333 | 0.286 | 0.339 | 0.293 | 0.320 | 0.256 | 0.308 | 0.235 | 0.312 | 0.309 | 0.353 | 0.247 | 0.300 | 0.311 | 0.360 |
| | 336 | 0.275 | 0.337 | 0.281 | 0.337 | 0.310 | 0.358 | 0.335 | 0.369 | 0.411 | 0.414 | 0.315 | 0.346 | 0.293 | 0.348 | 0.353 | 0.383 | 0.295 | 0.334 | 0.350 | 0.383 |
| | 720 | 0.338 | 0.380 | 0.351 | 0.387 | 0.395 | 0.407 | 0.414 | 0.416 | 0.478 | 0.437 | 0.409 | 0.405 | 0.427 | 0.428 | 0.430 | 0.431 | 0.372 | 0.386 | 0.446 | 0.435 |
| | Avg | 0.256 | 0.324 | 0.258 | 0.320 | 0.299 | 0.352 | 0.315 | 0.356 | 0.347 | 0.360 | 0.293 | 0.331 | 0.288 | 0.344 | 0.335 | 0.371 | 0.276 | 0.320 | 0.333 | 0.372 |
| ETTh1 | 96 | 0.340 | 0.381 | 0.348 | 0.385 | 0.382 | 0.408 | 0.454 | 0.434 | 0.421 | 0.401 | 0.444 | 0.409 | 0.349 | 0.379 | 0.377 | 0.392 | 0.380 | 0.398 | 0.364 | 0.389 |
| | 192 | 0.377 | 0.405 | 0.393 | 0.418 | 0.400 | 0.420 | 0.522 | 0.465 | 0.472 | 0.432 | 0.502 | 0.443 | 0.395 | 0.413 | 0.398 | 0.421 | 0.440 | 0.434 | 0.386 | 0.407 |
| | 336 | 0.399 | 0.422 | 0.422 | 0.440 | 0.404 | 0.426 | 0.559 | 0.484 | 0.510 | 0.455 | 0.580 | 0.460 | 0.447 | 0.453 | 0.413 | 0.425 | 0.514 | 0.473 | 0.404 | 0.422 |
| | 720 | 0.428 | 0.450 | 0.481 | 0.493 | 0.420 | 0.447 | 0.714 | 0.549 | 0.514 | 0.481 | 0.605 | 0.495 | 0.457 | 0.462 | 0.469 | 0.463 | 0.705 | 0.568 | 0.424 | 0.448 |
| | Avg | 0.386 | 0.415 | 0.411 | 0.434 | 0.402 | 0.425 | 0.562 | 0.483 | 0.479 | 0.442 | 0.533 | 0.452 | 0.412 | 0.427 | 0.414 | 0.425 | 0.510 | 0.469 | 0.395 | 0.417 |
| ETTh2 | 96 | 0.259 | 0.325 | 0.271 | 0.333 | 0.298 | 0.362 | 0.316 | 0.359 | 0.326 | 0.355 | 0.306 | 0.338 | 0.292 | 0.352 | 0.323 | 0.355 | 0.287 | 0.325 | 0.277 | 0.335 |
| | 192 | 0.324 | 0.370 | 0.327 | 0.376 | 0.336 | 0.385 | 0.374 | 0.398 | 0.397 | 0.400 | 0.396 | 0.394 | 0.347 | 0.379 | 0.372 | 0.406 | 0.347 | 0.367 | 0.334 | 0.373 |
| | 336 | 0.360 | 0.401 | 0.354 | 0.402 | 0.353 | 0.399 | 0.381 | 0.410 | 0.431 | 0.413 | 0.423 | 0.417 | 0.406 | 0.419 | 0.373 | 0.413 | 0.377 | 0.393 | 0.362 | 0.402 |
| | 720 | 0.403 | 0.441 | 0.381 | 0.435 | 0.395 | 0.432 | 0.408 | 0.434 | 0.446 | 0.444 | 0.442 | 0.439 | 0.439 | 0.447 | 0.429 | 0.457 | 0.404 | 0.421 | 0.408 | 0.444 |
| | Avg | 0.337 | 0.384 | 0.333 | 0.387 | 0.346 | 0.395 | 0.370 | 0.400 | 0.400 | 0.403 | 0.392 | 0.397 | 0.371 | 0.399 | 0.374 | 0.408 | 0.354 | 0.377 | 0.345 | 0.389 |
| Weather | 96 | 0.160 | 0.207 | 0.157 | 0.205 | 0.200 | 0.260 | 0.190 | 0.236 | - | - | 0.186 | 0.208 | 0.157 | 0.211 | 0.194 | 0.234 | 0.177 | 0.208 | 0.183 | 0.242 |
| | 192 | 0.202 | 0.247 | 0.205 | 0.251 | 0.239 | 0.288 | 0.261 | 0.293 | - | - | 0.238 | 0.258 | 0.208 | 0.256 | 0.252 | 0.279 | 0.219 | 0.249 | 0.229 | 0.285 |
| | 336 | 0.252 | 0.288 | 0.253 | 0.289 | 0.279 | 0.315 | 0.332 | 0.340 | - | - | 0.313 | 0.353 | 0.255 | 0.290 | 0.299 | 0.316 | 0.277 | 0.292 | 0.289 | 0.330 |
| | 720 | 0.307 | 0.327 | 0.320 | 0.336 | 0.340 | 0.357 | 0.385 | 0.381 | - | - | 0.416 | 0.415 | 0.405 | 0.397 | 0.355 | 0.361 | 0.365 | 0.350 | 0.359 | 0.370 |
| | Avg | 0.230 | 0.267 | 0.234 | 0.270 | 0.265 | 0.305 | 0.292 | 0.313 | - | - | 0.288 | 0.309 | 0.256 | 0.289 | 0.275 | 0.298 | 0.260 | 0.275 | 0.265 | 0.307 |
| Electricity | 96 | 0.134 | 0.234 | 0.132 | 0.229 | 0.209 | 0.307 | 0.210 | 0.312 | - | - | - | - | - | - | 0.175 | 0.269 | 0.152 | 0.242 | 0.166 | 0.263 |
| | 192 | 0.161 | 0.258 | 0.152 | 0.250 | 0.219 | 0.315 | 0.239 | 0.337 | - | - | - | - | - | - | 0.178 | 0.273 | 0.171 | 0.259 | 0.191 | 0.286 |
| | 336 | 0.193 | 0.287 | 0.173 | 0.271 | 0.236 | 0.330 | 0.284 | 0.372 | - | - | - | - | - | - | 0.190 | 0.287 | 0.192 | 0.278 | 0.237 | 0.336 |
| | 720 | 0.224 | 0.320 | 0.218 | 0.311 | 0.273 | 0.328 | 0.456 | 0.479 | - | - | - | - | - | - | 0.248 | 0.335 | 0.236 | 0.313 | 0.292 | 0.384 |
| | Avg | 0.178 | 0.275 | 0.169 | 0.265 | 0.234 | 0.320 | 0.297 | 0.375 | - | - | - | - | - | - | 0.198 | 0.291 | 0.188 | 0.273 | 0.222 | 0.317 |
| Traffic | 96 | 0.435 | 0.314 | - | - | 0.572 | 0.407 | 0.526 | 0.368 | - | - | 0.562 | 0.378 | - | - | - | - | - | - | 0.514 | 0.347 |
| | 192 | 0.465 | 0.328 | - | - | 0.575 | 0.406 | 0.561 | 0.385 | - | - | 0.579 | 0.412 | - | - | - | - | - | - | 0.543 | 0.373 |
| | 336 | 0.525 | 0.355 | - | - | 0.588 | 0.411 | 0.614 | 0.412 | - | - | 0.594 | 0.420 | - | - | - | - | - | - | 0.575 | 0.389 |
| | 720 | 0.670 | 0.411 | - | - | 0.618 | 0.422 | 0.749 | 0.464 | - | - | 0.723 | 0.472 | - | - | - | - | - | - | 0.622 | 0.433 |
| | Avg | 0.524 | 0.352 | - | - | 0.588 | 0.412 | 0.613 | 0.407 | - | - | 0.615 | 0.421 | - | - | - | - | - | - | 0.564 | 0.386 |
| Solar | 96 | 0.185 | 0.272 | 0.204 | 0.230 | 0.524 | 0.557 | 0.591 | 0.504 | 0.408 | 0.345 | 0.373 | 0.304 | 0.304 | 0.345 | 0.771 | 0.594 | 0.682 | 0.688 | 0.863 | 0.664 |
| | 192 | 0.198 | 0.282 | 0.221 | 0.248 | 0.507 | 0.550 | 0.689 | 0.567 | 0.466 | 0.373 | 0.363 | 0.303 | 0.309 | 0.342 | 0.800 | 0.618 | 0.694 | 0.695 | 0.823 | 0.695 |
| | 336 | 0.211 | 0.294 | 0.225 | 0.260 | 0.508 | 0.553 | 0.831 | 0.636 | 0.526 | 0.407 | 0.391 | 0.319 | 0.433 | 0.450 | 0.855 | 0.672 | 0.719 | 0.706 | 0.835 | 0.741 |
| | 720 | 0.218 | 0.307 | 0.233 | 0.272 | 0.479 | 0.534 | 0.972 | 0.710 | 0.601 | 0.461 | 0.444 | 0.349 | 0.599 | 0.576 | 0.952 | 0.793 | 0.759 | 0.725 | 0.738 | 0.738 |
| | Avg | 0.203 | 0.289 | 0.221 | 0.252 | 0.505 | 0.549 | 0.771 | 0.604 | 0.500 | 0.397 | 0.393 | 0.319 | 0.411 | 0.428 | 0.845 | 0.669 | 0.714 | 0.704 | 0.815 | 0.710 |
| PEMS08 | 96 | 0.463 | 0.513 | - | - | 1.373 | 0.995 | 0.625 | 0.580 | 1.131 | 0.759 | 1.538 | 0.983 | - | - | 1.152 | 0.833 | - | - | 1.284 | 0.888 |
| | 192 | 0.599 | 0.577 | - | - | 1.365 | 0.979 | 0.798 | 0.661 | 1.609 | 0.944 | 1.719 | 0.983 | - | - | 1.259 | 0.875 | - | - | 1.638 | 1.039 |
| | 336 | 0.560 | 0.546 | - | - | 1.338 | 0.960 | 0.910 | 0.716 | 1.568 | 0.939 | 1.768 | 0.998 | - | - | 1.309 | 0.903 | - | - | 1.979 | 1.160 |
| | 720 | 0.629 | 0.570 | - | - | 1.401 | 0.980 | 1.131 | 0.824 | 1.632 | 0.986 | 1.802 | 1.133 | - | - | 1.290 | 0.906 | - | - | 2.020 | 1.175 |
| | Avg | 0.563 | 0.552 | - | - | 1.369 | 0.979 | 0.866 | 0.695 | 1.485 | 0.907 | 1.707 | 1.024 | - | - | 1.253 | 0.879 | - | - | 1.730 | 1.066 |
| Wind | 96 | 0.951 | 0.664 | 0.931 | 0.646 | 1.072 | 0.747 | 0.946 | 0.659 | 1.229 | 0.722 | 1.273 | 0.738 | - | - | 1.112 | 0.724 | 0.992 | 0.656 | 1.077 | 0.701 |
| | 192 | 1.115 | 0.747 | 1.136 | 0.751 | 1.209 | 0.804 | 1.142 | 0.758 | 1.503 | 0.835 | 1.439 | 0.817 | - | - | 1.295 | 0.806 | 1.221 | 0.765 | 1.350 | 0.825 |
| | 336 | 1.229 | 0.799 | 1.285 | 0.820 | 1.318 | 0.848 | 1.300 | 0.830 | 1.550 | 0.869 | 1.550 | 0.869 | - | - | 1.526 | 0.890 | 1.403 | 0.844 | 1.473 | 0.891 |
| | 720 | 1.309 | 0.840 | 1.390 | 0.870 | 1.404 | 0.881 | 1.417 | 0.884 | 1.982 | 0.997 | 1.649 | 0.914 | - | - | 1.765 | 0.973 | 1.581 | 0.915 | 1.447 | 0.899 |
| | Avg | 1.151 | 0.763 | 1.186 | 0.772 | 1.251 | 0.820 | 1.201 | 0.783 | 1.613 | 0.870 | 1.478 | 0.834 | - | - | 1.425 | 0.848 | 1.299 | 0.795 | 1.337 | 0.829 |
| NYSE | 24 | 0.267 | 0.373 | 0.370 | 0.393 | - | - | 0.388 | 0.409 | 0.235 | 0.322 | 0.480 | 0.449 | - | - | 0.755 | 0.623 | - | - | - | - |
| | 36 | 0.420 | 0.477 | 0.681 | 0.566 | - | - | 0.778 | 0.613 | 0.472 | 0.467 | 0.912 | 0.655 | - | - | 1.058 | 0.763 | - | - | - | - |
| | 48 | 0.605 | 0.574 | 1.043 | 0.729 | - | - | 1.200 | 0.792 | 0.723 | 0.591 | 1.371 | 0.818 | - | - | 1.353 | 0.879 | - | - | - | - |
| | 60 | 0.820 | 0.681 | 1.425 | 0.880 | - | - | 1.584 | 1.000 | 1.061 | 0.763 | 1.754 | 0.958 | - | - | 1.712 | 1.016 | - | - | - | - |
| | Avg | 0.528 | 0.526 | 0.880 | 0.642 | - | - | 0.988 | 0.704 | 0.623 | 0.536 | 1.129 | 0.720 | - | - | 1.220 | 0.820 | - | - | - | - |
| 1st Count | | 27 | 21 | 11 | 13 | 3 | 1 | 0 | 0 | 1 | 2 | 0 | 0 | 2 | 2 | 0 | 0 | 0 | 5 | 0 | 0 |

Table 14: Full results of zero-shot probabilistic forecasting experiments on datasets from ProbTS. Lower CRPS or NMAE values indicate better predictions. ('-') denotes datasets included in the model's pretraining and therefore excluded from testing. ('/') denotes the excessive time consumption. **Red**: the best, Blue: the 2nd best.

| Type | | Zero-shot Foundation Models | | | | | | | | | | | | | | Full-shot Probabilistic End-to-end Supervised Models | | | | | | | |
|---|---|---|---|---|---|---|---|---|---|---|---|---|---|---|---|---|---|---|---|---|---|---|---|
| Models | | Aurora (Ours) | | Sundial (2025) | | Chronos (2024) | | MOIRAI (2024) | | Lag-Llama (2023) | | TSDiff (2023) | | CSDI (2022) | | TimeGrad (2022) | | GRU MAF (2021) | |
| Metrics | | CRPS | NMAE | CRPS | NMAE | CRPS | NMAE | CRPS | NMAE | CRPS | NMAE | CRPS | NMAE | CRPS | NMAE | CRPS | NMAE | CRPS | NMAE |
| ETTm1 | 96 | 0.261 | **0.278** | 0.253 | 0.308 | 0.360 | 0.422 | 0.464 | 0.522 | 0.354 | 0.402 | 0.344 | 0.441 | **0.236** | 0.308 | 0.522 | 0.645 | 0.295 | 0.402 |
| | 192 | **0.278** | **0.296** | 0.279 | 0.337 | 0.404 | 0.450 | 0.467 | 0.531 | 0.368 | 0.415 | 0.345 | 0.441 | 0.291 | 0.377 | 0.603 | 0.748 | 0.389 | 0.476 |
| | 336 | 0.292 | **0.309** | **0.291** | 0.350 | 0.425 | 0.456 | 0.524 | 0.558 | 0.387 | 0.436 | 0.462 | 0.571 | 0.322 | 0.419 | 0.601 | 0.759 | 0.429 | 0.522 |
| | 720 | **0.312** | **0.330** | 0.318 | 0.380 | 0.461 | 0.478 | 0.514 | 0.535 | 0.403 | 0.466 | 0.478 | 0.622 | 0.448 | 0.578 | 0.621 | 0.793 | 0.536 | 0.711 |
| | Avg | 0.286 | **0.303** | **0.285** | 0.344 | 0.413 | 0.451 | 0.492 | 0.536 | 0.378 | 0.430 | 0.407 | 0.519 | 0.324 | 0.421 | 0.587 | 0.736 | 0.412 | 0.528 |
| ETTm2 | 96 | 0.131 | 0.148 | 0.128 | 0.153 | 0.134 | 0.158 | 0.176 | 0.186 | 0.163 | 0.192 | 0.175 | 0.224 | **0.115** | **0.146** | 0.427 | 0.525 | 0.177 | 0.212 |
| | 192 | **0.149** | **0.168** | 0.150 | 0.177 | 0.163 | 0.183 | 0.197 | 0.207 | 0.181 | 0.207 | 0.255 | 0.316 | 0.147 | 0.189 | 0.424 | 0.530 | 0.411 | 0.535 |
| | 336 | **0.163** | **0.182** | 0.167 | 0.195 | 0.190 | 0.204 | 0.229 | 0.227 | 0.206 | 0.229 | 0.328 | 0.397 | 0.190 | 0.248 | 0.469 | 0.566 | 0.377 | 0.407 |
| | 720 | **0.182** | **0.201** | 0.189 | 0.217 | 0.223 | 0.230 | 0.321 | 0.258 | 0.227 | 0.249 | 0.344 | 0.416 | 0.239 | 0.306 | 0.470 | 0.561 | 0.272 | 0.355 |
| | Avg | **0.156** | **0.175** | 0.158 | 0.186 | 0.178 | 0.194 | 0.231 | 0.220 | 0.194 | 0.219 | 0.276 | 0.338 | 0.173 | 0.222 | 0.448 | 0.546 | 0.309 | 0.377 |
| ETTh1 | 96 | 0.277 | 0.314 | **0.260** | **0.307** | 0.293 | 0.341 | 0.469 | 0.488 | 0.297 | 0.340 | 0.395 | 0.510 | 0.437 | 0.557 | 0.455 | 0.585 | 0.293 | 0.371 |
| | 192 | **0.297** | **0.336** | 0.297 | 0.347 | 0.348 | 0.384 | 0.496 | 0.492 | 0.312 | 0.356 | 0.467 | 0.596 | 0.496 | 0.625 | 0.516 | 0.680 | 0.348 | 0.430 |
| | 336 | **0.314** | **0.357** | 0.314 | 0.365 | 0.384 | 0.409 | 0.503 | 0.489 | 0.326 | 0.370 | 0.450 | 0.581 | 0.454 | 0.574 | 0.512 | 0.666 | 0.377 | 0.462 |
| | 720 | 0.334 | 0.381 | **0.312** | **0.366** | 0.413 | 0.425 | 0.526 | 0.508 | 0.340 | 0.405 | 0.516 | 0.657 | 0.528 | 0.657 | 0.523 | 0.672 | 0.393 | 0.496 |
| | Avg | 0.306 | 0.347 | **0.296** | **0.346** | 0.360 | 0.390 | 0.498 | 0.494 | 0.319 | 0.368 | 0.457 | 0.586 | 0.479 | 0.603 | 0.502 | 0.651 | 0.353 | 0.440 |
| ETTh2 | 96 | **0.148** | **0.174** | 0.157 | 0.186 | 0.163 | 0.191 | 0.212 | 0.238 | 0.178 | 0.204 | 0.336 | 0.421 | 0.164 | 0.214 | 0.358 | 0.448 | 0.239 | 0.292 |
| | 192 | **0.166** | **0.193** | 0.183 | 0.213 | 0.195 | 0.218 | 0.229 | 0.250 | 0.193 | 0.217 | 0.265 | 0.339 | 0.226 | 0.294 | 0.457 | 0.575 | 0.313 | 0.376 |
| | 336 | **0.186** | **0.212** | 0.203 | 0.232 | 0.226 | 0.241 | 0.253 | 0.263 | 0.211 | 0.230 | 0.350 | 0.427 | 0.274 | 0.353 | 0.481 | 0.606 | 0.376 | 0.454 |
| | 720 | **0.208** | **0.234** | 0.203 | 0.233 | 0.255 | 0.263 | 0.279 | 0.273 | 0.222 | 0.238 | 0.406 | 0.482 | 0.302 | 0.382 | 0.445 | 0.550 | 0.990 | 1.092 |
| | Avg | **0.177** | **0.203** | 0.186 | 0.216 | 0.210 | 0.228 | 0.243 | 0.256 | 0.201 | 0.222 | 0.339 | 0.417 | 0.242 | 0.311 | 0.435 | 0.545 | 0.480 | 0.554 |
| Weather | 96 | **0.064** | **0.070** | 0.078 | 0.091 | 0.136 | 0.155 | 0.161 | 0.141 | 0.085 | 0.094 | 0.104 | 0.113 | 0.068 | 0.087 | 0.130 | 0.164 | 0.139 | 0.176 |
| | 192 | **0.069** | **0.075** | 0.083 | 0.097 | 0.145 | 0.165 | 0.173 | 0.139 | 0.094 | 0.102 | 0.134 | 0.144 | 0.068 | 0.086 | 0.127 | 0.158 | 0.143 | 0.166 |
| | 336 | **0.072** | **0.077** | 0.090 | 0.106 | 0.136 | 0.151 | 0.149 | 0.134 | 0.098 | 0.109 | 0.137 | 0.138 | 0.083 | 0.098 | 0.130 | 0.162 | 0.129 | 0.168 |
| | 720 | **0.076** | **0.081** | 0.096 | 0.113 | 0.151 | 0.161 | 0.233 | 0.155 | 0.107 | 0.120 | 0.152 | 0.141 | 0.087 | 0.102 | 0.113 | 0.136 | 0.122 | 0.149 |
| | Avg | **0.070** | **0.076** | 0.087 | 0.102 | 0.142 | 0.158 | 0.179 | 0.143 | 0.096 | 0.106 | 0.132 | 0.134 | 0.077 | 0.093 | 0.125 | 0.155 | 0.133 | 0.165 |
| Electricity | 96 | **0.066** | **0.081** | 0.068 | 0.083 | - | - | 0.231 | 0.275 | - | - | 0.344 | 0.441 | 0.153 | 0.203 | 0.096 | 0.119 | 0.083 | 0.108 |
| | 192 | 0.075 | 0.091 | **0.074** | **0.090** | - | - | 0.236 | 0.279 | - | - | 0.345 | 0.441 | 0.200 | 0.264 | 0.100 | 0.124 | 0.093 | 0.120 |
| | 336 | 0.087 | 0.106 | **0.083** | **0.100** | - | - | 0.245 | 0.289 | - | - | 0.462 | 0.571 | / | / | 0.102 | 0.126 | 0.095 | 0.122 |
| | 720 | 0.111 | 0.134 | **0.098** | **0.118** | - | - | 0.274 | 0.318 | - | - | 0.478 | 0.622 | / | / | 0.108 | 0.134 | 0.106 | 0.136 |
| | Avg | 0.085 | 0.103 | **0.081** | **0.098** | - | - | 0.247 | 0.290 | - | - | 0.407 | 0.519 | / | / | 0.102 | 0.126 | 0.094 | 0.122 |
| Traffic | 96 | **0.199** | **0.241** | - | - | 0.250 | 0.289 | - | - | 0.255 | 0.297 | 0.294 | 0.342 | / | / | 0.214 | 0.252 | 0.215 | 0.274 |
| | 192 | **0.207** | **0.250** | - | - | 0.249 | 0.278 | - | - | 0.295 | 0.343 | 0.306 | 0.354 | / | / | 0.223 | 0.259 | / | / |
| | 336 | **0.223** | **0.269** | - | - | 0.269 | 0.290 | - | - | 0.335 | 0.384 | 0.317 | 0.392 | / | / | 0.229 | 0.271 | / | / |
| | 720 | 0.250 | 0.289 | - | - | 0.310 | 0.321 | - | - | 0.434 | 0.517 | 0.391 | 0.478 | / | / | **0.233** | **0.274** | / | / |
| | Avg | **0.220** | **0.262** | - | - | 0.269 | 0.295 | - | - | 0.330 | 0.385 | 0.327 | 0.392 | / | / | 0.225 | 0.264 | / | / |
| Exchange | 96 | 0.024 | 0.026 | 0.024 | 0.027 | **0.021** | **0.025** | 0.025 | 0.030 | 0.042 | 0.051 | 0.079 | 0.090 | 0.028 | 0.036 | 0.068 | 0.079 | 0.026 | 0.033 |
| | 192 | **0.032** | **0.036** | 0.034 | 0.038 | 0.032 | 0.036 | 0.034 | 0.039 | 0.047 | 0.058 | 0.093 | 0.106 | 0.045 | 0.058 | 0.087 | 0.100 | 0.034 | 0.044 |
| | 336 | **0.045** | **0.048** | 0.046 | 0.050 | 0.045 | 0.048 | 0.047 | 0.052 | 0.061 | 0.073 | 0.081 | 0.106 | 0.060 | 0.076 | 0.074 | 0.086 | 0.058 | 0.074 |
| | 720 | 0.075 | **0.078** | 0.076 | 0.079 | 0.078 | 0.080 | **0.073** | 0.081 | 0.078 | 0.094 | 0.082 | 0.142 | 0.143 | 0.173 | 0.099 | 0.113 | 0.160 | 0.182 |
| | Avg | **0.044** | **0.047** | 0.045 | 0.049 | 0.044 | 0.047 | 0.045 | 0.050 | 0.057 | 0.069 | 0.084 | 0.111 | 0.069 | 0.086 | 0.082 | 0.095 | 0.070 | 0.083 |
| ILI | 24 | 0.110 | 0.124 | **0.108** | **0.123** | 0.120 | 0.139 | 0.150 | 0.196 | 0.135 | 0.173 | 0.228 | 0.242 | 0.250 | 0.263 | 0.275 | 0.296 | 0.231 | 0.275 |
| | 36 | 0.155 | 0.176 | **0.152** | **0.169** | 0.179 | 0.205 | 0.171 | 0.222 | 0.163 | 0.227 | 0.235 | 0.246 | 0.285 | 0.298 | 0.272 | 0.298 | 0.242 | 0.258 |
| | 48 | 0.168 | 0.186 | 0.164 | 0.184 | 0.186 | 0.215 | **0.151** | **0.184** | 0.171 | 0.233 | 0.265 | 0.275 | 0.285 | 0.301 | 0.295 | 0.320 | 0.280 | 0.303 |
| | 60 | **0.156** | **0.177** | 0.168 | 0.190 | 0.196 | 0.228 | 0.163 | 0.188 | 0.156 | 0.211 | 0.263 | 0.272 | 0.283 | 0.299 | 0.295 | 0.325 | 0.295 | 0.314 |
| | Avg | **0.147** | **0.166** | 0.148 | 0.166 | 0.170 | 0.197 | 0.159 | 0.197 | 0.156 | 0.211 | 0.248 | 0.259 | 0.276 | 0.290 | 0.284 | 0.310 | 0.262 | 0.288 |
| 1st Count | | **19** | **24** | 8 | 8 | 1 | 1 | 2 | 1 | 0 | 0 | 0 | 0 | 4 | 1 | 1 | 1 | 0 | 0 |

Table 15: Full results of zero-shot Aurora v.s full-shot end-to-end supervised models. Lower MSE or MAE values indicate better predictions. **Red**: the best, Blue: the 2nd best.

| Models | | Aurora (Ours) | | TimeKAN (2025) | | AMD (2025) | | TimePro (2025) | | TimeXer (2024) | | Fredformer (2024) | | iTransformer (2024) | | PatchTST (2023) | | TimesNet (2023) | |
|---|---|---|---|---|---|---|---|---|---|---|---|---|---|---|---|---|---|---|---|
| Metric | | MSE | MAE | MSE | MAE | MSE | MAE | MSE | MAE | MSE | MAE | MSE | MAE | MSE | MAE | MSE | MAE | MSE | MAE |
| ETTm1 | 96 | **0.294** | **0.351** | 0.327 | 0.365 | 0.327 | 0.361 | 0.326 | 0.364 | 0.309 | 0.352 | 0.326 | 0.361 | 0.334 | 0.368 | 0.329 | 0.367 | 0.338 | 0.375 |
| | 192 | **0.331** | **0.374** | 0.363 | 0.387 | 0.366 | 0.383 | 0.367 | 0.383 | 0.355 | 0.378 | 0.363 | 0.384 | 0.377 | 0.391 | 0.367 | 0.385 | 0.374 | 0.387 |
| | 336 | **0.359** | **0.391** | 0.389 | 0.407 | 0.398 | 0.404 | 0.402 | 0.409 | 0.387 | 0.399 | 0.395 | 0.406 | 0.426 | 0.420 | 0.399 | 0.410 | 0.410 | 0.411 |
| | 720 | **0.398** | **0.414** | 0.457 | 0.445 | 0.464 | 0.437 | 0.469 | 0.446 | 0.448 | 0.435 | 0.456 | 0.441 | 0.491 | 0.459 | 0.454 | 0.439 | 0.478 | 0.450 |
| | Avg | **0.346** | **0.383** | 0.384 | 0.401 | 0.389 | 0.396 | 0.391 | 0.401 | 0.375 | 0.391 | 0.385 | 0.398 | 0.407 | 0.410 | 0.387 | 0.400 | 0.400 | 0.406 |
| ETTm2 | 96 | 0.179 | 0.270 | 0.178 | 0.262 | 0.176 | 0.259 | 0.178 | 0.260 | **0.171** | **0.255** | 0.177 | 0.258 | 0.180 | 0.264 | 0.175 | 0.259 | 0.187 | 0.267 |
| | 192 | **0.232** | 0.307 | 0.244 | 0.308 | 0.242 | 0.302 | 0.242 | 0.303 | 0.238 | **0.300** | 0.243 | 0.301 | 0.250 | 0.309 | 0.241 | 0.302 | 0.249 | 0.309 |
| | 336 | **0.275** | **0.337** | 0.305 | 0.346 | 0.298 | 0.337 | 0.303 | 0.342 | 0.301 | 0.340 | 0.302 | 0.340 | 0.311 | 0.348 | 0.305 | 0.343 | 0.321 | 0.351 |
| | 720 | **0.338** | **0.380** | 0.402 | 0.404 | 0.396 | 0.394 | 0.400 | 0.399 | 0.401 | 0.397 | 0.404 | 0.398 | 0.412 | 0.407 | 0.402 | 0.400 | 0.408 | 0.403 |
| | Avg | **0.256** | 0.324 | 0.282 | 0.330 | 0.278 | **0.323** | 0.281 | 0.326 | 0.278 | 0.323 | 0.281 | 0.324 | 0.288 | 0.332 | 0.281 | 0.326 | 0.291 | 0.333 |
| ETTh1 | 96 | **0.340** | **0.381** | 0.374 | 0.391 | 0.375 | 0.392 | 0.375 | 0.398 | 0.377 | 0.397 | 0.378 | 0.395 | 0.386 | 0.405 | 0.414 | 0.419 | 0.384 | 0.402 |
| | 192 | **0.377** | **0.405** | 0.421 | 0.421 | 0.430 | 0.422 | 0.427 | 0.429 | 0.425 | 0.425 | 0.435 | 0.424 | 0.441 | 0.436 | 0.460 | 0.445 | 0.436 | 0.429 |
| | 336 | **0.399** | **0.422** | 0.464 | 0.440 | 0.471 | 0.443 | 0.472 | 0.450 | 0.457 | 0.441 | 0.485 | 0.447 | 0.487 | 0.458 | 0.501 | 0.466 | 0.491 | 0.469 |
| | 720 | **0.428** | **0.450** | 0.466 | 0.462 | 0.478 | 0.464 | 0.476 | 0.474 | 0.464 | 0.463 | 0.496 | 0.472 | 0.503 | 0.491 | 0.500 | 0.488 | 0.521 | 0.500 |
| | Avg | **0.386** | **0.415** | 0.431 | 0.429 | 0.438 | 0.430 | 0.438 | 0.438 | 0.431 | 0.432 | 0.448 | 0.435 | 0.454 | 0.448 | 0.469 | 0.455 | 0.458 | 0.450 |
| ETTh2 | 96 | **0.259** | **0.325** | 0.293 | 0.343 | 0.287 | 0.338 | 0.293 | 0.345 | 0.289 | 0.340 | 0.291 | 0.342 | 0.297 | 0.349 | 0.302 | 0.348 | 0.340 | 0.374 |
| | 192 | **0.324** | **0.370** | 0.375 | 0.396 | 0.367 | 0.388 | 0.367 | 0.394 | 0.370 | 0.391 | 0.372 | 0.390 | 0.380 | 0.400 | 0.388 | 0.400 | 0.402 | 0.414 |
| | 336 | **0.360** | **0.401** | 0.429 | 0.441 | 0.410 | 0.424 | 0.419 | 0.431 | 0.422 | 0.434 | 0.419 | 0.431 | 0.428 | 0.432 | 0.426 | 0.433 | 0.452 | 0.452 |
| | 720 | **0.403** | 0.441 | 0.466 | 0.468 | 0.421 | **0.440** | 0.427 | 0.445 | 0.429 | 0.445 | 0.431 | 0.450 | 0.427 | 0.445 | 0.431 | 0.446 | 0.462 | 0.468 |
| | Avg | **0.337** | **0.384** | 0.391 | 0.412 | 0.371 | 0.397 | 0.377 | 0.404 | 0.378 | 0.403 | 0.378 | 0.403 | 0.383 | 0.407 | 0.387 | 0.407 | 0.414 | 0.427 |
| Weather | 96 | **0.160** | **0.207** | 0.164 | 0.210 | 0.174 | 0.221 | 0.166 | 0.207 | 0.168 | 0.209 | 0.163 | 0.207 | 0.174 | 0.214 | 0.177 | 0.218 | 0.172 | 0.220 |
| | 192 | **0.202** | **0.247** | 0.209 | 0.250 | 0.219 | 0.259 | 0.216 | 0.254 | 0.220 | 0.254 | 0.224 | 0.258 | 0.221 | 0.254 | 0.255 | 0.259 | 0.219 | 0.261 |
| | 336 | **0.252** | **0.288** | 0.264 | 0.290 | 0.273 | 0.296 | 0.273 | 0.296 | 0.276 | 0.296 | 0.278 | 0.298 | 0.278 | 0.296 | 0.278 | 0.297 | 0.280 | 0.306 |
| | 720 | **0.307** | **0.327** | 0.343 | 0.342 | 0.349 | 0.345 | 0.351 | 0.346 | 0.353 | 0.347 | 0.357 | 0.350 | 0.358 | 0.349 | 0.354 | 0.348 | 0.365 | 0.359 |
| | Avg | **0.230** | **0.267** | 0.245 | 0.273 | 0.254 | 0.280 | 0.252 | 0.276 | 0.254 | 0.277 | 0.256 | 0.278 | 0.258 | 0.278 | 0.266 | 0.281 | 0.259 | 0.287 |
| Electricity | 96 | **0.134** | **0.234** | 0.174 | 0.266 | 0.147 | 0.251 | 0.139 | 0.234 | 0.151 | 0.247 | 0.148 | 0.242 | 0.148 | 0.240 | 0.195 | 0.285 | 0.168 | 0.272 |
| | 192 | 0.161 | 0.258 | 0.182 | 0.273 | 0.176 | 0.262 | **0.156** | **0.249** | 0.165 | 0.261 | 0.165 | 0.257 | 0.162 | 0.253 | 0.199 | 0.289 | 0.184 | 0.289 |
| | 336 | 0.193 | 0.287 | 0.197 | 0.286 | 0.193 | 0.281 | **0.172** | **0.267** | 0.183 | 0.280 | 0.180 | 0.274 | 0.180 | 0.274 | 0.178 | 0.269 | 0.198 | 0.300 |
| | 720 | 0.224 | 0.320 | 0.236 | 0.320 | 0.232 | 0.329 | **0.209** | **0.299** | 0.220 | 0.305 | 0.220 | 0.309 | 0.218 | 0.305 | 0.225 | 0.317 | 0.220 | 0.320 |
| | Avg | 0.178 | 0.275 | 0.197 | 0.286 | 0.187 | 0.281 | **0.169** | **0.262** | 0.180 | 0.274 | 0.178 | 0.270 | 0.178 | 0.270 | 0.216 | 0.304 | 0.193 | 0.295 |
| Traffic | 96 | 0.435 | 0.314 | 0.423 | 0.286 | 0.443 | 0.298 | 0.426 | 0.292 | 0.416 | 0.280 | 0.403 | 0.274 | **0.395** | **0.268** | 0.544 | 0.359 | 0.593 | 0.321 |
| | 192 | 0.465 | 0.328 | 0.442 | 0.295 | 0.496 | 0.323 | 0.439 | 0.298 | 0.435 | 0.288 | 0.429 | 0.289 | **0.417** | **0.276** | 0.540 | 0.354 | 0.617 | 0.336 |
| | 336 | 0.525 | 0.355 | 0.473 | 0.335 | 0.520 | 0.330 | 0.449 | 0.307 | 0.451 | 0.296 | 0.441 | 0.295 | **0.433** | **0.283** | 0.551 | 0.358 | 0.629 | 0.336 |
| | 720 | 0.670 | 0.411 | 0.481 | 0.357 | 0.540 | 0.344 | 0.475 | 0.309 | 0.484 | 0.314 | **0.463** | **0.300** | 0.467 | 0.302 | 0.586 | 0.375 | 0.640 | 0.350 |
| | Avg | 0.524 | 0.352 | 0.455 | 0.318 | 0.500 | 0.324 | 0.447 | 0.302 | 0.447 | 0.295 | 0.434 | 0.289 | **0.428** | **0.282** | 0.555 | 0.362 | 0.620 | 0.336 |
| 1st Count | | **20** | **18** | 0 | 0 | 0 | 1 | 3 | 3 | 1 | 2 | 1 | 1 | 3 | 3 | 0 | 0 | 0 | 0 |

Table 16: Studies on Aurora (zero-shot) and Aurora (10% few shot) v.s few-shot (10% and 20%) end-to-end small models, i.e., PatchTST, iTransformer, TimesNet, DLinear. **Red**: the best, Blue: the 2nd best.

| Models | | Aurora (Zero) | | Aurora (10%) | | PatchTST (10%) | | iTransformer (10%) | | TimesNet (10%) | | DLinear (10%) | | PatchTST (20%) | | iTransformer (20%) | | TimesNet (20%) | | DLinear (20%) | |
|---|---|---|---|---|---|---|---|---|---|---|---|---|---|---|---|---|---|---|---|---|---|
| Metric | | MSE | MAE | MSE | MAE | MSE | MAE | MSE | MAE | MSE | MAE | MSE | MAE | MSE | MAE | MSE | MAE | MSE | MAE | MSE | MAE |
| Agriculture | 6 | 0.184 | 0.295 | 0.127 | 0.233 | 3.738 | 1.238 | 4.908 | 1.396 | 6.681 | 1.622 | 8.882 | 2.053 | 0.914 | 0.700 | 1.238 | 0.723 | 1.363 | 0.789 | 3.732 | 1.442 |
| | 8 | 0.242 | 0.335 | 0.190 | 0.289 | 4.836 | 1.389 | 7.042 | 1.651 | 4.984 | 1.514 | 9.940 | 2.045 | 1.484 | 0.773 | 1.428 | 0.773 | 1.354 | 0.754 | 4.484 | 1.521 |
| | 10 | 0.297 | 0.365 | 0.236 | 0.310 | 6.824 | 1.645 | 7.406 | 1.718 | 7.026 | 1.684 | 11.069 | 2.199 | 1.781 | 0.830 | 1.822 | 0.871 | 2.226 | 0.930 | 4.600 | 1.845 |
| | 12 | 0.365 | 0.398 | 0.295 | 0.340 | 7.773 | 1.778 | 9.151 | 1.938 | 11.685 | 2.208 | 12.634 | 2.329 | 2.048 | 0.912 | 2.087 | 0.921 | 2.606 | 1.050 | 6.037 | 1.721 |
| | Avg | 0.272 | 0.348 | 0.212 | 0.293 | 5.793 | 1.512 | 7.127 | 1.676 | 7.594 | 1.757 | 10.631 | 2.157 | 1.557 | 0.804 | 1.644 | 0.822 | 1.887 | 0.881 | 4.713 | 1.632 |
| Climate | 6 | 0.859 | 0.747 | 0.857 | 0.744 | 1.063 | 0.844 | 1.153 | 0.890 | 1.409 | 0.956 | 1.161 | 0.881 | 1.515 | 1.005 | 1.470 | 0.990 | 1.573 | 1.021 | 1.445 | 0.983 |
| | 8 | 0.858 | 0.746 | 0.858 | 0.745 | 1.028 | 0.829 | 1.176 | 0.886 | 1.128 | 0.872 | 1.163 | 0.882 | 1.532 | 1.021 | 1.437 | 0.978 | 1.604 | 1.034 | 1.425 | 0.977 |
| | 10 | 0.868 | 0.748 | 0.863 | 0.744 | 1.021 | 0.820 | 1.221 | 0.905 | 1.208 | 0.895 | 1.129 | 0.866 | 1.615 | 1.035 | 1.546 | 1.008 | 1.479 | 0.996 | 1.387 | 0.959 |
| | 12 | 0.875 | 0.753 | 0.869 | 0.749 | 1.020 | 0.819 | 1.477 | 0.977 | 1.606 | 1.029 | 1.148 | 0.870 | 1.505 | 0.991 | 1.669 | 1.043 | 1.439 | 0.984 | 1.412 | 0.964 |
| | Avg | 0.865 | 0.749 | 0.862 | 0.746 | 1.033 | 0.828 | 1.257 | 0.915 | 1.338 | 0.938 | 1.150 | 0.875 | 1.542 | 1.013 | 1.531 | 1.005 | 1.524 | 1.009 | 1.417 | 0.971 |
| Economy | 6 | 0.035 | 0.150 | 0.015 | 0.095 | 0.224 | 0.384 | 0.159 | 0.320 | 0.258 | 0.405 | 0.750 | 0.715 | 0.035 | 0.150 | 0.209 | 0.137 | 0.045 | 0.175 | 0.174 | 0.330 |
| | 8 | 0.033 | 0.145 | 0.015 | 0.099 | 0.232 | 0.389 | 0.183 | 0.335 | 0.256 | 0.427 | 0.417 | 0.523 | 0.041 | 0.161 | 0.033 | 0.144 | 0.049 | 0.185 | 0.136 | 0.307 |
| | 10 | 0.032 | 0.143 | 0.016 | 0.101 | 0.224 | 0.382 | 0.194 | 0.350 | 0.359 | 0.506 | 0.426 | 0.531 | 0.031 | 0.142 | 0.034 | 0.148 | 0.048 | 0.174 | 0.148 | 0.316 |
| | 12 | 0.032 | 0.144 | 0.016 | 0.102 | 0.247 | 0.405 | 0.179 | 0.349 | 0.238 | 0.410 | 0.499 | 0.585 | 0.037 | 0.155 | 0.040 | 0.164 | 0.045 | 0.174 | 0.172 | 0.349 |
| | Avg | 0.033 | 0.146 | 0.016 | 0.099 | 0.232 | 0.390 | 0.179 | 0.339 | 0.278 | 0.437 | 0.523 | 0.589 | 0.036 | 0.152 | 0.079 | 0.148 | 0.047 | 0.177 | 0.158 | 0.326 |
| Energy | 12 | 0.117 | 0.245 | 0.097 | 0.212 | 0.741 | 0.648 | 0.855 | 0.738 | 1.458 | 0.919 | 0.605 | 0.593 | 1.147 | 0.819 | 0.583 | 0.603 | 0.680 | 0.622 | 0.700 | 0.644 |
| | 24 | 0.226 | 0.354 | 0.199 | 0.322 | 1.381 | 0.906 | 1.944 | 1.010 | 3.484 | 1.537 | 1.013 | 0.773 | 1.223 | 0.843 | 1.399 | 0.878 | 1.127 | 0.791 | 1.228 | 0.852 |
| | 36 | 0.292 | 0.409 | 0.271 | 0.352 | 1.648 | 0.980 | 3.012 | 1.322 | 4.406 | 1.563 | 1.472 | 0.954 | 1.749 | 1.058 | 1.654 | 0.967 | 1.790 | 1.056 | 1.541 | 0.957 |
| | 48 | 0.383 | 0.472 | 0.352 | 0.431 | 1.864 | 1.038 | 4.851 | 1.668 | 7.460 | 2.164 | 1.846 | 1.110 | 2.099 | 1.053 | 2.097 | 1.084 | 2.015 | 1.168 | 1.637 | 0.992 |
| | Avg | 0.255 | 0.370 | 0.230 | 0.329 | 1.408 | 0.893 | 2.666 | 1.185 | 4.202 | 1.546 | 1.234 | 0.858 | 1.555 | 0.943 | 1.433 | 0.883 | 1.403 | 0.909 | 1.277 | 0.861 |
| Environment | 48 | 0.281 | 0.380 | 0.269 | 0.372 | 0.638 | 0.567 | 0.538 | 0.498 | 0.541 | 0.502 | 0.782 | 0.718 | 0.437 | 0.458 | 0.441 | 0.468 | 0.487 | 0.472 | 0.752 | 0.682 |
| | 96 | 0.284 | 0.382 | 0.271 | 0.373 | 0.682 | 0.560 | 0.551 | 0.518 | 0.548 | 0.503 | 0.802 | 0.732 | 0.483 | 0.495 | 0.462 | 0.482 | 0.504 | 0.486 | 0.782 | 0.703 |
| | 192 | 0.270 | 0.375 | 0.269 | 0.374 | 0.664 | 0.567 | 0.567 | 0.528 | 0.571 | 0.519 | 0.828 | 0.752 | 0.503 | 0.506 | 0.488 | 0.492 | 0.518 | 0.497 | 0.812 | 0.743 |
| | 336 | 0.269 | 0.377 | 0.251 | 0.368 | 0.623 | 0.561 | 0.591 | 0.536 | 0.597 | 0.529 | 0.834 | 0.767 | 0.516 | 0.518 | 0.512 | 0.510 | 0.530 | 0.511 | 0.858 | 0.777 |
| | Avg | 0.276 | 0.379 | 0.265 | 0.372 | 0.652 | 0.564 | 0.562 | 0.520 | 0.564 | 0.513 | 0.812 | 0.742 | 0.485 | 0.494 | 0.476 | 0.488 | 0.510 | 0.492 | 0.801 | 0.726 |
| Health | 12 | 1.093 | 0.668 | 0.992 | 0.641 | 2.419 | 1.084 | 2.121 | 1.052 | 2.455 | 1.060 | 2.172 | 1.000 | 2.139 | 1.068 | 1.969 | 0.965 | 1.889 | 0.977 | 2.319 | 1.115 |
| | 24 | 1.572 | 0.849 | 1.332 | 0.796 | 2.432 | 1.076 | 2.400 | 1.055 | 3.306 | 1.258 | 2.288 | 0.986 | 2.292 | 1.075 | 2.467 | 1.090 | 2.603 | 1.120 | 2.547 | 1.184 |
| | 36 | 1.688 | 0.920 | 1.467 | 0.818 | 3.117 | 1.240 | 2.748 | 1.178 | 3.535 | 1.435 | 2.416 | 1.022 | 2.521 | 1.091 | 2.449 | 1.073 | 2.600 | 1.091 | 2.411 | 1.125 |
| | 48 | 1.857 | 0.963 | 1.579 | 0.847 | 3.158 | 1.268 | 3.112 | 1.200 | 3.274 | 1.281 | 2.737 | 1.132 | 2.727 | 1.050 | 2.569 | 1.150 | 3.026 | 1.677 | 2.386 | 1.088 |
| | Avg | 1.553 | 0.850 | 1.343 | 0.776 | 2.781 | 1.167 | 2.595 | 1.121 | 3.143 | 1.259 | 2.403 | 1.035 | 2.420 | 1.071 | 2.364 | 1.070 | 2.530 | 1.216 | 2.416 | 1.128 |
| Security | 6 | 67.572 | 3.909 | 64.513 | 3.798 | 77.436 | 4.474 | 69.331 | 4.075 | 76.943 | 5.200 | 78.251 | 5.482 | 77.295 | 4.462 | 74.295 | 4.371 | 75.285 | 5.023 | 76.051 | 4.982 |
| | 8 | 70.576 | 4.013 | 67.828 | 3.930 | 84.319 | 4.735 | 70.328 | 4.582 | 78.820 | 5.320 | 80.295 | 5.692 | 82.258 | 4.629 | 76.285 | 4.592 | 77.256 | 5.156 | 79.295 | 5.292 |
| | 10 | 74.173 | 4.148 | 72.423 | 4.092 | 90.961 | 5.142 | 83.843 | 4.469 | 84.928 | 5.721 | 86.285 | 6.025 | 84.256 | 4.824 | 78.295 | 4.825 | 78.296 | 5.332 | 84.285 | 6.382 |
| | 12 | 77.579 | 4.264 | 75.482 | 4.132 | 89.994 | 5.079 | 84.069 | 4.587 | 88.295 | 5.829 | 89.925 | 6.325 | 86.925 | 4.925 | 81.256 | 5.025 | 80.285 | 5.425 | 87.290 | 6.478 |
| | Avg | 72.475 | 4.084 | 70.062 | 3.988 | 85.677 | 4.858 | 76.893 | 4.428 | 82.246 | 5.518 | 83.689 | 5.881 | 82.684 | 4.710 | 77.533 | 4.703 | 77.781 | 5.234 | 81.730 | 5.784 |
| Social Good | 6 | 0.701 | 0.442 | 0.689 | 0.427 | 10.213 | 1.301 | 5.302 | 0.980 | 5.159 | 1.074 | 5.000 | 1.037 | 1.925 | 0.647 | 2.194 | 0.672 | 1.973 | 0.586 | 1.985 | 0.654 |
| | 8 | 0.804 | 0.493 | 0.784 | 0.461 | 11.077 | 1.435 | 6.140 | 1.121 | 5.908 | 1.085 | 5.458 | 1.172 | 2.133 | 0.659 | 2.550 | 0.747 | 2.076 | 0.690 | 2.273 | 0.780 |
| | 10 | 0.886 | 0.543 | 0.850 | 0.532 | 11.515 | 1.476 | 6.810 | 1.181 | 7.254 | 1.228 | 5.858 | 1.254 | 2.894 | 0.775 | 2.695 | 0.733 | 2.331 | 0.780 | 2.389 | 0.823 |
| | 12 | 0.960 | 0.587 | 0.931 | 0.554 | 13.642 | 1.788 | 7.715 | 1.284 | 7.816 | 1.332 | 6.206 | 1.332 | 2.539 | 0.759 | 3.033 | 0.795 | 2.427 | 0.811 | 2.539 | 0.876 |
| | Avg | 0.838 | 0.516 | 0.814 | 0.494 | 11.612 | 1.500 | 6.492 | 1.142 | 6.534 | 1.180 | 5.631 | 1.199 | 2.373 | 0.710 | 2.618 | 0.737 | 2.202 | 0.717 | 2.297 | 0.783 |
| Traffic | 6 | 0.154 | 0.285 | 0.149 | 0.292 | 2.391 | 1.096 | 2.317 | 1.096 | 2.486 | 1.126 | 2.790 | 1.318 | 2.134 | 1.004 | 1.723 | 0.829 | 1.798 | 0.889 | 2.344 | 1.205 |
| | 8 | 0.158 | 0.286 | 0.155 | 0.284 | 2.489 | 1.093 | 2.166 | 1.004 | 2.921 | 1.334 | 2.614 | 1.234 | 1.924 | 0.922 | 1.734 | 0.816 | 2.420 | 1.203 | 2.148 | 1.119 |
| | 10 | 0.163 | 0.289 | 0.160 | 0.287 | 2.907 | 1.180 | 2.143 | 0.998 | 2.779 | 1.308 | 2.536 | 1.209 | 2.149 | 1.028 | 1.943 | 0.894 | 2.143 | 0.935 | 2.121 | 1.104 |
| | 12 | 0.168 | 0.294 | 0.165 | 0.296 | 2.664 | 1.115 | 2.539 | 1.030 | 3.099 | 1.239 | 2.627 | 1.244 | 1.975 | 0.955 | 1.819 | 0.843 | 2.230 | 1.082 | 2.176 | 1.129 |
| | Avg | 0.161 | 0.289 | 0.157 | 0.290 | 2.613 | 1.121 | 2.291 | 1.032 | 2.821 | 1.252 | 2.642 | 1.251 | 2.046 | 0.977 | 1.805 | 0.846 | 2.148 | 1.027 | 2.197 | 1.139 |

# C SHOWCASES

## C.1 SHOWCASES OF DATASETS WITH SIMILAR HISTORIES BUT DISTINCT FUTURES

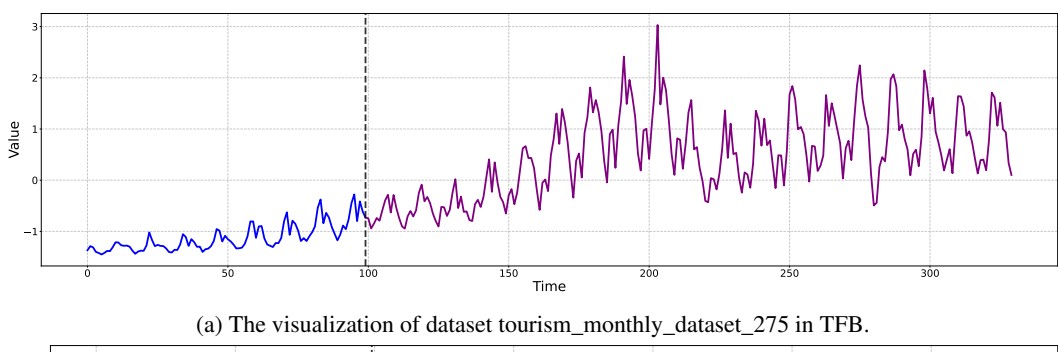

(a) The visualization of dataset tourism_monthly_dataset_275 in TFB.

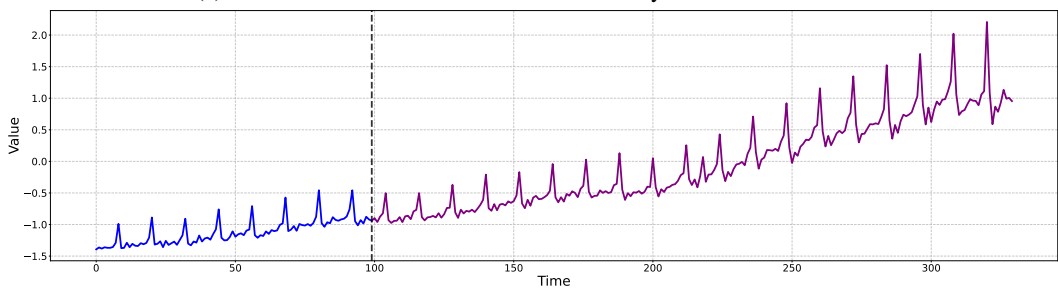

(b) The visualization of dataset m4_monthly_dataset_14569 in TFB.

Figure 10: Visual comparisons between datasets tourism_monthly_dataset_275 and m4_monthly_dataset_14569 from distinct domains. Blue part indicates the historical similar time series, and purple part indicates the distinct future horizons.

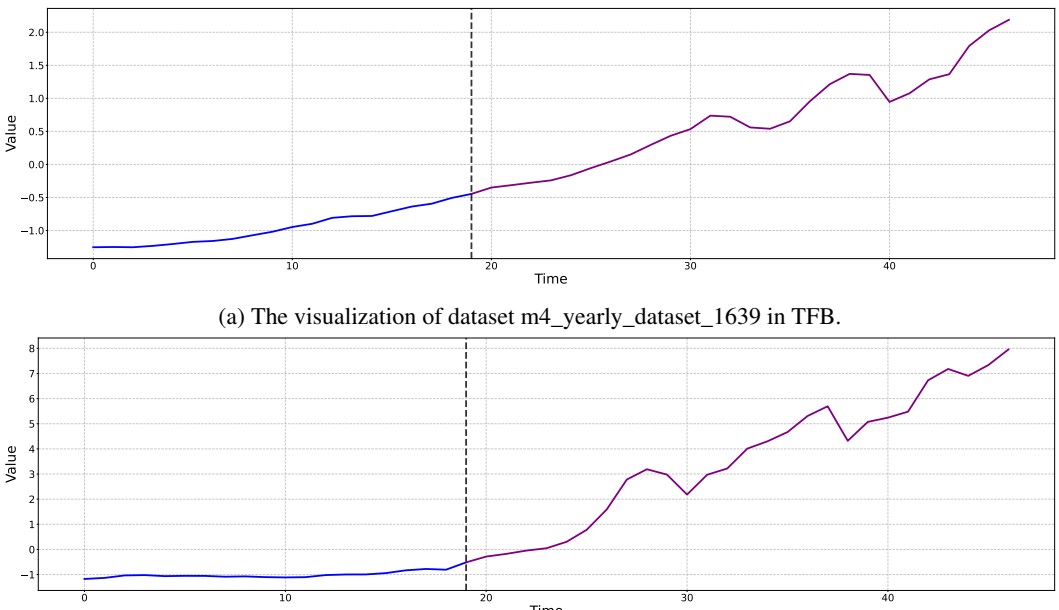

(a) The visualization of dataset m4_yearly_dataset_1639 in TFB.

(b) The visualization of dataset finance_87 in TFB.

Figure 11: Visual comparisons between datasets m4_yearly_dataset_1639 and finance_87 from distinct domains. Blue part indicates the historical similar time series, and purple part indicates the distinct future horizons.

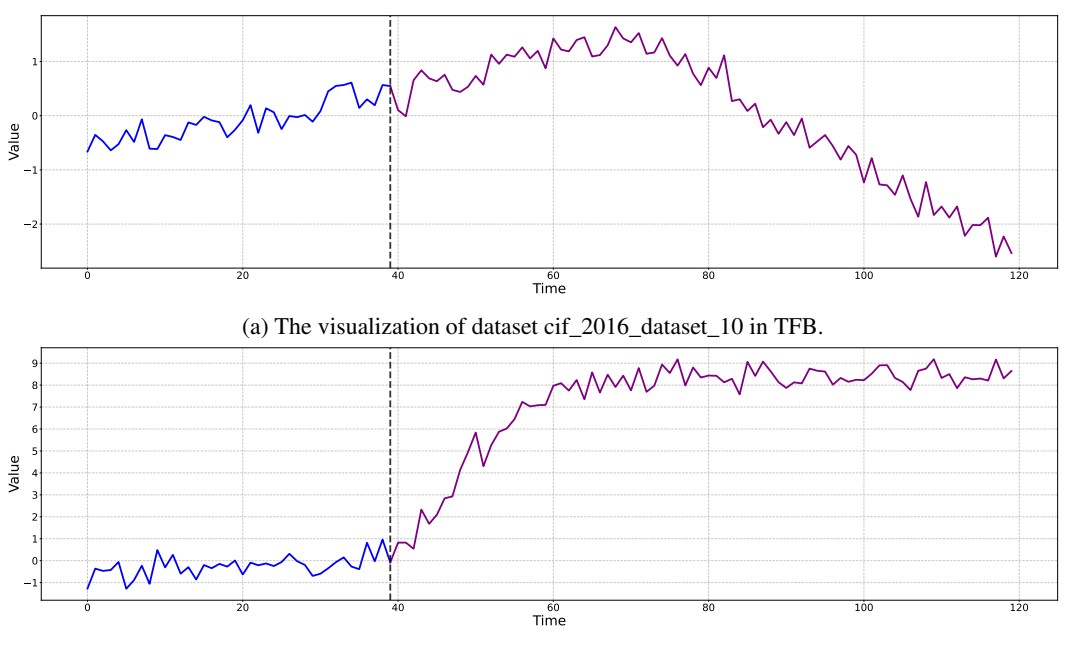

(a) The visualization of dataset cif_2016_dataset_10 in TFB.

(b) The visualization of dataset cif_2016_dataset_8 in TFB.

Figure 12: Visual comparisons between datasets cif_2016_dataset_10 and cif_2016_dataset_8 from the same domains. Blue part indicates the historical similar time series, and purple part indicates the distinct future horizons.

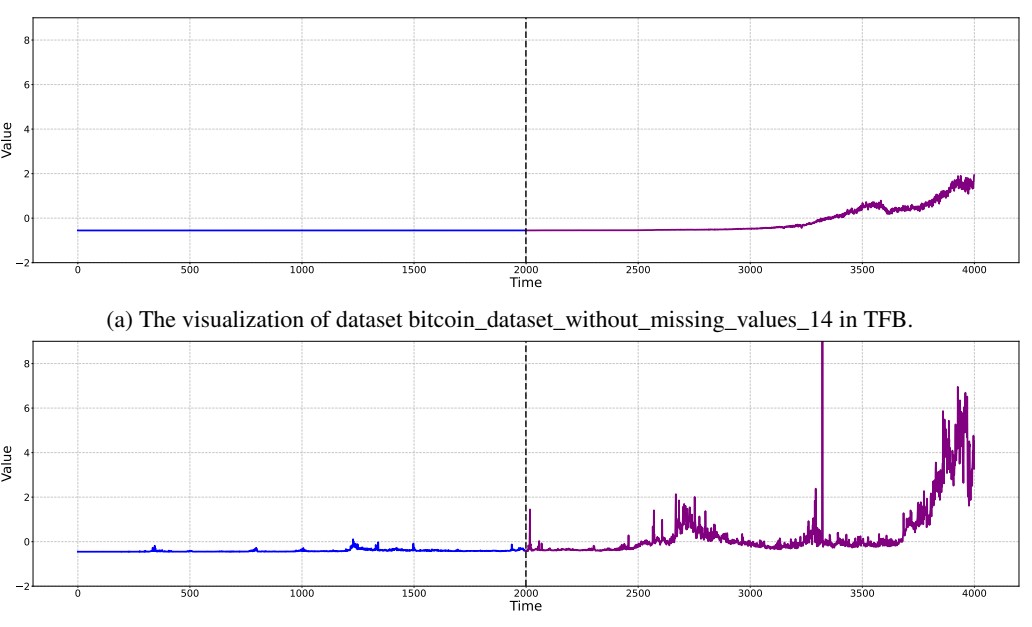

(a) The visualization of dataset bitcoin_dataset_without_missing_values_14 in TFB.

(b) The visualization of dataset bitcoin_dataset_without_missing_values_12 in TFB.

Figure 13: Visual comparisons between datasets bitcoin_dataset_without_missing_values_14 and bitcoin_dataset_without_missing_values_12 from the same domain. Blue part indicates the historical similar time series, and purple part indicates the distinct future horizons.

## C.2 VISUALIZATION OF THE PROTOTYPEBANK

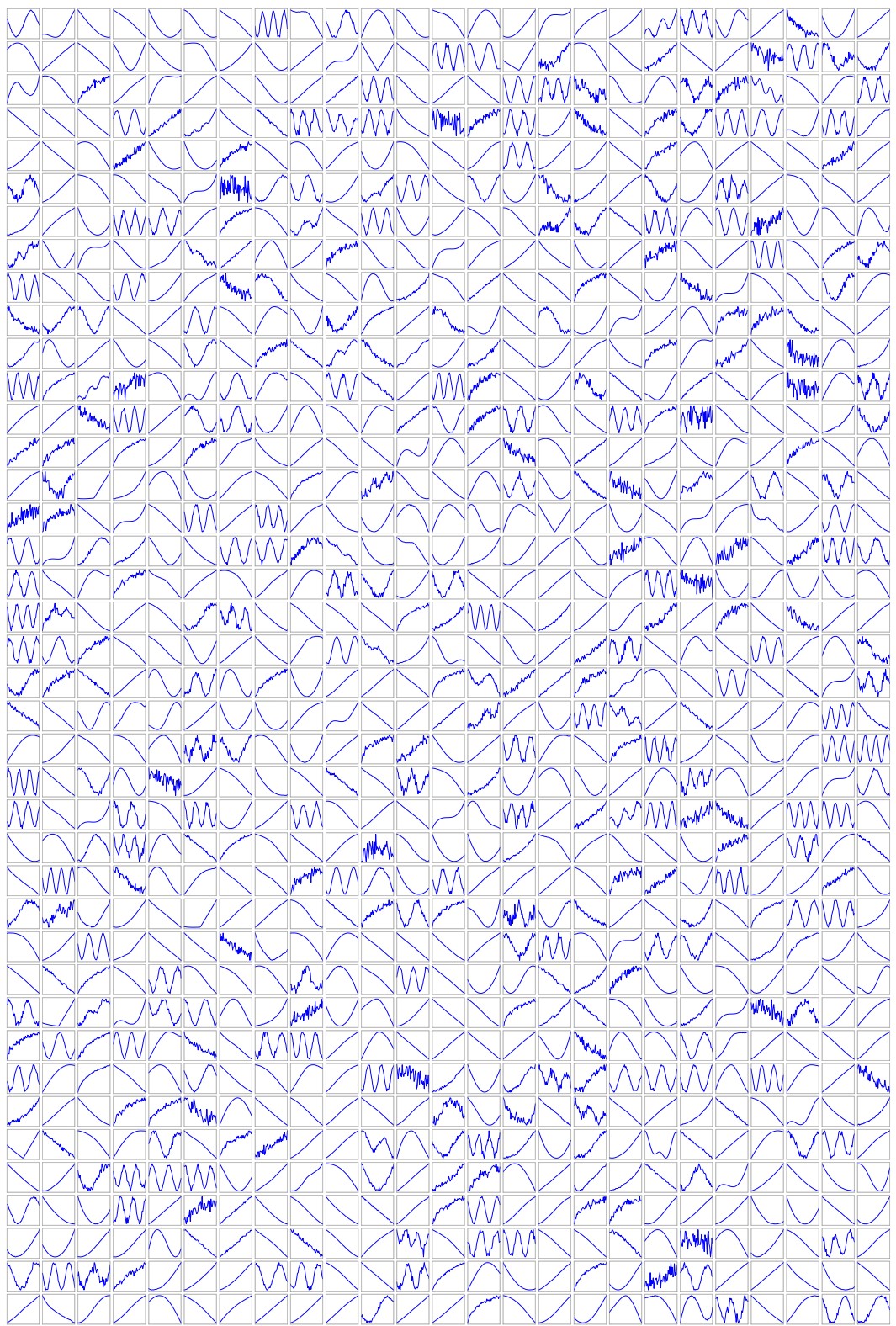

Figure 14: The visualization of all 1,000 prototypes in PrototypeBank. Note that though some prototypes may look similar due to drawing, they actually differ in their magnitutdes and phases.

## C.3 VISUALIZATION OF GENERATED PROTOTYPES FOR PREDICTIONS

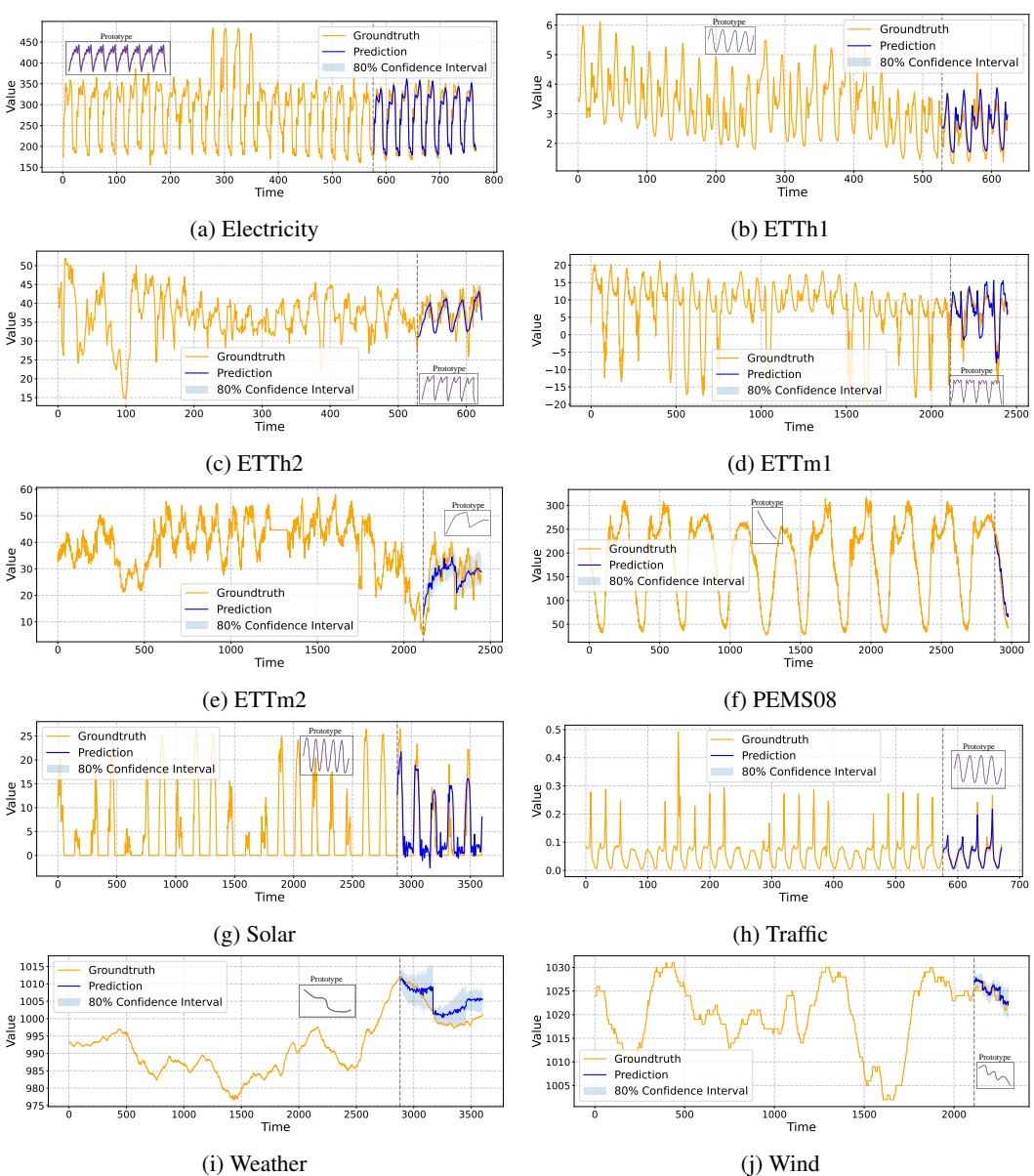

Figure 15: Visualization of TSFM-Bench

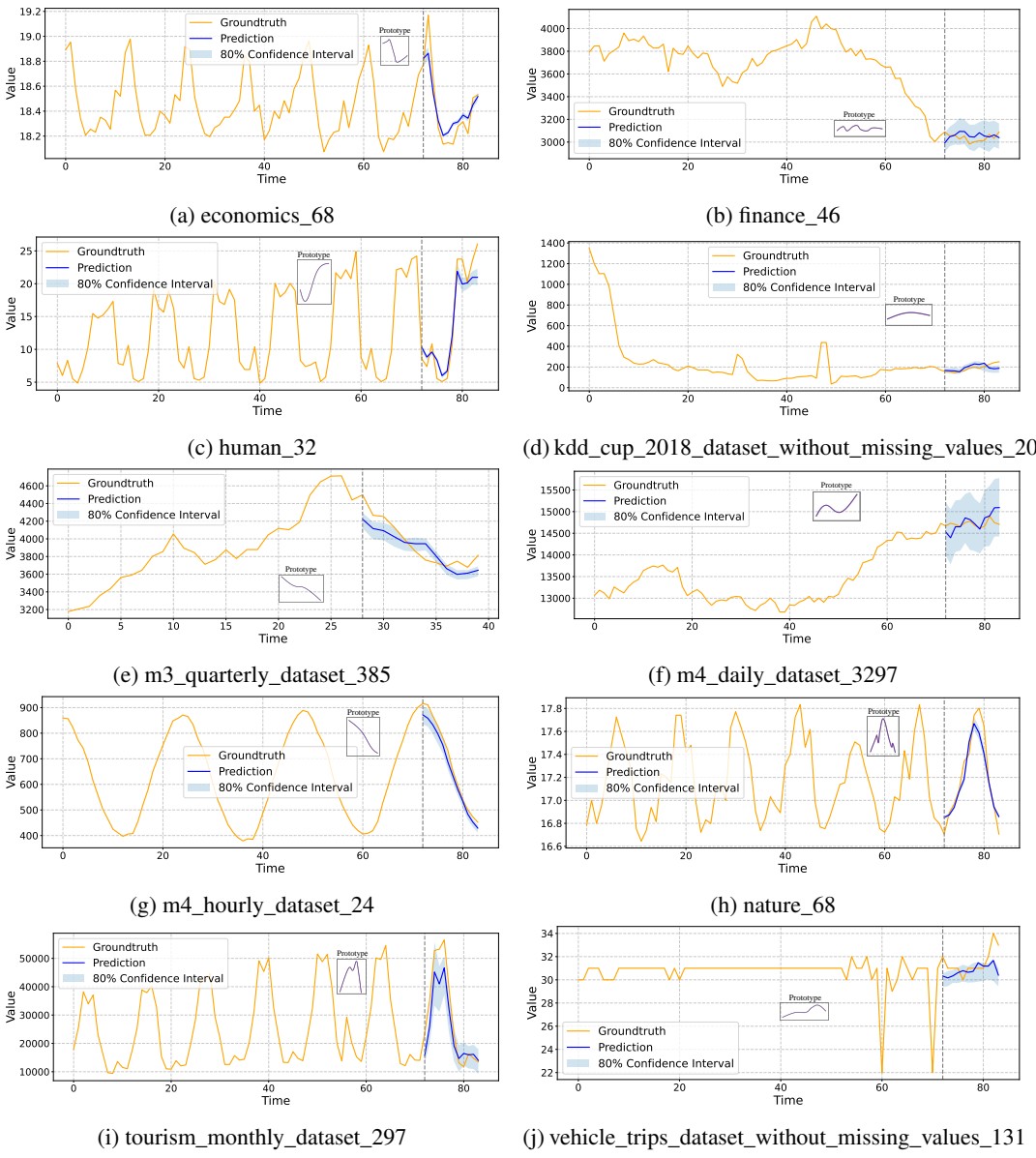

Figure 16: Visualization of Univariate Datasets in TFB

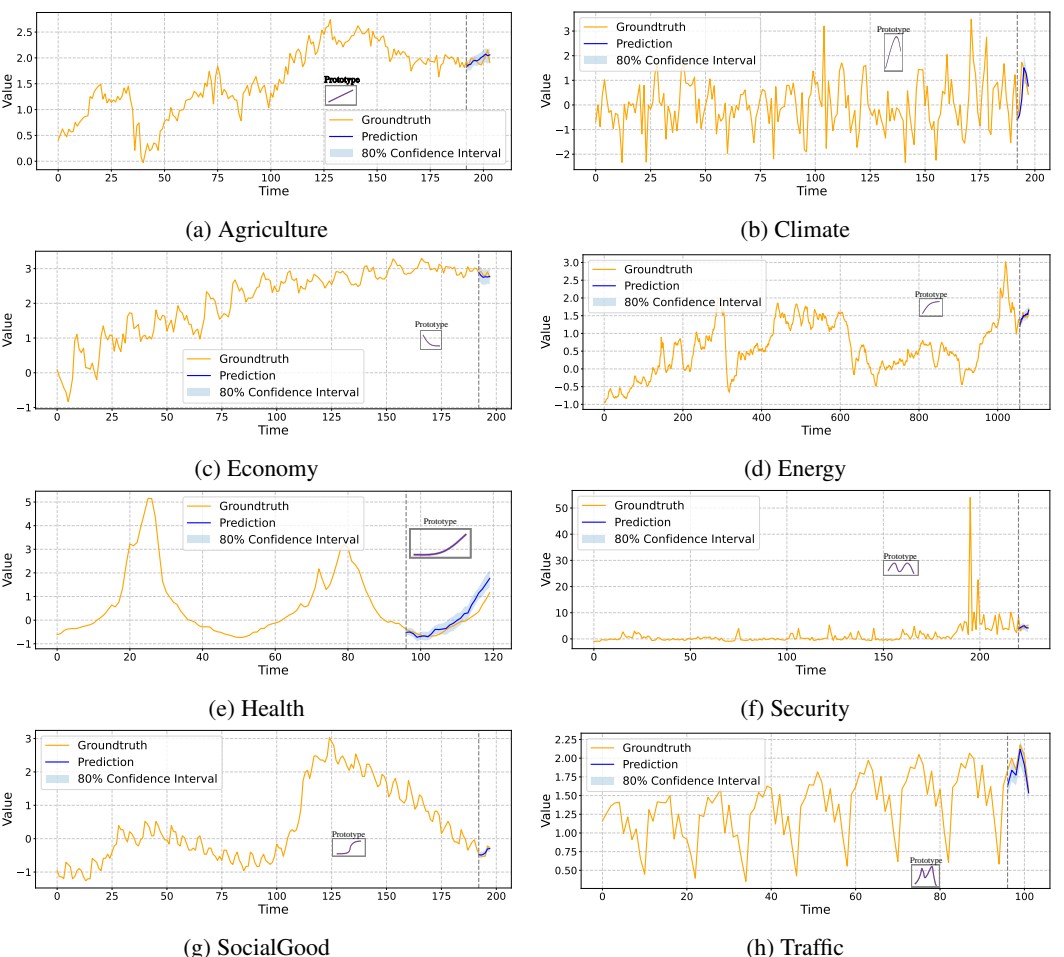

Figure 17: Visualization of TimeMMD

## C.4 VISUALIZATION OF MODALITY-GUIDED ATTENTION WEIGHTS

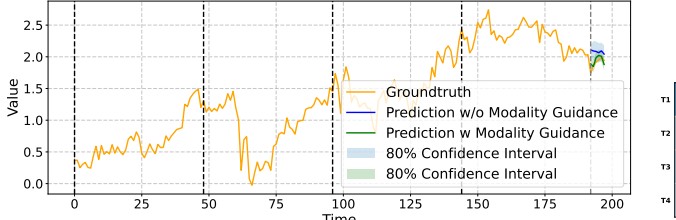 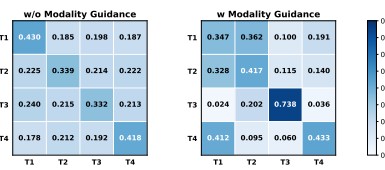

(a) Predictions on Agriculture, without Modality Guidance v.s with Modality Guidance. With Patch Size equals 48 (Separated by black dashed lines in the figure).

(b) The visualization of attention weights, without Modality Guidance v.s with Modality Guidance.

Figure 18: It is observed that the predictions with modality guidance are more accurate. The 4 patches (T1 – T4) of the contextual time series show similar correlations without modality guidance. While with the modality guidance, the correlations between T1 and T2 are further focused on, because their correlations are similar to the T4 and future values, with a trend of first decreasing and then increasing. Additionally, the correlations between T4 and T1 are also focused on for prediction.

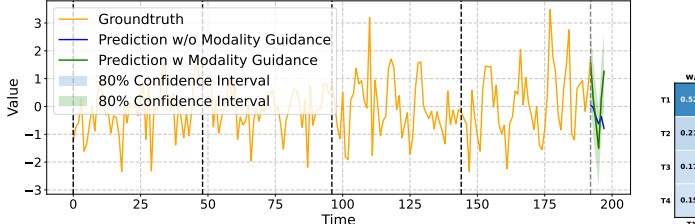 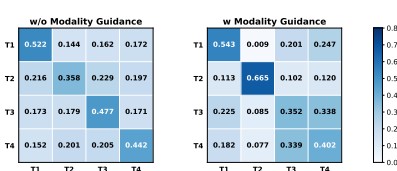

(a) Predictions on Climate, without Modality Guidance v.s with Modality Guidance. With Patch Size equals 48 (Separated by black dashed lines in the figure).

(b) The visualization of attention weights, without Modality Guidance v.s with Modality Guidance.

Figure 19: It is observed that the predictions with modality guidance are more accurate. The 4 patches (T1 – T4) of the contextual time series show similar correlations without modality guidance. While with the modality guidance, the correlations between T3 and T4 are further focused on, because their correlations are similar to the T4 and future values, simply copying the trend can lead to higher accuracy.

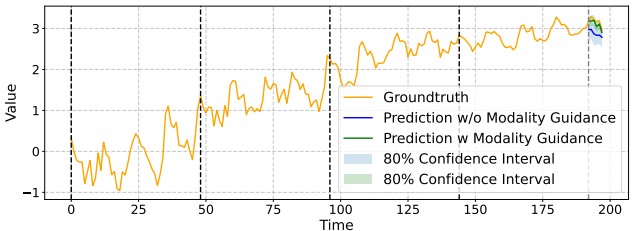 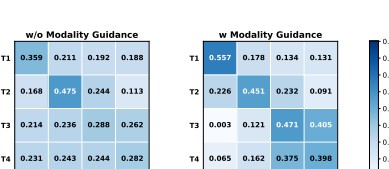

(a) Predictions on Economy, without Modality Guidance v.s with Modality Guidance. With Patch Size equals 48 (Separated by black dashed lines in the figure).

(b) The visualization of attention weights, without Modality Guidance v.s with Modality Guidance.

Figure 20: It is observed that the predictions with modality guidance are more accurate. The 4 patches (T1 – T4) of the contextual time series show similar correlations without modality guidance. While with the modality guidance, the correlations between T3 and T4 are further focused on, because their correlations are similar to the T4 and future values, simply copying the trend can lead to higher accuracy.

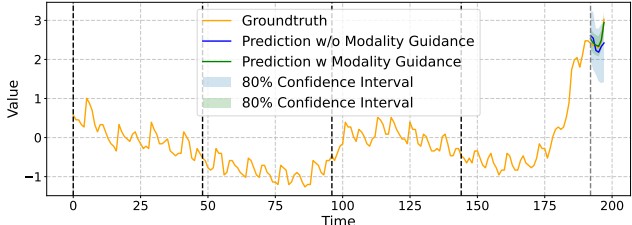 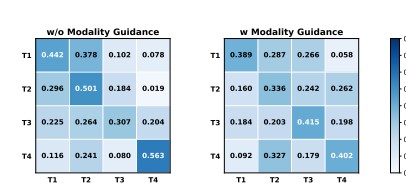

(a) Predictions on SocialGood, without Modality Guidance v.s with Modality Guidance. With Patch Size equals 48 (Separated by black dashed lines in the figure).

(b) The visualization of attention weights, without Modality Guidance v.s with Modality Guidance.

Figure 21: It is observed that the predictions with modality guidance are more accurate. The 3 patches (T1 – T3) of the contextual time series show similar correlations without modality guidance, and are highly dissimiliar to T4. While with the modality guidance, the correlations between T4 and T2 are further focused on, because their trends share potential similarity, first decreasing then increasing. And the correlations between T2 and T3 are also focused, which are similar to the correlations between T4 and future values.

# D   MORE MODEL ANALYTICS

Table 17: The studies on different sizes of the PrototypeBank.

| Models | 500 | | 1000 | | 5000 | |
|---|---|---|---|---|---|---|
| Metrics | MSE | MAE | MSE | MAE | MSE | MAE |
| Agriculture | 0.288 | 0.364 | 0.272 | 0.348 | **0.269** | **0.344** |
| Climate | 0.893 | 0.760 | 0.865 | **0.749** | **0.863** | 0.752 |
| Economy | 0.034 | 0.152 | 0.033 | 0.146 | **0.032** | **0.144** |
| Energy | 0.271 | 0.391 | **0.255** | **0.370** | 0.260 | 0.373 |
| Environment | 0.302 | 0.413 | 0.276 | 0.379 | **0.268** | **0.372** |
| Health | 1.587 | 0.866 | **1.553** | **0.850** | 1.561 | 0.854 |
| Security | 75.017 | 4.118 | 72.475 | 4.084 | **71.551** | **4.036** |
| Social Good | 0.862 | 0.547 | 0.838 | 0.516 | **0.833** | **0.507** |
| Traffic | 0.187 | 0.304 | **0.161** | **0.289** | 0.164 | 0.296 |

Table 18: The ablation studies on modalities.

| Models | Aurora | | Time-Only | | Time + Text | | Time + Image | |
|---|---|---|---|---|---|---|---|---|
| Metrics | MSE | MAE | MSE | MAE | MSE | MAE | MSE | MAE |
| Agriculture | **0.272** | **0.348** | 0.337 | 0.382 | 0.304 | 0.355 | 0.294 | 0.351 |
| Climate | **0.865** | **0.749** | 1.287 | 0.926 | 1.167 | 0.904 | 1.228 | 0.897 |
| Economy | **0.033** | **0.146** | 0.064 | 0.198 | 0.046 | 0.167 | 0.039 | 0.152 |
| Energy | **0.255** | **0.370** | 0.324 | 0.426 | 0.292 | 0.413 | 0.285 | 0.406 |
| Environment | **0.276** | **0.379** | 0.352 | 0.404 | 0.334 | 0.397 | 0.325 | 0.394 |
| Health | **1.553** | **0.850** | 2.305 | 1.147 | 1.962 | 0.987 | 1.874 | 0.972 |
| Security | **72.475** | **4.084** | 92.822 | 5.092 | 81.294 | 4.800 | 77.928 | 4.628 |
| Social Good | **0.838** | **0.516** | 1.387 | 0.692 | 1.018 | 0.576 | 1.037 | 0.572 |
| Traffic | **0.161** | **0.289** | 0.345 | 0.472 | 0.271 | 0.418 | 0.198 | 0.334 |

Table 19: Ablation studies on Modality-Guided Attention.

| Models | Aurora | | Text-Guidance | | Image-Guidance | | w/o W | |
|---|---|---|---|---|---|---|---|---|
| Metrics | MSE | MAE | MSE | MAE | MSE | MAE | MSE | MAE |
| Agriculture | **0.272** | **0.348** | 0.287 | 0.350 | 0.279 | 0.353 | 0.274 | 0.351 |
| Climate | **0.865** | **0.749** | 0.885 | 0.767 | 0.898 | 0.785 | 0.876 | 0.755 |
| Economy | **0.033** | **0.146** | 0.040 | 0.157 | 0.038 | 0.154 | 0.034 | 0.148 |
| Energy | **0.255** | **0.370** | 0.274 | 0.387 | 0.262 | 0.374 | 0.265 | 0.376 |
| Environment | **0.276** | **0.379** | 0.294 | 0.386 | 0.285 | 0.389 | 0.280 | 0.381 |
| Health | **1.553** | **0.850** | 1.750 | 0.944 | 1.688 | 0.923 | 1.568 | 0.859 |
| Security | **72.475** | **4.084** | 75.742 | 4.382 | 76.922 | 4.482 | 73.294 | 4.187 |
| **Social Good** | **0.838** | **0.516** | 0.882 | 0.545 | 0.868 | 0.531 | 0.848 | 0.522 |
| Traffic | **0.161** | **0.289** | 0.184 | 0.296 | 0.188 | 0.304 | 0.166 | 0.293 |

Table 20: Zero-shot comparisions among Aurora and other foundation models.

| Models | Aurora (zero-shot) | | Sundial (zero-shot) | | VisionTS (zero-shot) | | ROSE (zero-shot) | | MOIRAI (zero-shot) | |
|---|---|---|---|---|---|---|---|---|---|---|
| Metric | MSE | MAE | MSE | MAE | MSE | MAE | MSE | MAE | MSE | MAE |
| Agriculture | **0.272** | 0.348 | 0.373 | 0.392 | 0.290 | **0.336** | 0.345 | 0.372 | 0.272 | 0.403 |
| Climate | **0.865** | **0.749** | 1.154 | 0.881 | 1.307 | 0.930 | 1.475 | 0.987 | 1.921 | 1.095 |
| Economy | **0.033** | **0.146** | 0.291 | 0.432 | 0.301 | 0.442 | 0.289 | 0.433 | 0.405 | 0.512 |
| Energy | **0.255** | 0.370 | 0.272 | **0.367** | 0.304 | 0.420 | 0.386 | 0.479 | 0.324 | 0.417 |
| Environment | **0.276** | **0.379** | 0.336 | 0.416 | 0.354 | 0.436 | 0.392 | 0.456 | 0.351 | 0.403 |
| Health | **1.553** | **0.850** | 1.970 | 0.992 | 2.436 | 1.221 | 2.598 | 1.201 | 2.736 | 1.241 |
| Security | 72.475 | 4.084 | **70.441** | **4.005** | 79.598 | 4.597 | 84.324 | 4.765 | 93.245 | 5.173 |
| Social Good | **0.838** | **0.516** | 1.036 | 0.573 | 1.126 | 0.618 | 1.141 | 0.581 | 1.430 | 0.651 |
| Traffic | **0.161** | **0.289** | 0.271 | 0.405 | 0.281 | 0.407 | 0.341 | 0.451 | 0.406 | 0.468 |

Table 21: 10% few shot comparisions among Aurora and other end-to-end multimodal models.

| Models | Aurora (10% few-shot) | | GPT4MTS (10% few-shot) | | TATS (10% few-shot) | | CALF (10% few-shot) | | TimeVLM (10% few-shot) | |
|---|---|---|---|---|---|---|---|---|---|---|
| Metric | MSE | MAE | MSE | MAE | MSE | MAE | MSE | MAE | MSE | MAE |
| Agriculture | **0.212** | **0.293** | 7.277 | 1.695 | 5.793 | 1.512 | 0.275 | 0.344 | 0.332 | 0.365 |
| Climate | **0.862** | **0.746** | 1.015 | 0.821 | 1.033 | 0.828 | 1.428 | 0.970 | 1.477 | 0.983 |
| Economy | **0.016** | **0.099** | 0.274 | 0.424 | 0.232 | 0.390 | 0.034 | 0.150 | 0.273 | 0.414 |
| Energy | **0.230** | **0.329** | 0.948 | 0.730 | 1.408 | 0.893 | 0.473 | 0.536 | 0.331 | 0.433 |
| Environment | **0.265** | **0.372** | 0.738 | 0.596 | 0.652 | 0.564 | 0.334 | 0.397 | 0.437 | 0.472 |
| Health | **1.343** | **0.776** | 3.885 | 1.377 | 2.781 | 1.167 | 1.762 | 0.939 | 1.947 | 0.992 |
| Security | **70.062** | **3.988** | 81.078 | 4.670 | 85.677 | 4.858 | 181.619 | 7.312 | 103.113 | 5.344 |
| Social Good | **0.814** | **0.494** | 10.579 | 1.716 | 11.612 | 1.500 | 1.037 | 0.457 | 1.017 | 0.527 |
| Traffic | **0.157** | **0.290** | 3.013 | 1.340 | 2.613 | 1.121 | 0.334 | 0.422 | 0.280 | 0.397 |

# E  RELATED WORKS

Time Series Analysis assumes a position of utmost significance within a wide array of fields, including the economy (Qiu et al., 2025b;a; Mei et al., 2025), transportation (Wu et al., 2025f; 2024; Guo et al., 2014; Liu et al., 2025a), health (Lu et al., 2023; Miao et al., 2024a; Lu et al., 2024; Wang et al., 2026a), weather (Li et al., 2025a; Miao et al., 2024b; Tian et al., 2026; 2025), and energy (Sun et al., 2025a; Feng et al.). It encompasses numerous crucial tasks, such as forecasting (Li et al., 2025b; Cheng et al., 2023; Shang et al., 2026; 2024), anomaly detection (Wang et al., 2025d; Qiu et al., 2025c; Wu et al., 2025e), classification (Liu et al., 2024d), imputation (Wu et al., 2023; 2025b; Wang et al., 2025a; Shang & Chen, 2024), irregular forecasting (Liu et al., 2026b;a; Yu et al., 2025a), motion forecasting (Ma et al., 2025a; 2024a;b; 2025c;b) and others (Qiu et al., 2025f; Wang et al., 2023; Yu et al., 2025b; Cheng et al., 2026; Pan et al., 2023). Among these tasks, Time Series Forecasting is the one most widely applied in real-world situations.

Time series forecasting (TSF) entails the prediction of future observations based on historical ones. Research findings have indicated that features learned through certain methods may exhibit superior performance compared to human-designed features (Liu et al., 2025d; Sun et al., 2025b;a; Niu et al., 2025; Chen et al., 2023). By harnessing the representation learning capabilities of deep neural networks (DNNs), a multitude of deep-learning based approaches have emerged. For instance, TimesNet (Wu et al., 2023) and SegRNN (Lin et al., 2023) model time series as vector sequences, employing CNNs or RNNs to capture temporal dependencies. Moreover, Transformer architectures, including Informer (Zhou et al., 2021), Dsformer (Yu et al., 2023), TimeFilter (Hu et al., 2025b), TimeBridge (Liu et al., 2025c), PDF (Dai et al., 2024b), Triformer (Cirstea et al., 2022), PatchTST (Nie et al., 2023), ROSE (Wang et al., 2025f), LightGTS (Wang et al., 2025e), and FLAME (Wu et al., 2025a), are capable of more accurately capturing the complex relationships between time points, thereby significantly enhancing forecasting performance. MLP-based methods, such as DUET (Qiu et al., 2025e), $K^2$VAE (Wu et al., 2025d), SRSNet (Wu et al., 2025c), AMD (Hu et al., 2025a), SparseTSF (Lin et al., 2024a), CycleNet (Lin et al., 2024b), NLinear (Zeng et al., 2023), and DLinear (Zeng et al., 2023), adopt relatively simpler architectures with fewer parameters yet still manage to achieve highly competitive forecasting accuracy. Other approaches model the inductive biases within temporal data more effectively from the perspective of loss functions, such as DBLoss (Qiu et al., 2025d), Time-O1 (Wang et al., 2025c), FreDF (Wang et al., 2025b), and DsitDF (Wang et al., 2026b).

