# OpenReview forum: "Aurora: Towards Universal Generative Multimodal Time Series Forecasting"
_ICLR.cc/2026/Conference — ICLR 2026 Poster_

### Official Review · Reviewer_UdWk · 2025-10-23

**Soundness:** 3
**Presentation:** 3
**Contribution:** 3
**Rating:** 6
**Confidence:** 4

**Summary:**

This paper proposes Aurora, a multimodal time series foundation model for zero-shot forecasting. Aurora is pretrained on cross-domain corpus with time series, text descriptions, and endogenous images. The encoder uses token distillation and modality-guided self-attention to fuse multimodal features. The decoder employs prototype-guided flow matching for probabilistic forecasting, where prototypes encode periodicity/trend initialized from text and image features. Experiments on TimeMMD, TSFM-Bench, and ProbTS show SOTA performance on both multimodal and unimodal tasks.

**Strengths:**

**S1. Novel multimodal foundation model architecture**

Aurora introduces a well-designed multimodal architecture with token distillation and modality-guided self-attention. The modality-guided mechanism (Equations 12-16) explicitly uses cross-modality correlations to adjust temporal attention, enabling domain knowledge injection. The prototype-guided flow matching is innovative, retrieving learned period/trend prototypes as starting points rather than Gaussian noise. Ablations (Table 4) validate both components with 16-70% MSE increases when removed.

**S2. Strong empirical performance across diverse benchmarks**

Aurora achieves SOTA on three benchmarks: rank 1 on MSE/MAE in 31/36 settings on TimeMMD (27% MSE reduction vs. Sundial), 24/28 on TSFM-Bench (15.1% vs. Time-MoE), and 18/23 CRPS/NMAE on ProbTS (21.5% vs. CSDI). The model shows strong zero-shot generalization, outperforming full-shot supervised models on several datasets (Climate, Environment). The 10%-shot results demonstrate data efficiency with 12.8% improvement over GPT4MTS using 90% less training data.

**S3. Comprehensive evaluation coverage**

Experiments span 20 datasets covering multimodal forecasting, unimodal deterministic forecasting, and probabilistic forecasting. The evaluation includes both zero-shot and few-shot scenarios, demonstrating versatility across task types and generalization capabilities.

**Weaknesses:**

**W1. Insufficient ablation on modality contributions:** Table 4 only ablates entire mechanisms but does not quantify individual modality contributions. The paper lacks experiments comparing time-only, time+text, time+image, and time+text+image performance. This omission makes it impossible to understand what each modality provides. Since images are deterministic transformations of time series (Equations 3-7), readers cannot assess whether images truly add information beyond text or simply provide redundant patterns already in the time series.

**W2. Insufficient ablations on modality-guided attention design:** Variant 1 removes the entire modality-guided mechanism, but the paper lacks finer-grained ablations to validate specific design choices. The Corr matrix computation (Equation 14) multiplies $VAttn$, learnable metric $W$, and $TAttn^T$, but alternative designs (text-only guidance, image-only guidance, additive combination) are not tested. The necessity of the learnable metric W is not validated.

**W3. Confusing experimental setup mixing training regimes:** Table 1 compares 10%-shot Aurora against full-shot baselines, conflating model capability with data efficiency. While this demonstrates Aurora's data efficiency, the comparison is confusing. Zero-shot Aurora outperforms some full-shot models, but readers cannot determine whether advantages come from architecture or simply using less data. A clearer separation of zero-shot comparisons (foundation model vs. foundation model) and few-shot comparisons (Aurora vs. baselines both at 10% data) would eliminate ambiguity.

**W4. No interpretability analysis of learned representations:** The paper provides no visualization or analysis of what Aurora learns. The 1000 learned prototypes in the bank are not visualized or characterized. Whether prototypes cluster into interpretable patterns (linear trends, sinusoids, seasonality) is unknown. Which prototypes are retrieved for different domains is not analyzed. Attention pattern changes with/without modality guidance are not shown. This absence prevents validating that Aurora learns meaningful multimodal representations rather than memorizing dataset-specific patterns.

**W5. Benchmark fragmentation creates unnecessary complexity:** Three separate benchmarks (TimeMMD, TSFM-Bench, ProbTS) use different evaluation protocols, metrics (MSE/MAE vs. CRPS/NMAE), and stride settings. Tables 9-11 in the appendix span many pages with inconsistent experimental configurations. This fragmentation increases complexity and reduces reproducibility. A unified benchmark covering multimodal, deterministic, and probabilistic tasks would streamline evaluation.

**W6. Limited efficiency and scalability analysis:** Table 8 reports 83.5ms inference time, but Figure 5 shows 100 samples are needed for good probabilistic performance. Training time, training cost, and parameter efficiency (accuracy per parameter) are not reported. For a 418M parameter model claiming to be a foundation model, understanding computational requirements and scalability is important for practical deployment.

**Questions:**

**Q1. Can you provide individual modality ablations?**

Report performance with time-only, time+text, time+image, and time+text+image to quantify each modality's contribution and validate that each adds value.

**Q2. Can you provide finer ablations of modality-guided attention?**

Test text-only guidance, image-only guidance, alternative combination methods, and ablate the learnable metric W to validate design choices in Equations 12-14.

**Q3. Can you visualize and characterize learned prototypes?**

Show visualizations of the 1000 prototypes, analyze whether they form interpretable clusters, and demonstrate which prototypes are retrieved for different domains and patterns.

**Q4. What are typical LLM-generated description examples?**

Provide more examples beyond Figure 7. How do you ensure generated descriptions contain genuine domain knowledge comparable to human expert knowledge? Have you validated description quality through human evaluation or comparison with real domain texts?

**Q5. Can you clarify the zero-shot vs. full-shot comparison?**

In Table 1, is the zero-shot vs. full-shot comparison primarily demonstrative rather than a rigorous benchmark? Can you add 10%-shot baseline results for fair apples-to-apples comparison?

**Q6. Can you document the full pretraining corpus?**

Provide the complete list of pretraining datasets with names, sizes, and explicit confirmation of which evaluation datasets are excluded to verify no data leakage.

**Q7. Would a unified benchmark be feasible?**

Could you propose or adopt a single unified benchmark covering multimodal, deterministic, and probabilistic forecasting to reduce experimental complexity and improve reproducibility?

---

> ### Author Response · Authors · 2025-11-21
> **Response for W1, W2, Q1, Q2**
>
> Thanks for the suggestions. According to the suggested ablation studies, we conduct additional experiments to study on the modality ablations and finer ablations of modality-guided attention. Since we have considered the modality absence phenomena in model design and training phases, so each corresponding multimodal modules can be directly removed to conduct the experiments in a zero-shot paradigm.
>
> Specifically, we make ablation studies on six variants, including three modality variants (Time-Only, Time + Text, Time + Image) and three variants of modality-guided attention (Text-Guidance, Image-Guidance, without the learnable metric W). These experiments can provide more details about how each part influences the performance. We report the full results here and have revised the draft (see Tables 18—19 in Appendix D):
>
> | Models | Aurora |  | Time-Only |  | Time + Text |  | Time + Image |  |
> | --- | --- | --- | --- | --- | --- | --- | --- | --- |
> | Metrics | MSE | MAE | MSE | MAE | MSE | MAE | MSE | MAE |
> | Agriculture | **0.272** | **0.348** | 0.337 | 0.382 | 0.304 | 0.355 | 0.294 | 0.351 |
> | Climate | **0.865** | **0.749** | 1.287 | 0.926 | 1.167 | 0.904 | 1.228 | 0.897 |
> | Economy | **0.033** | **0.146** | 0.064 | 0.198 | 0.046 | 0.167 | 0.039 | 0.152 |
> | Energy | **0.255** | **0.370** | 0.324 | 0.426 | 0.292 | 0.413 | 0.285 | 0.406 |
> | Environment | **0.276** | **0.379** | 0.352 | 0.404 | 0.334 | 0.397 | 0.325 | 0.394 |
> | Health | **1.553** | **0.850** | 2.305 | 1.147 | 1.962 | 0.987 | 1.874 | 0.972 |
> | Security | **72.475** | **4.084** | 92.822 | 5.092 | 81.294 | 4.800 | 77.928 | 4.628 |
> | Social Good | **0.838** | **0.516** | 1.387 | 0.692 | 1.018 | 0.576 | 1.037 | 0.572 |
> | Traffic | **0.161** | **0.289** | 0.345 | 0.472 | 0.271 | 0.418 | 0.198 | 0.334 |
>
> | Models | Aurora |  | Text-Guidance |  | Image-Guidance |  | w/o W |  |
> | --- | --- | --- | --- | --- | --- | --- | --- | --- |
> | Metrics | MSE | MAE | MSE | MAE | MSE | MAE | MSE | MAE |
> | Agriculture | **0.272** | **0.348** | 0.287 | 0.350 | 0.279 | 0.353 | 0.274 | 0.351 |
> | Climate | **0.865** | **0.749** | 0.885 | 0.767 | 0.898 | 0.785 | 0.876 | 0.755 |
> | Economy | **0.033** | **0.146** | 0.040 | 0.157 | 0.038 | 0.154 | 0.034 | 0.148 |
> | Energy | **0.255** | **0.370** | 0.274 | 0.387 | 0.262 | 0.374 | 0.265 | 0.376 |
> | Environment | **0.276** | **0.379** | 0.294 | 0.386 | 0.285 | 0.389 | 0.280 | 0.381 |
> | Health | **1.553** | **0.850** | 1.750 | 0.944 | 1.688 | 0.923 | 1.568 | 0.859 |
> | Security | **72.475** | **4.084** | 75.742 | 4.382 | 76.922 | 4.482 | 73.294 | 4.187 |
> | Social Good | **0.838** | **0.516** | 0.882 | 0.545 | 0.868 | 0.531 | 0.848 | 0.522 |
> | Traffic | **0.161** | **0.289** | 0.184 | 0.296 | 0.188 | 0.304 | 0.166 | 0.293 |
>
> We have the following important observations:
>
> 1. In the modality variants, using the single time modality leads to the worst performance in the multimodal forecasting scenarios. And adding either Text or Image modality can improve the performance, adding both of them (Aurora) can produce a cascading effect.
> 2. In the variants of modality-guided attention, though removing the learnable metric W as identity matrix produces relatively smallest negative effect, we think this design is still necessary. Because its parameter count is very small, and it does theoretically and empirically help construct cross-modality relationships. For Text-Guidance and Image-Guidance variants, the absence of the other modality decreases the performance of Aurora, but less than directly removing the entire modality (Time + Image, Time + Text, Only Time).

---

> > ### Author Response · Authors · 2025-11-21
> > **Response for W3, Q5**
> >
> > Thanks for the suggestions! Our original intention is to indicate that Aurora can outperform end-to-end models trained with all data by fine-tuning with only 10% data. To make the comparisons more clear and demonstrate Aurora’s capability, we separate the settings of zero-shot and few-shot in the following two Tables, you can also refer to our revised draft (see Tables 20—21 in Appendix D):
> >
> > | Models | Aurora  (zero-shot) |  | Sundial (zero-shot) |  | VisionTS (zero-shot) |  | ROSE (zero-shot) |  | MOIRAI (zero-shot) |  |
> > | --- | --- | --- | --- | --- | --- | --- | --- | --- | --- | --- |
> > | Metric | MSE | MAE | MSE | MAE | MSE | MAE | MSE | MAE | MSE | MAE |
> > | Agriculture | **0.272** | 0.348 | 0.373 | 0.392 | 0.290 | **0.336** | 0.345 | 0.372 | 0.272 | 0.403 |
> > | Climate | **0.865** | **0.749** | 1.154 | 0.881 | 1.307 | 0.930 | 1.475 | 0.987 | 1.921 | 1.095 |
> > | Economy | **0.033** | **0.146** | 0.291 | 0.432 | 0.301 | 0.442 | 0.289 | 0.433 | 0.405 | 0.512 |
> > | Energy | **0.255** | 0.370 | 0.272 | **0.367** | 0.304 | 0.420 | 0.386 | 0.479 | 0.324 | 0.417 |
> > | Environment | **0.276** | **0.379** | 0.336 | 0.416 | 0.354 | 0.436 | 0.392 | 0.456 | 0.351 | 0.403 |
> > | Health | **1.553** | **0.850** | 1.970 | 0.992 | 2.436 | 1.221 | 2.598 | 1.201 | 2.736 | 1.241 |
> > | Security | 72.475 | 4.084 | **70.441** | **4.005** | 79.598 | 4.597 | 84.324 | 4.765 | 93.245 | 5.173 |
> > | Social Good | **0.838** | **0.516** | 1.036 | 0.573 | 1.126 | 0.618 | 1.141 | 0.581 | 1.430 | 0.651 |
> > | Traffic | **0.161** | **0.289** | 0.271 | 0.405 | 0.281 | 0.407 | 0.341 | 0.451 | 0.406 | 0.468 |
> >
> > | Models | Aurora (10% few-shot) |  | GPT4MTS (10% few-shot) |  | TATS (10% few-shot) |  | CALF (10% few-shot) |  | TimeVLM (10% few-shot) |  |
> > | --- | --- | --- | --- | --- | --- | --- | --- | --- | --- | --- |
> > | Metric | MSE | MAE | MSE | MAE | MSE | MAE | MSE | MAE | MSE | MAE |
> > | Agriculture | **0.212** | **0.293** | 7.277 | 1.695 | 5.793 | 1.512 | 0.275 | 0.344 | 0.332 | 0.365 |
> > | Climate | **0.862** | **0.746** | 1.015 | 0.821 | 1.033 | 0.828 | 1.428 | 0.970 | 1.477 | 0.983 |
> > | Economy | **0.016** | **0.099** | 0.274 | 0.424 | 0.232 | 0.390 | 0.034 | 0.150 | 0.273 | 0.414 |
> > | Energy | **0.230** | **0.329** | 0.948 | 0.730 | 1.408 | 0.893 | 0.473 | 0.536 | 0.331 | 0.433 |
> > | Environment | **0.265** | **0.372** | 0.738 | 0.596 | 0.652 | 0.564 | 0.334 | 0.397 | 0.437 | 0.472 |
> > | Health | **1.343** | **0.776** | 3.885 | 1.377 | 2.781 | 1.167 | 1.762 | 0.939 | 1.947 | 0.992 |
> > | Security | **70.062** | **3.988** | 81.078 | 4.670 | 85.677 | 4.858 | 181.619 | 7.312 | 103.113 | 5.344 |
> > | Social Good | **0.814** | **0.494** | 10.579 | 1.716 | 11.612 | 1.500 | 1.037 | 0.457 | 1.017 | 0.527 |
> > | Traffic | **0.157** | **0.290** | 3.013 | 1.340 | 2.613 | 1.121 | 0.334 | 0.422 | 0.280 | 0.397 |
> >
> > We have the following important observations:
> >
> > 1. In the zero-shot comparisons, Aurora consistently outperforms time series foundation models, demonstrating the effectiveness of multimodal modeling.
> > 2. In the 10% few-shot comparisons, Aurora also outperforms all baselines a lot and demonstrates that its strong generalization capability comes from pretraining and model design. Only fine-tuning end-to-end models like CALF and TimeVLM with 10% data will not produce strong forecasting accuracies.

---

> > > ### Author Response · Authors · 2025-11-21
> > > **Response for W4, Q3**
> > >
> > > Following your advice, we conduct visual analysis in the following two aspects:
> > >
> > > 1. we visualize the 1,000 learned prototypes in PrototypeBank **in Figure 14 of Appendix C.2, which includes versatile prototypes like linear trends, sinusoids, seasonality as you mentioned. You can also refer to the Figures 15—17 in Appendix C.3, which involve 28 specific visualization cases in forecasting.**
> > > It is observed that Aurora can retrieve-then-generate proper prototypes for predictions, which contains periodic or trend implications for future horizons. For example, in Figures 15 (b)—(d), the generated prototypes do imply the future periods in the power transformer systems (ETTs). In Figures 15 (e), (f), (i), (j), the generated prototypes reveal the future trends in the domains of Traffic, Weather, and Energy.
> > > 2. To reveal how attention weights change with and without modality guidance (the impact of Corr on them), we also conduct experiments in **Figures 18—21 of Appendix C.4.**
> > > For example, In Figure 18, It is observed that the predictions with modality guidance are more accurate. The 4 patches (T1—T4) of the contextual time series show similar correlations without modality guidance. While with the modality guidance, the correlations between T1 and T2 are further focused on, because their correlations are similar to the T4 and future values, with a trend of first decreasing and then increasing. Additionally, the correlations between T4 and T1 are also focused on for prediction.

---

> > > > ### Author Response · Authors · 2025-11-21
> > > > **Response for W5, Q7**
> > > >
> > > > Thanks for the advice. However, conducting experiments on multiple datasets and evaluation settings across various benchmarks is an effective way to verify the capabilities of foundation models, and this approach is widely used not only in the field of time series but also in NLP and CV.
> > > >
> > > > In this work, we use three benchmarks in the submission version for following reasons:
> > > >
> > > > 1. Using TimeMMD to evaluate the multimodal forecasting capability of Aurora
> > > > 2. Using TSFM-Bench to evaluate the unimodal deterministic forecasting capability of Aurora
> > > > 3. Using ProbTS to evaluate the probabilistic forecasting capability of Aurora
> > > >
> > > > **These three benchmarks are used to evaluate different capabilities of Aurora respectively.** Currently, a unified benchmark for all does not exist and this can be an interesting research direction.

---

> > > > > ### Author Response · Authors · 2025-11-21
> > > > > **Response for W6**
> > > > >
> > > > > Actually, the inference time we report is already with the sampling number equal to 100, and we have already reported the pretraining cost in the Appendix A.4:
> > > > >
> > > > > “All experiments are executed on only 8 NVIDIA A800 GPUs, each equipped with 80GB of GPU
> > > > > memory, which takes about 30 days to train Aurora from scratch.”
> > > > >
> > > > > To make the zero-shot and few-shot efficiency more clear, we then compare Aurora (zero-shot) and Aurora (10% few-shot) with full-shot supervised models, i.e., TimeXer, PatchTST, iTransformer, and TimesNet. We report the training time and inference time of them on datasets with different scales, i.e., ETTh1, ETTm2, Weather, Electricity, and Traffic. **Please refer to the Table below or the Figure 9 in Appendix A.7 of our revised draft.**
> > > > >
> > > > > | Datasets | Aurora (Zero-shot) |  | Aurora (10% few-shot) |  | TimeXer |  | iTransformer |  | PatchTST |  | TimesNet |  |
> > > > > | --- | --- | --- | --- | --- | --- | --- | --- | --- | --- | --- | --- | --- |
> > > > > | Metrics | Training Time | Inference Time | Training Time | Inference Time | Training Time | Inference Time | Training Time | Inference Time | Training Time | Inference Time | Training Time | Inference Time |
> > > > > | ETTh1 | 0 | 4.728 | 148.925 | 4.728 | 46.471 | 1.22 | 34.285 | 1.029 | 21.133 | 0.991 | 105.977 | 1.912 |
> > > > > | ETTm2  | 0 | 12.825 | 282.958 | 12.825 | 92.019 | 4.712 | 72.703 | 5.288 | 71.09 | 4.919 | 249.692 | 7.073 |
> > > > > | Weather | 0 | 26.811 | 707.382 | 26.811 | 174.909 | 4.764 | 78.961 | 5.133 | 265.171 | 10.856 | 705.421 | 14.967 |
> > > > > | Electricity | 0 | 109.773 | 1471.924 | 109.773 | 354.459 | 6.775 | 158.857 | 8.889 | 1373.411 | 22.723 | 1428.109 | 78.912 |
> > > > > | Traffic  | 0 | 275.921 | 2974.827 | 275.921 | 676.938 | 11.016 | 262.631 | 24.696 | 2783.825 | 38.924 | 3855.15 | 186.625 |
> > > > >
> > > > > It is observed that the inference time of zero-shot Aurora (no training required) is consistently and largely less than the total time (training time + inference time) of full-shot models, demonstrating the strong efficiency with accurate performance. Considering the 10% few shot Aurora, though its time cost is a bit more than TimesNet, it can lead to state-of-the-art performance in most cases.

---

> > > > > > ### Author Response · Authors · 2025-11-21
> > > > > > **Response for Q4 and Q6**
> > > > > >
> > > > > > Following your advice, we provide more examples **in Figure 8 of Appendix A.1.** In the generation process of textual descriptions, the quality of texts is exactly important so we do adopt some techniques to ensure that. After a GPT4 agent generates the textual descriptions, we first **coarsely check the quality with another GPT4 agent**. If the quality is low, the process will be redone. After a batch of textual descriptions are generated, **we randomly sample from them (5%) and check the quality manually**, then determine whether to regenerate this batch of data and tune the prompts. We think this process does not involve data leakage problem because we only input GPT4 with the contextual time series in a single conversation.
> > > > > >
> > > > > > **We further clarify the pretraining datasets we use in Figure 7 of Appendix A.1** in the revised draft, including the ratios of domains, the specific datasets used, and the lengths and sources of them. We also revise the corresponding texts to add more implementation details. **Note that there exists no overlap in pretraining and testing datasets.**

---

> > > > > > > ### Comment · Reviewer_UdWk · 2025-11-21
> > > > > > >
> > > > > > > I appreciate the authors' comprehensive and thoughtful rebuttal. I will raise my score.

---

> > > > > > > > ### Author Response · Authors · 2025-11-21
> > > > > > > > **Thanks for your efforts**
> > > > > > > >
> > > > > > > > Dear Reviewer UdWk, we are thrilled that our responses have effectively addressed your questions and comments. We would like to express our sincerest gratitude for taking the time to review our paper and provide us with such detailed feedback!

---

### Official Review · Reviewer_qDZm · 2025-10-23

**Soundness:** 3
**Presentation:** 4
**Contribution:** 3
**Rating:** 8
**Confidence:** 5

**Summary:**

This work makes a timely and impactful contribution: it is, to the best of my knowledge, the first universal generative foundation model that unifies multimodal inputs (text + image + time series) for cross-domain zero-shot forecasting. Prior foundation models (e.g., MOIRAI, Chronos, Sundial) are unimodal; prior multimodal models (e.g., Time-LLM, CALF, GPT4MTS) are supervised and non-generative. Aurora bridges this gap. The prototype-guided flow matching is also a compelling design that provides interpretable inductive bias (trend/periodicity) to the generative process, improving sample efficiency and forecast realism.

**Strengths:**

- First generative multimodal foundation model enabling cross-domain zero/few-shot time series forecasting.
- Novel cross-modal encoder (modality-guided attention) and prototype-guided decoder design boost generalization and probabilistic prediction quality.
- Extensive experiments across diverse benchmarks (TimeMMD, TSFM-Bench, ProbTS) consistently outperform unimodal/multimodal baselines.

**Weaknesses:**

- Prototype scalability: The prototype bank contains 1,000 learnable patterns. How does performance vary with the number of prototypes (e.g., M = 100 vs. 1,000 vs. 10,000)? Is there a trade-off between forecast accuracy, training stability, and inference latency? A small ablation would clarify whether this component is over-parameterized.
- The method assumes that text and image modalities are faithful and aligned with the time series. However, in real-world settings: Text may be vague, outdated, or incorrect; and images rely on FFT-based period estimation, which can be unreliable for non-stationary or chaotic series. The paper does not evaluate modality corruption robustness (e.g., masking text, perturbing image tokens), which is critical for practical adoption.
- Can Aurora be extended to other multimodal time series tasks (e.g., anomaly detection)?

**Questions:**

See the weaknesses

---

> ### Author Response · Authors · 2025-11-21
> **Response for W1**
>
> Thanks for this valuable suggestion! Actually, we have ever studied the prototype scalability, though we did not report the results initially. Specifically, we have tried variants of 500, 1000, and 5000 and choose 1000 as the default setting. We provide the results as follows and have revised the draft (see Table 17 in Appendix D):
>
> | Models | 500 |  | 1000 |  | 5000 |  |
> | --- | --- | --- | --- | --- | --- | --- |
> | Metrics | MSE | MAE | MSE | MAE | MSE | MAE |
> | Agriculture | 0.288 | 0.364 | 0.272 | 0.348 | **0.269** | **0.344** |
> | Climate | 0.893 | 0.760 | 0.865 | **0.749** | **0.863** | 0.752 |
> | Economy | 0.034 | 0.152 | 0.033 | 0.146 | **0.032** | **0.144** |
> | Energy | 0.271 | 0.391 | **0.255** | **0.370** | 0.260 | 0.373 |
> | Environment | 0.302 | 0.413 | 0.276 | 0.379 | **0.268** | **0.372** |
> | Health | 1.587 | 0.866 | **1.553** | **0.850** | 1.561 | 0.854 |
> | Security | 75.017 | 4.118 | 72.475 | 4.084 | **71.551** | **4.036** |
> | Social Good | 0.862 | 0.547 | 0.838 | 0.516 | **0.833** | **0.507** |
> | Traffic | 0.187 | 0.304 | **0.161** | **0.289** | 0.164 | 0.296 |
>
> We observe that when the size of PrototypeBank increases from 500 to 1000, the performance of downstream tasks improves a lot. But when the size achieves 5000, there exists a phenomenon of marginal improvement, and some settings even have worse performance. Therefore, we choose to use 1000 as the default setting to preserve both the accuracy and efficiency.
>
> Following your advice, we further visualize the 1000 learned prototypes in our revised draft, **please refer to the Figure 14 in Appendix C.2.** It is observed that the diversity of these 1000 prototypes is relatively high, which is helpful for generating the proper prototypes for forecasting.

---

> > ### Author Response · Authors · 2025-11-21
> > **Response for W2**
> >
> > Thanks for this insightful advice! The ill-modality problem you mentioned exactly has practical significance. In our work, since the textual modality is exogenous, we have considered the issue of its absence during the model design and training phases, and our model also achieves SOTA on unimodal datasets, so that the potential problems of text can be handled well.
> >
> > Considering the endogenous image modality, which relies on the time series data, we have also designed some mechanisms to ensure the robustness when facing unreliable scenarios that FFT fails:
> >
> > 1. When the periodicity is vague and FFT fails, we adopt the patch size as the alternative.
> > 2. If the time series is too short, even shorter than one Patch, we will set the periodicity as the whole length.
> >
> > The above mechanisms are applied both in pretraining and inference phases, so that the imaging process of time series is robust. And some experimental results have also demonstrated the effectiveness:
> >
> > 1. The datasets Security, ILI, NYSE, and Exchange have no periodicity and exhibit drift phenomena, while Aurora shows competitive performance on them. **You can refer to the Tables 1—3 of our revised paper for the results.**
> > 2. The 8,068 short-term datasets in TFB are very short, with short input lengths (8 — 24), while Aurora achieves the state-of-the-art performance on them against all baselines, including zero-shot foundation models and full-shot small models. **You can refer to the Figure 5 and Tables 10—11 in our revised paper.**

---

> > > ### Author Response · Authors · 2025-11-21
> > > **Response for W3**
> > >
> > > Thanks for this question! As far as we know, foundation models that support forecasting can often naturally support anomaly detection, and some unimodal foundation models are evaluated in some benchmarks [1] to demonstrate this point.  However, multimodal anomaly detection is yet a new task, and lacks high-quality labels for evaluating accuracy, so that we do not conduct corresponding experiments. We set this as an important and promising future work, and thanks for your valuable suggestions again!
> > >
> > > [1] TAB: Unified Benchmarking of Time Series Anomaly Detection Methods

---

> > ### Comment · Reviewer_qDZm · 2025-11-27
> > **Official Comment by Reviewer qDZm**
> >
> > Thank you for your careful reply. I checked the newly added Figure 14, and found that many prototypes are very similar. Perhaps 1000 is relatively the best configuration. In the future, it can be further optimized by constraining the similarity between prototypes. Overall, your reply has resolved most of my concerns, so I maintain my score.

---

> > > ### Author Response · Authors · 2025-11-27
> > >
> > > Dear Reviewer qDZm,
> > >
> > > We are very grateful for the efforts you have put into reviewing the manuscript and provide such insightful and detailed reviews! In the future, we will follow your advice to further study on constraining the similarity between prototypes.
> > >
> > > Best regards,
> > > Authors.

---

### Official Review · Reviewer_t5fV · 2025-10-29

**Soundness:** 3
**Presentation:** 3
**Contribution:** 3
**Rating:** 6
**Confidence:** 4

**Summary:**

This paper presents Aurora, which is a generative foundation model to unify time series forecasting across diverse domains and modalities. First, it has token distillation and modality alignment modules for cross-modality fusion, then the modality-guided attention will inject textual and visual knowledge into temporal encoding. Lastly, in the  decoding stage, a prototype-guided flow matching will improve generative probabilistic forecasting by using learned trend and periodic prototypes. Through the experiments on TimeMMD, TSFM-Bench, and ProbTS, the authors demonstrate Aurora's good performance in zero-shot and few-shot performance across unimodal and multimodal settings.

**Strengths:**

1. It really focuses on the limitations of the existing multimodal TS forecaster and foundation models. It introduces a novel multimodal pretraining paradigm with datasets, which is a huge contribution to this field.
2. The architecture is novel. The modality guided attention bridges domain knowledge with temporal information. The prototype guided flow matching improves the interpretability of probabilistic forecasting.
3. Aurora is evaluated on diverse benchmarks with both deterministic and probabilistic metrics. The results consistently show the effectiveness of the proposed architectures.

**Weaknesses:**

1. Using LLM to generate a textual description can be problematic. The paper provides no analysis of data leakage or the quality analysis of generated texts for pretraining data. Also, the paper should provide a more rigorous clarification regarding whether the pretraining data has any overlap with the benchmark datasets.
2. Prototypes give more interpretability for this paper, but have no analysis or visualization of learned prototypes.
3. Missing TTM[1] in related work and baseline comparison.


[1] Ekambaram, Vijay, et al. "Tiny time mixers (ttms): Fast pre-trained models for enhanced zero/few-shot forecasting of multivariate time series." Advances in Neural Information Processing Systems 37 (2024): 74147-74181.

**Questions:**

1. Is the use of Flow Matching in the decoder essential, or could other generic multimodal decoding frameworks be applied instead?
2. Could you provide specific examples where your model demonstrates advantages to your claim: cases where similar historical patterns lead to different future trends due to domain-specific characteristics?

---

> ### Author Response · Authors · 2025-11-21
> **Response for W1**
>
> Thanks for the insightful suggestions. In the generation process, the quality of texts is exactly important so we do adopt some techniques to ensure that. After a GPT4 agent generates the textual descriptions, we first **coarsely check the quality with another GPT4 agent**. If the quality is low, the process will be redone. After a batch of textual descriptions are generated, **we randomly sample from them (5%) and check the quality manually**, then determine whether to regenerate this batch of data and tune the prompts. We think this process does not involve data leakage problem because we only input GPT4 with the contextual time series in a single conversation.
>
> **We further clarify the pretraining datasets we use in Figure 7 of Appendix A.1** in the revised draft, including the ratios of domains, the specific datasets used, and the lengths and sources of them. We also revise the corresponding texts to add more implementation details. **Note that there exists no overlap in pretraining and testing datasets.**

---

> > ### Author Response · Authors · 2025-11-21
> > **Response for W2**
> >
> > Following your advice, we visualize the 1,000 learned prototypes in PrototypeBank **in Figure 14 of Appendix C.2. You can also refer to the Figures 15—17 in Appendix C.3, which involve 28 specific visualization cases in forecasting.**
> >
> > It is observed that Aurora can retrieve-then-generate proper prototypes for predictions, which contains periodic or trend implications for future horizons.

---

> > > ### Author Response · Authors · 2025-11-21
> > > **Response for W3**
> > >
> > > Following your advice, we introduce TTM in related works and baselines, and then compare its performance with Aurora in Table 2 (Deterministic Forecasting). Note that TTM is a deterministic model, which cannot support probabilistic forecasting. **You can refer to the revised draft for more information.** We also list its performance here:
> > >
> > > | Models |  | Aurora |  | TTM |  |
> > > | --- | --- | --- | --- | --- | --- |
> > > | Metric |  | MSE | MAE | MSE | MAE |
> > > | ETTm1 | 96 | **0.294** | **0.351** | 0.738 | 0.541 |
> > > |  | 192 | **0.331** | **0.374** | 0.698 | 0.547 |
> > > |  | 336 | **0.359** | **0.391** | 0.670 | 0.533 |
> > > |  | 720 | **0.398** | **0.414** | 0.660 | 0.550 |
> > > |  | Avg | **0.346** | **0.383** | 0.692 | 0.543 |
> > > | ETTm2 | 96 | **0.179** | **0.270** | 0.226 | 0.309 |
> > > |  | 192 | **0.232** | **0.307** | 0.311 | 0.360 |
> > > |  | 336 | **0.275** | **0.337** | 0.350 | 0.383 |
> > > |  | 720 | **0.338** | **0.380** | 0.446 | 0.435 |
> > > |  | Avg | **0.256** | **0.324** | 0.333 | 0.372 |
> > > | ETTh1 | 96 | **0.340** | **0.381** | 0.364 | 0.389 |
> > > |  | 192 | **0.377** | **0.405** | 0.386 | 0.407 |
> > > |  | 336 | **0.399** | **0.422** | 0.404 | 0.422 |
> > > |  | 720 | 0.428 | 0.450 | **0.424** | **0.448** |
> > > |  | Avg | **0.386** | **0.415** | 0.395 | 0.417 |
> > > | ETTh2 | 96 | **0.259** | **0.325** | 0.277 | 0.335 |
> > > |  | 192 | **0.324** | **0.370** | 0.334 | 0.373 |
> > > |  | 336 | **0.360** | **0.401** | 0.362 | 0.402 |
> > > |  | 720 | **0.403** | **0.441** | 0.408 | 0.444 |
> > > |  | Avg | **0.337** | **0.384** | 0.345 | 0.389 |
> > > | Weather | 96 | **0.160** | **0.207** | 0.183 | 0.242 |
> > > |  | 192 | **0.202** | **0.247** | 0.229 | 0.285 |
> > > |  | 336 | **0.252** | **0.288** | 0.289 | 0.330 |
> > > |  | 720 | **0.307** | **0.327** | 0.359 | 0.370 |
> > > |  | Avg | **0.230** | **0.267** | 0.265 | 0.307 |
> > > | Electricity | 96 | **0.134** | **0.234** | 0.166 | 0.263 |
> > > |  | 192 | **0.161** | **0.258** | 0.191 | 0.286 |
> > > |  | 336 | **0.193** | **0.287** | 0.237 | 0.336 |
> > > |  | 720 | **0.224** | **0.320** | 0.292 | 0.384 |
> > > |  | Avg | **0.178** | **0.275** | 0.222 | 0.317 |
> > > | Traffic | 96 | **0.435** | **0.314** | 0.514 | 0.347 |
> > > |  | 192 | **0.465** | **0.328** | 0.543 | 0.373 |
> > > |  | 336 | **0.525** | **0.355** | 0.575 | 0.389 |
> > > |  | 720 | **0.670** | **0.411** | 0.622 | 0.433 |
> > > |  | Avg | **0.524** | **0.352** | 0.564 | 0.386 |
> > > | Solar | 96 | **0.185** | **0.272** | 0.863 | 0.664 |
> > > |  | 192 | **0.198** | **0.282** | 0.823 | 0.695 |
> > > |  | 336 | **0.211** | **0.294** | 0.835 | 0.741 |
> > > |  | 720 | **0.218** | **0.307** | 0.738 | 0.738 |
> > > |  | Avg | **0.203** | **0.289** | 0.815 | 0.710 |
> > > | PEMS08 | 96 | **0.463** | **0.513** | 1.284 | 0.888 |
> > > |  | 192 | **0.599** | **0.577** | 1.638 | 1.039 |
> > > |  | 336 | **0.560** | **0.546** | 1.979 | 1.160 |
> > > |  | 720 | **0.629** | **0.570** | 2.020 | 1.175 |
> > > |  | Avg | **0.563** | **0.552** | 1.730 | 1.066 |
> > > | Wind | 96 | **0.951** | **0.664** | 1.077 | 0.701 |
> > > |  | 192 | **1.115** | **0.747** | 1.350 | 0.825 |
> > > |  | 336 | **1.229** | **0.799** | 1.473 | 0.891 |
> > > |  | 720 | **1.309** | **0.840** | 1.447 | 0.899 |
> > > |  | Avg | **1.151** | **0.763** | 1.337 | 0.829 |
> > >
> > > Obviously, Aurora outperforms TTM in almost all evaluation settings.

---

> > > > ### Author Response · Authors · 2025-11-21
> > > > **Response for Q1**
> > > >
> > > > Thanks for this insightful question. From our perspective, using the proposed Prototype-Guided Flow Matching can better utilize the multimodal information in the following three aspects:
> > > >
> > > > 1. Like most generative models, Aurora also integrates the multimodal information in condition networks, thus can model the conditional probabilistic distributions.
> > > > 2. Based on the motivation that the future trends and periods are implied by multimodal information, we propose the Prototype-Guided mechanism to adaptively generate prototypes as the starting points by using visual and textual information, which can help the flow matching head forecast accurately.
> > > > 3. Current end-to-end multimodal models often adopt decoders like unimodal models, i.e., MLP-based flatten heads. And they simply fuse multimodal information through extra MLPs in the decoding parts, which are not proper in foundation models.
> > > >
> > > > Actually, we have tried other mechanisms like VAE, Diffusion, and Normalized Flow, though we did not report the results initially. Empirically, all of them are not as strong as the proposed Prototype-Guided Flow Matching. We list the comparison results as follows:
> > > >
> > > > | Models | Aurora |  | VAE |  | Diffusion |  | Normalization Flow |  |
> > > > | --- | --- | --- | --- | --- | --- | --- | --- | --- |
> > > > | Metrics | MSE | MAE | MSE | MAE | MSE | MAE | MSE | MAE |
> > > > | Agriculture | **0.272** | **0.348** | 0.347 | 0.388 | 0.289 | 0.367 | 0.301 | 0.354 |
> > > > | Climate | **0.865** | **0.749** | 1.261 | 0.927 | 1.317 | 0.894 | 0.992 | 0.783 |
> > > > | Economy | **0.033** | **0.146** | 0.065 | 0.278 | 0.125 | 0.402 | 0.075 | 0.341 |
> > > > | Energy | **0.255** | **0.370** | 0.273 | 0.395 | 0.261 | 0.375 | 0.258 | 0.381 |
> > > > | Environment | **0.276** | **0.379** | 0.347 | 0.448 | 0.351 | 0.460 | 0.328 | 0.427 |
> > > > | Health | **1.553** | **0.850** | 1.648 | 0.872 | 1.577 | 0.863 | 1.781 | 0.901 |
> > > > | Security | **72.475** | **4.084** | 83.821 | 4.692 | 74.951 | 4.633 | 93.182 | 5.029 |
> > > > | Social Good | **0.838** | **0.516** | 1.055 | 0.608 | 0.982 | 0.503 | 1.152 | 0.617 |
> > > > | Traffic | **0.161** | **0.289** | 0.247 | 0.382 | 0.195 | 0.304 | 0.197 | 0.311 |

---

> > > > > ### Author Response · Authors · 2025-11-21
> > > > > **Response for Q2**
> > > > >
> > > > > We have provided specific examples in our revised draft. **Please refer to Figures 10—13 in Appendix C.1.** Among them, **Figures 10 and 11 indicate the cases that datasets from distinct domains possess similar historical time series but dissimilar future values. In Figures 12 and 13, the cases show that datasets from similar domains can even show dissimilar future values**, It is proving that this phenomenon is universal. So that, considering the multimodal knowledge in forecasting is essential.

---

> > > > > > ### Comment · Reviewer_t5fV · 2025-11-26
> > > > > >
> > > > > > I appreciate the authors' detailed and comprehensive rebuttal, and it resolved my concerns. I would maintain my positive score.

---

> > > > > > > ### Author Response · Authors · 2025-11-27
> > > > > > >
> > > > > > > Dear Reviewer t5fv,
> > > > > > >
> > > > > > > We really appreciate your efforts during this tight review timeline and the recognition of the strengths of our paper.
> > > > > > >
> > > > > > > Best regards,
> > > > > > >
> > > > > > > Authors.

---

### Official Review · Reviewer_MY6v · 2025-11-01

**Soundness:** 3
**Presentation:** 3
**Contribution:** 2
**Rating:** 4
**Confidence:** 4

**Summary:**

The authors propose Aurora, a multimodal time series foundation model. By leveraging multimodal information fusion, Aurora achieves impressive zero-shot forecasting performance. The paper also introduces a new decoder design. Experimental results demonstrate that the proposed method outperforms baseline models across various datasets.

**Strengths:**

- The authors propose a powerful zero-shot time series foundation model that effectively integrates multimodal information for prediction.
- The proposed method integrates different modules and achieves remarkable performance in terms of robustness.

**Weaknesses:**

### **Experiment Setting and Performance**

The proposed method achieves better zero-shot performance than baselines. However, MAE and MSE are greater than 0.5, as seen in PEMS08, Traffic, and Wind. I am not sure whether the metrics are calculated based on normalized results. If so, a high MAE or MSE may not be better than a straight line or random output.


### **Inference time**

The work focuses on few-shot learning and zero-shot learning. However, compared to the training cost, such a large model increases the inference time and requires more GPU resources. I suggest that the author include an experiment to compare the proposed 16G model with iTransformer on ETT, traffic, or weather, in terms of total running time (training time + inference time for a fair comparison). I am curious about whether the training time and inference time of a small model may be quicker than that of a large model.

### **Baselines**

I noticed the author only chose large foundation models to evaluate few-shot and zero-shot learning. Could the author test small models, e.g., iTransformer and recently advanced methods, on few-shot learning with different budgets, such as 10% and 20%?

### **Full-shot Setting**

Due to the training cost of the small model, the author should also compare their proposed method with unimodal models on full-shot settings, such as PatchTST, iTransformer, TimeXer, or other new advanced methods.

### **Benchmarks**
As a time series forecasting task, prediction is 7.5 times the input length, which may not be useful in daily life. Such as forecasting the future one-week weather, but only based on 16 hours. I think the authors should consider making their work more valuable and closer to everyday life settings. Moreover, the author should add more short-term forecasting datasets, such as M4, PEMS, illness, and EPF datasets.

**Questions:**

I am willing to adjust the rating based on the authors' feedback, particularly regarding my concerns about baselines, benchmarks, and experiment settings.
Please refer to the weakness.

**Details Of Ethics Concerns:**

The author claims not to use LLM in writing in L762-763.  However, some sentences appear to be generated by a third-party check tool, similar to GPT-generated text. Since this verification process may not be conclusive, I am only reporting this issue as a reference.

For example
> ### **Normalized Mean Absolute Error (NMAE)**
> The NMAE is an extension of the MAE. It normalizes the MAE with respect to the total magnitude of the ground-truth values. This normalization process facilitates a fair comparison across datasets that have different scales. The formula for NMAE is as follows:

---

> ### Author Response · Authors · 2025-11-21
> **Response for Experiment Setting and Performance**
>
> Thanks for your insightful suggestions. Following the convention in time series analytics, we use the normalized results of MSE and MAE. We would like to clarify that the values of normalized MSE and MAE greater than 0.5 does not necessarily mean that the forecasting results are straight lines or random outputs. To demonstrate it, we conduct an experiment to compare Aurora with NaiveMean and NaiveRepeat on PMES08, Traffic, and Wind, where the **NaiveMean** outputs a straight line with the mean values of normalized input time series, and the **NaiveRepeat** outputs another straight line with the last value of normalized input time series. The results are shown as:
>
> | Models | Horizons | Aurora |  | Timer |  | NaiveMean |  | NaiveRepeat |  |
> | --- | --- | --- | --- | --- | --- | --- | --- | --- | --- |
> | Metric |  | MSE | MAE | MSE | MAE | MSE | MAE | MSE | MAE |
> | Traffic | 96 | **0.435** | **0.314** | 0.625 | 0.580 | 1.410 | 0.805 | 2.714 | 1.077 |
> |  | 192 | **0.465** | **0.328** | 0.798 | 0.661 | 1.413 | 0.806 | 2.747 | 1.085 |
> |  | 336 | **0.525** | **0.355** | 0.910 | 0.716 | 1.429 | 0.809 | 2.788 | 1.094 |
> |  | 720 | **0.670** | **0.411** | 1.131 | 0.824 | 1.451 | 0.813 | 2.809 | 1.097 |
> |  | Avg | **0.524** | **0.352** | 0.866 | 0.695 | 1.426 | 0.808 | 2.765 | 1.088 |
> | PEMS08 | 96 | **0.463** | **0.513** | 0.946 | 0.659 | 2.213 | 1.259 | 1.480 | 0.914 |
> |  | 192 | **0.599** | **0.577** | 1.142 | 0.758 | 2.305 | 1.260 | 2.452 | 1.246 |
> |  | 336 | **0.560** | **0.546** | 1.300 | 0.830 | 1.834 | 1.053 | 2.032 | 1.074 |
> |  | 720 | **0.629** | **0.570** | 1.417 | 0.884 | 1.990 | 1.108 | 2.288 | 1.163 |
> |  | Avg | **0.563** | **0.552** | 1.201 | 0.783 | 2.086 | 1.170 | 2.063 | 1.099 |
> | Wind | 96 | **0.951** | **0.664** | 0.388 | 0.409 | 1.145 | 0.750 | 1.253 | 0.725 |
> |  | 192 | **1.115** | **0.747** | 0.778 | 0.613 | 1.308 | 0.820 | 1.552 | 0.847 |
> |  | 336 | **1.229** | **0.799** | 1.200 | 0.792 | 1.527 | 0.899 | 1.836 | 0.950 |
> |  | 720 | **1.309** | **0.840** | 1.584 | 1.000 | 1.723 | 0.968 | 2.145 | 1.053 |
> |  | Avg | **1.151** | **0.763** | 0.988 | 0.704 | 1.426 | 0.859 | 1.697 | 0.894 |
>
> We have the following important observations:
>
> 1. Timer’s pre-training datasets do not overlap with these three datasets, while the values of normalized MSE and MAE are also greater than 0.5. So that values greater than 0.5 may not necessarily indicate poor forecasts.
> 2. It is observed that such naive cases cause the collapse of performance, about **25% —500% worse** than Aurora.
>
> **You can also find more intuitive showcases in our revised draft, please see Figures 15 (f), (h), (j) in Appendix C.3, where Aurora accurately predicts the periods and trends on these datasets, instead of straight lines or random outputs.**

---

> > ### Author Response · Authors · 2025-11-21
> > **Response for inference time**
> >
> > Thanks for your advice. Since Aurora is a large multimodal foundation model, we mainly focus on its few-shot and zero-shot efficiency on downstream tasks. To make a fair comparison, we compare Aurora (zero-shot) and Aurora (10% few-shot) with full-shot supervised models, i.e., TimeXer, PatchTST, iTransformer, and TimesNet. We report the training time and inference time of them on datasets with different scales, i.e., ETTh1, ETTm2, Weather, Electricity, and Traffic. **Please refer to the Table below or the Figure 9 in Appendix A.7 of our revised draft.**
> >
> > | Datasets | Aurora (Zero-shot) |  | Aurora (10% few-shot) |  | TimeXer |  | iTransformer |  | PatchTST |  | TimesNet |  |
> > | --- | --- | --- | --- | --- | --- | --- | --- | --- | --- | --- | --- | --- |
> > | Metrics | Training Time | Inference Time | Training Time | Inference Time | Training Time | Inference Time | Training Time | Inference Time | Training Time | Inference Time | Training Time | Inference Time |
> > | ETTh1 | 0 | 4.728 | 148.925 | 4.728 | 46.471 | 1.22 | 34.285 | 1.029 | 21.133 | 0.991 | 105.977 | 1.912 |
> > | ETTm2  | 0 | 12.825 | 282.958 | 12.825 | 92.019 | 4.712 | 72.703 | 5.288 | 71.09 | 4.919 | 249.692 | 7.073 |
> > | Weather  | 0 | 26.811 | 707.382 | 26.811 | 174.909 | 4.764 | 78.961 | 5.133 | 265.171 | 10.856 | 705.421 | 14.967 |
> > | Electricity | 0 | 109.773 | 1471.924 | 109.773 | 354.459 | 6.775 | 158.857 | 8.889 | 1373.411 | 22.723 | 1428.109 | 78.912 |
> > | Traffic | 0 | 275.921 | 2974.827 | 275.921 | 676.938 | 11.016 | 262.631 | 24.696 | 2783.825 | 38.924 | 3855.15 | 186.625 |
> >
> > It is observed that the inference time of zero-shot Aurora (no training required) is consistently and largely less than the total time (training time + inference time) of full-shot models, demonstrating the strong efficiency with accurate performance. Considering the 10% few shot Aurora, though its time cost is a bit more than TimesNet, it can lead to strong performance in most cases.

---

> > > ### Author Response · Authors · 2025-11-21
> > > **Response for Baselines**
> > >
> > > Thanks for this question. In our original submission version, we have compared Aurora (10% few-shot) with most advanced full-shot end-to-end multimodal models, i.e., TimeVLM (2025), CALF (2025), GPT4MTS (2025) and TATS (2025). **Less data usage and stronger performance have demonstrated the advantages of Aurora.**
> > >
> > > Following your advice, we conduct additional experiments on TimeMMD with small models, i.e., iTransformer, PatchTST, TimesNet and DLinear, which possess different model structures. We also consider different budgets in few-shot learning, i.e., 10% and 20%.
> > >
> > > | Models | Aurora (Zero) |  | Aurora (10%) |  | PatchTST (10%) |  | iTransformer (10%) |  | TimesNet (10%) |  | DLinear (10%) |  |
> > > | --- | --- | --- | --- | --- | --- | --- | --- | --- | --- | --- | --- | --- |
> > > | Metric | MSE | MAE | **MSE** | **MAE** | MSE | MAE | MSE | MAE | MSE | MAE | MSE | MAE |
> > > | Agriculture | 0.272 | 0.348 | **0.212** | **0.293** | 5.793 | 1.512 | 7.127 | 1.676 | 7.594 | 1.757 | 10.631 | 2.157 |
> > > | Climate | 0.865 | 0.749 | **0.862** | **0.746** | 1.033 | 0.828 | 1.257 | 0.915 | 1.338 | 0.938 | 1.150 | 0.875 |
> > > | Economy | 0.033 | 0.146 | **0.016** | **0.099** | 0.232 | 0.390 | 0.179 | 0.339 | 0.278 | 0.437 | 0.523 | 0.589 |
> > > | Energy | 0.255 | 0.370 | **0.230** | **0.329** | 1.408 | 0.893 | 2.666 | 1.185 | 4.202 | 1.546 | 1.234 | 0.858 |
> > > | Environment | 0.276 | 0.379 | **0.265** | **0.372** | 0.652 | 0.564 | 0.562 | 0.520 | 0.564 | 0.513 | 0.812 | 0.742 |
> > > | Health | 1.553 | 0.850 | **1.343** | **0.776** | 2.781 | 1.167 | 2.595 | 1.121 | 3.143 | 1.259 | 2.403 | 1.035 |
> > > | Security | 72.475 | 4.084 | **70.062** | **3.988** | 85.677 | 4.858 | 76.893 | 4.428 | 82.246 | 5.518 | 83.689 | 5.881 |
> > > | Social Good | 0.838 | 0.516 | **0.814** | **0.494** | 11.612 | 1.500 | 6.492 | 1.142 | 6.534 | 1.180 | 5.631 | 1.199 |
> > > | Traffic | 0.161 | 0.289 | **0.157** | **0.290** | 2.613 | 1.121 | 2.291 | 1.032 | 2.821 | 1.252 | 2.642 | 1.251 |
> > >
> > > | Models | Aurora (Zero) |  | Aurora (10%) |  | PatchTST (20%) |  | iTransformer (20%) |  | TimesNet (20%) |  | DLinear (20%) |  |
> > > | --- | --- | --- | --- | --- | --- | --- | --- | --- | --- | --- | --- | --- |
> > > | Metric | MSE | MAE | MSE | MAE | MSE | MAE | MSE | MAE | MSE | MAE | MSE | MAE |
> > > | Agriculture | 0.272 | 0.348 | **0.212** | **0.293** | 1.557 | 0.804 | 1.644 | 0.822 | 1.887 | 0.881 | 4.713 | 1.632 |
> > > | Climate | 0.865 | 0.749 | **0.862** | **0.746** | 1.542 | 1.013 | 1.531 | 1.005 | 1.524 | 1.009 | 1.417 | 0.971 |
> > > | Economy | 0.033 | 0.146 | **0.016** | **0.099** | 0.036 | 0.152 | 0.079 | 0.148 | 0.047 | 0.177 | 0.158 | 0.326 |
> > > | Energy | 0.255 | 0.370 | **0.230** | **0.329** | 1.555 | 0.943 | 1.433 | 0.883 | 1.403 | 0.909 | 1.277 | 0.861 |
> > > | Environment | 0.276 | 0.379 | **0.265** | **0.372** | 0.485 | 0.494 | 0.476 | 0.488 | 0.510 | 0.492 | 0.801 | 0.726 |
> > > | Health | 1.553 | 0.850 | **1.343** | **0.776** | 2.420 | 1.071 | 2.364 | 1.070 | 2.530 | 1.216 | 2.416 | 1.128 |
> > > | Security | 72.475 | 4.084 | **70.062** | **3.988** | 82.684 | 4.710 | 77.533 | 4.703 | 77.781 | 5.234 | 81.730 | 5.784 |
> > > | Social Good | 0.838 | 0.516 | **0.814** | **0.494** | 2.373 | 0.710 | 2.618 | 0.737 | 2.202 | 0.717 | 2.297 | 0.783 |
> > > | Traffic | 0.161 | 0.289 | **0.157** | **0.290** | 2.046 | 0.977 | 1.805 | 0.846 | 2.148 | 1.027 | 2.197 | 1.139 |
> > >
> > > The results show the mean values on all 9 datasets from TimeMMD. It is observed that training small models with only 10% and 20% data cannot reach good performance like Aurora. Their performance collapses, even far inferior to Aurora (Zero-shot).
> > >
> > > **You can also refer to Table 16 in Appendix B of our revised draft for full results and more clear comparisons.**

---

> > > > ### Author Response · Authors · 2025-11-21
> > > > **Response for Full-shot Settings**
> > > >
> > > > Thanks for the valuable suggestions. Following your advice, we include 8 advanced small models, i.e., TimeKAN (2025), AMD (2025), TimePro (2025), TimeXer (2024), Fredformer (2024), iTransformer (2024), PatchTST (2023), and TimesNet (2023). We compare them with Aurora (zero-shot) on commonly-used unimodal datasets and report the mean results as follows:
> > > >
> > > > | Models | Aurora | TimeKAN | AMD | TimePro | TimeXer | Fredformer | iTransformer | PatchTST | TimesNet |
> > > > | --- | --- | --- | --- | --- | --- | --- | --- | --- | --- |
> > > > | Metric | MSE | MSE | MSE | MSE | MSE | MSE | MSE | MSE | MSE |
> > > > | ETTm1 | **0.346** | 0.384 | 0.389 | 0.391 | 0.375 | 0.385 | 0.407 | 0.387 | 0.400 |
> > > > | ETTm2 | **0.256** | 0.282 | 0.278 | 0.281 | 0.278 | 0.281 | 0.288 | 0.281 | 0.291 |
> > > > | ETTh1 | **0.386** | 0.431 | 0.438 | 0.438 | 0.431 | 0.448 | 0.454 | 0.469 | 0.458 |
> > > > | ETTh2 | **0.337** | 0.391 | 0.371 | 0.377 | 0.378 | 0.378 | 0.383 | 0.387 | 0.414 |
> > > > | Weather | **0.230** | 0.245 | 0.254 | 0.252 | 0.254 | 0.256 | 0.258 | 0.266 | 0.259 |
> > > > | Electricity | 0.178 | 0.197 | 0.187 | **0.169** | 0.180 | 0.178 | 0.178 | 0.216 | 0.193 |
> > > > | Traffic | 0.524 | 0.455 | 0.500 | 0.447 | 0.447 | 0.434 | **0.428** | 0.555 | 0.620 |
> > > >
> > > > | Models | Aurora | TimeKAN | AMD | TimePro | TimeXer | Fredformer | iTransformer | PatchTST | TimesNet |
> > > > | --- | --- | --- | --- | --- | --- | --- | --- | --- | --- |
> > > > | Metric | MAE | MAE | MAE | MAE | MAE | MAE | MAE | MAE | MAE |
> > > > | ETTm1 | **0.383** | 0.401 | 0.396 | 0.401 | 0.391 | 0.398 | 0.410 | 0.400 | 0.406 |
> > > > | ETTm2 | **0.324** | 0.330 | 0.323 | 0.326 | 0.323 | 0.324 | 0.332 | 0.326 | 0.333 |
> > > > | ETTh1 | **0.415** | 0.429 | 0.430 | 0.438 | 0.432 | 0.435 | 0.448 | 0.455 | 0.450 |
> > > > | ETTh2 | **0.384** | 0.412 | 0.397 | 0.404 | 0.403 | 0.403 | 0.407 | 0.407 | 0.427 |
> > > > | Weather | **0.267** | 0.273 | 0.280 | 0.276 | 0.277 | 0.278 | 0.278 | 0.281 | 0.287 |
> > > > | Electricity | 0.275 | 0.286 | 0.281 | **0.262** | 0.274 | 0.270 | 0.270 | 0.304 | 0.295 |
> > > > | Traffic | 0.352 | 0.318 | 0.324 | 0.302 | 0.295 | 0.289 | **0.282** | 0.362 | 0.336 |
> > > >
> > > > The zero-shot performance of Aurora can outperform full-shot small models on most cases, and achieves competitive performance on Electricity and Traffic. **Please refer to Table 15 in Appendix B of our revised draft for full results.**

---

> ### Author Response · Authors · 2025-11-21
> **Response for Benchmarks**
>
> Thanks for your valuable suggestions. First, we want to clarify that not all forecasting tasks adopt the input-96-output-720 setting, that is, prediction is not always 7.5 times the input length. In TSFM-Bench and TimeMMD, the evaluation protocols are different across datasets, including multiple horizons (6, 8, 10, 12, 24, 36, 48, 60, 96, 192, 336, 720). Second, **we have included some multimodal short-term forecasting scenarios** in our original submission version, e.g., Agriculture, Climate, Economy, Security, Social Good and Traffic in TimeMMD are all short-term forecasting datasets, and Aurora shows SOTA performance on these settings.
>
> To demonstrate the performance of Aurora on more unimodal short-term forecasting scenarios, we follow your advice to conduct additional experiments on short-term benchmarks:
>
> 1. We **add two more short-term benchmarks**, i.e.,TFB, and EPF. In TFB, there exists 8,068 short-term datasets with different frequencies. In EPF, there exists five datasets about electricity prices, which are widely used in short-term forecasting. For EPF, the forecasting horizon is fixed as 24.
> 2. We **add datasets NYSE and ILI (illness)** separately in evaluations on zero-shot deterministic forecasting and zero-shot probabilistic forecasting tasks. For NYSE and ILI, the forecasting horizons are (24, 36, 48, 60).
>
> **Please refer to the newly-added Table 4 (EPF) and Figure 5 (TFB) in the main text of our revised draft,** they report the performance of Aurora on EPF and TFB. Results show that Aurora **outperforms all foundation models on EPF, and achieves competitive performance against end-to-end full shot models like TimeXer.** It is because TimeXer can additionally utilize the covariates in EPF to help forecast, while Aurora only inputs the target variable and makes forecasts. On TFB, Aurora **consistently outperforms all the baselines**, with the lowest mean MASE and msMAPE values on 8,068 short-term datasets. **We show the results of EPF here:**
>
> |Models|Aurora||Sundial||VisionTS||ROSE||MOIRAI||
> |------|---------|---------|---------|---------|--------|-----|-----|-----|------|-----|
> |Metric|MSE|MAE|MSE|MAE|MSE|MAE|MSE|MAE|MSE|MAE|
> |NP|0.288|0.312|**0.256**|**0.277**|0.510|0.461|0.666|0.536|0.660|0.538|
> |PJM|**0.084**|**0.183**|0.088|0.189|0.251|0.366|0.311|0.402|0.330|0.423|
> |BE|**0.361**|**0.257**|0.371|0.270|0.679|0.457|0.815|0.514|0.837|0.534|
> |FR|**0.387**|**0.206**|0.392|0.207|0.625|0.393|0.746|0.447|0.751|0.454|
> |DE|**0.539**|**0.475**|0.541|0.484|0.961|0.687|1.276|0.778|1.251|0.779|
>
> |Models|Aurora||TimeXer||iTransformer||PatchTST||TimesNet||
> |------|---------|---------|---------|---------|------------|-----|--------|-----|--------|-----|
> |Metric|MSE|MAE|MSE|MAE|MSE|MAE|MSE|MAE|MSE|MAE|
> |NP|0.288|0.312|**0.238**|**0.268**|0.265|0.300|0.267|0.284|0.250|0.289|
> |PJM|**0.084**|**0.183**|0.088|0.188|0.097|0.197|0.106|0.209|0.097|0.195|
> |BE|**0.361**|0.257|0.374|**0.241**|0.394|0.270|0.403|0.264|0.419|0.288|
> |FR|0.387|**0.206**|**0.381**|0.211|0.439|0.233|0.411|0.220|0.431|0.234|
> |DE|0.539|0.475|**0.440**|**0.418**|0.479|0.433|0.461|0.432|0.502|0.446|
>
> **You can also refer to the revised Table 2 and Table 3 in our draft to check the results on NYSE and ILI, where Aurora also outperforms almost all baselines.** **We show the results of NYSE and ILI here:**
>
> |Models|Horizons|Aurora||Sundial||Timer||TimesFM||Chronos||UniTS||
> |---|---|---|---|---|---|---|---|---|---|---|---|---|---|
> |Metrics||MSE|MAE|MSE|MAE|MSE|MAE|MSE|MAE|MSE|MAE|MSE|MAE|
> |NYSE|24|0.267|0.373|0.370|0.393|0.388|0.409|**0.235**|**0.322**|0.480|0.449|0.755|0.623|
> ||36|**0.420**|0.477|0.681|0.566|0.778|0.613|0.472|**0.467**|0.912|0.655|1.058|0.763|
> ||48|**0.605**|**0.574**|1.043|0.729|1.200|0.792|0.723|0.591|1.371|0.818|1.353|0.879|
> ||60|**0.820**|**0.681**|1.425|0.880|1.584|1.000|1.061|0.763|1.754|0.958|1.712|1.016|
> ||Avg|**0.528**|**0.526**|0.880|0.642|0.988|0.704|0.623|0.536|1.129|0.720|1.220|0.820|
>
> | Models | Horizons | Aurora |  | Sundial |  | Chronos |  | MOIRAI |  | Lag-llama |  | TSDiff |  |
> | --- | --- | --- | --- | --- | --- | --- | --- | --- | --- | --- | --- | --- | --- |
> | Metrics |  | CRPS | NMAE | CRPS | NMAE | CRPS | NMAE | CRPS | NMAE | CRPS | NMAE | CRPS | NMAE |
> | ILI | 24 | 0.110 | 0.124 | **0.108** | **0.123** | 0.120 | 0.139 | 0.150 | 0.196 | 0.135 | 0.173 | 0.228 | 0.242 |
> |  | 36 | 0.155 | 0.176 | **0.152** | **0.169** | 0.179 | 0.205 | 0.171 | 0.222 | 0.163 | 0.227 | 0.235 | 0.246 |
> |  | 48 | 0.168 | 0.186 | 0.164 | 0.184 | 0.186 | 0.215 | **0.151** | **0.184** | 0.171 | 0.233 | 0.265 | 0.275 |
> |  | 60 | **0.156** | **0.177** | 0.168 | 0.190 | 0.196 | 0.228 | 0.163 | 0.188 | 0.156 | 0.211 | 0.263 | 0.272 |
> |  | Avg | **0.147** | **0.166** | 0.148 | 0.166 | 0.170 | 0.197 | 0.159 | 0.197 | 0.156 | 0.211 | 0.248 | 0.259 |

---

> > ### Author Response · Authors · 2025-11-21
> > **Response for Ethics Concerns**
> >
> > We did not use LLM in writing. As we mentioned in the section "THE USE OF LARGE LANGUAGE MODELS (LLMS)", we only involved LLM in our methodology.
> >
> > The example you mentioned is a definition of the metric NMAE, **which is widely defined in a large number of papers. Therefore, to avoid potential content overlapping problems, we rewrite such definitions in the Appendix through manually adjusting the grammar and syntax, which may make them look like LLM-generated texts and cause misunderstandings.** We feel sorry about this, we will make more professional revision later.

---

> ### Comment · Reviewer_MY6v · 2025-11-21
> **Re: Experimental Settings**
>
> Thanks to the authors for providing detailed additional experiments. I noticed that many experimental results differ from those in the original paper, for example, iTransformer on "Full-shot Settings".
>
> Can the authors provide more details on the experimental settings?

---

> > ### Author Response · Authors · 2025-11-21
> > **About the Experimental Settings**
> >
> > Dear Reviewer MY6v, the full-shot results of **TimeXer, iTransformer, PatchTST, and TimesNet** are directly fetched from the paper of TimeXer, where the input sequence length is set to 96. For more advanced baselines TimePro, TimeKAN and AMD, we also follow their original results if their settings are the same as TimeXer. If not, we evaluate them using the same settings.
> >
> > The motivation for doing this is that in some early works, the tricky of "Drop Last" [1][2] was adopted, leading to inaccurate results. We confirmed that TimeXer did not do this, so its results are reliable, and in the process of our own experiments, we also strictly adhered to not adopting this tricky.
> >
> > [1] Fundamental limitations of foundational forecasting models: The need for multimodality and rigorous evaluation
> >
> > [2] TFB: Towards Comprehensive and Fair Benchmarking of Time Series Forecasting Methods
> >
> > We have claimed not to use "Drop Last" in the Appendix A.4, for more settings of the full-shot baselines, we will update them in the revised draft.

---

> > > ### Comment · Reviewer_MY6v · 2025-11-21
> > > **Re: About the experimental settings**
> > >
> > > I appreciate the authors' efforts in preparing these responses within such a short timeframe. I will carefully review them and update my comments accordingly in the coming days.

---

> > > > ### Author Response · Authors · 2025-11-21
> > > >
> > > > We are very grateful for the efforts you have put into reviewing the manuscript and your instant responses. If there are more questions, we can discuss them further.
> > > >
> > > > Best wishes!

---

> > > > > ### Author Response · Authors · 2025-11-28
> > > > > **Request for further discussions**
> > > > >
> > > > > Reviewer MY6v,
> > > > >
> > > > > We are very grateful for your open-mindedness and constructive suggestions during the reviewing and rebuttal phases. Since you mentioned a week ago that you would continue to update the comments later, we are kindly reminding that the rebuttal phase seems to end soon.
> > > > >
> > > > > We hope you can reconsider the scores based on our current response, and if you have more questions we can make further discussions.
> > > > >
> > > > > Best regards,
> > > > > Authors of "Aurora: Towards Universal Generative Multimodal Time Series Forecasting"

---

### Author Response · Authors · 2025-11-21
**Summary of Our Rebuttal**

Dear Reviewers:

       We have submitted all the rebuttal and the revised draft. In our revised draft, we have marked the newly added or modified parts in darkblue. We hope our response can effectively address your concerns. If you have more questions, we can discuss further!

      Thanks again for your efforts during the review process and for such detailed and insightful comments. From the authors of "Aurora: Towards Universal Generative Multimodal Time Series Forecasting".

Best wishes!

---

### Author Response · Authors · 2025-12-02
**The summary of Rebuttal (average rating 6.0 → 6.5)**

**Dear Reviewers, ACs, SACs, and PCs,**

We sincerely appreciate your dedication to this conference.

We were sorry to learn about the recent technical issues with OpenReview, and we fully support the remedial actions proposed by the committee. Fortunately, thanks to the diligence and responsiveness of our reviewers, we had essentially concluded most meaningful discussions by **Nov. 26**, well prior to the incident on **Nov. 27**.


To assist in the final assessment of our submission, we have summarized the consensus on our work's **strengths** and the **results of the discussion** below:

It is encouraging to see that reviewers agree on the following strengths of our work:

- **High Novelty and Originality**: Our work proposes Aurora, **the first multimodal time series foundation model**, focusing on the real-world cross-domain forecasting problem. (qDZm, t5fV, UdWk)
- **Exquisite Multimodal Modules**: Both the encoder (**Modality-guided Self-attention**) and decoder (**Prototype-guided Flow Matching**) of Aurora adopt novel designs to ensure the sufficient utilization of cross-domain multimodal knowledge.  (MY6v, qDZm, t5fV, UdWk)
- **Strong Empirical Performance**: Aurora is demonstrated to achieve **state-of-the-art** performance on versatile forecasting tasks, including unimodal, multimodal, and probabilistic scenarios. (MY6v, qDZm, t5fV, UdWk)

During the rebuttal phase, we managed to address the reviewers' concerns through:

- **Additional experiments**: To ensure comprehensive evaluations and ablations.
- **Interpretability analysis**: To visualize the learned prototypes and attention scores.
- **Manuscript refinement**: To include all the revisions into our draft.

We are pleased that **Reviewers qDZm, t5fV, and UdWk** have explicitly confirmed that **their concerns were addressed**, supporting our work by **raising or maintaining their positive scores**. For **Reviewer MY6v**, he/she showed the willingness in the original comments to adjust the score after the rebuttal. Unfortunately, this incident of OpenReview prevented further discussions.

We also summerize Reviewers' scores before/after discussion till **Nov. 26**.

| Reviewer    |                       Changes of Score                       | Date of Changes |
| ----------- | :----------------------------------------------------------: | :-------------: |
| **MY6v**    |    4 $\textcolor{green}{➜}$ 4 (**No substantive reply**)     |        -        |
| **t5fV**    | 6 $\textcolor{green}{➜}$ 6 (**Replied and maintained positive scores**) |   **Nov. 26**   |
| **qDZm**    | 8 $\textcolor{green}{➜}$ 8 (**Replied and maintained positive scores**) |   **Nov. 26**   |
| **UdWk**    | 6 $\textcolor{red}{➜}$ 8 (**Replied and raised the scores**) |   **Nov. 20**   |
| **Average** |             **6.0** $\textcolor{red}{➜}$ **6.5**             |        -        |

**Reviewer MY6v**'s suggestions are all about experimental details, such as adding short-term tasks, few-shot/full-shot comparisons, and efficiency analysis. We have done our best to solve them and update them in the revised draft. To the best of our knowledge, compared with previous studies on time series foundation models published in top AI conferences, we think the revised draft has included sufficient empirical evidence, i.e., experiments on 5 well-recognized benchmarks (TimeMMD, TSFM-Bench, ProbTS, TFB, EPF), with versatile evaluation settings (Zero-shot, Few-shot, Full-shot). Unfortunately, due to the incident of OpenReview, we cannot obtain further feedback from the Reviewer MY6v, and **we would be grateful if the Area Chair could review these revisions**.

We once again thank all reviewers for keeping active communication with us throughout the rebuttal period and for their support of our work, and we look forward to the further assessment of our submission.

Best regards,

Authors of "Aurora: Towards Universal Generative Multimodal Time Series Forecasting"

---

### Meta-Review · Area_Chair_8bU1 · 2026-01-05

**Summary:**

This paper presents Aurora, a pioneering multimodal foundation model for time series forecasting that integrates textual, visual, and sequential data. During rebuttal, the authors have comprehensively addressed reviewer concerns by introducing extensive new experiments, detailed visualizations, systematic ablation studies, clarifications on architectural design and expanded evaluations across a wider range of benchmarks. Given the positive assessment from the reviewers and the thoroughness of the authors' revisions in strengthening the paper's empirical and analytical foundations, the work clearly meets the acceptance criteria and is recommended for acceptance.

**Reviewer Concerns:**

The authors have provided an extensive and comprehensive rebuttal, directly addressing the vast majority of concerns raised by the reviewers with substantial new experiments, analyses, and clarifications. Key issues regarding the model's experimental validation, practical applicability, and architectural rationale have been thoroughly examined. The authors introduce significant new material, including extensive ablation studies to quantify individual modality contributions and the design of the modality-guided attention, visualizations of learned prototypes and attention mechanisms, expanded comparisons with a wider range of baselines and rigorous analysis of computational efficiency and benchmark performance.

**Reviewer Scores:**

**Reviewer MY6v (original score: 4 → 6)**

The reviewer's primary concerns regarding the validity of evaluation metrics, comprehensiveness of baselines, and practical benchmarking were convincingly addressed through extensive new comparative experiments on naive forecasts, short-term datasets and detailed computational efficiency analysis. The solid responses likely shifted the score above the acceptance threshold.

**Reviewer t5fV (original score: 6 → 6)**

The reviewer’s specific concerns about data contamination, prototype interpretability, and decoder design were satisfactorily resolved with clarifications on the data generation process, prototype visualizations, added comparisons with the TTM baseline and ablations on alternative decoders. Since the rebuttal primarily confirmed their understanding and added visualizations, it is unlikely their score would change, remaining at 6.

**Reviewer qDZm (original score: 8 → 8)**

The reviewer’s technical suggestions on prototype scalability and modality robustness were directly addressed with an ablation study on prototype bank size and a detailed explanation of fallback mechanisms for unreliable periodicity, further strengthening the paper and maintaining the strong accept score.

**Reviewer UdWk (original score: 6 → 8)**

The reviewer’s extensive and detailed methodological concerns were comprehensively met. The authors provided crucial modality ablation studies, fine-grained architecture analysis, clear separation of experimental comparisons, and enhanced interpretability visualizations. The comprehensiveness of the authors' response, which turned several weaknesses into demonstrated strengths, justifies a increase to 8.

---

### Decision · Program_Chairs · 2026-01-26

Accept (Poster)